# Nuclear genetic control of mtDNA copy number and heteroplasmy in humans

Rahul Gupta[1,2,3 ✉], Masahiro Kanai[2,3], Timothy J. Durham[1,2], Kristin Tsuo[2,3], Jason G. McCoy[1,2], Anna V. Kotrys[1,2], Wei Zhou[2,3], Patrick F. Chinnery[4,5], Konrad J. Karczewski[2,3], Sarah E. Calvo[1,2], Benjamin M. Neale[2,3,7 ✉] & Vamsi K. Mootha[1,2,6,7 ✉]

Mitochondrial DNA (mtDNA) is a maternally inherited, high-copy-number genome required for oxidative phosphorylation[1]. Heteroplasmy refers to the presence of a mixture of mtDNA alleles in an individual and has been associated with disease and ageing. Mechanisms underlying common variation in human heteroplasmy, and the influence of the nuclear genome on this variation, remain insufficiently explored. Here we quantify mtDNA copy number (mtCN) and heteroplasmy using blood-derived whole-genome sequences from 274,832 individuals and perform genome-wide association studies to identify associated nuclear loci. Following blood cell composition correction, we find that mtCN declines linearly with age and is associated with variants at 92 nuclear loci. We observe that nearly everyone harbours heteroplasmic mtDNA variants obeying two principles: (1) heteroplasmic single nucleotide variants tend to arise somatically and accumulate sharply after the age of 70 years, whereas (2) heteroplasmic indels are maternally inherited as mixtures with relative levels associated with 42 nuclear loci involved in mtDNA replication, maintenance and novel pathways. These loci may act by conferring a replicative advantage to certain mtDNA alleles. As an illustrative example, we identify a length variant carried by more than 50% of humans at position chrM:302 within a G-quadruplex previously proposed to mediate mtDNA transcription/replication switching[2,3]. We find that this variant exerts *cis*-acting genetic control over mtDNA abundance and is itself associated in-*trans* with nuclear loci encoding machinery for this regulatory switch. Our study suggests that common variation in the nuclear genome can shape variation in mtCN and heteroplasmy dynamics across the human population.

Human mitochondria contain a tiny, high-copy-number circular genome (mitochondrial DNA (mtDNA)). Sequencing of the human mtDNA in 1981 (ref. 1) revealed that it encodes 13 core protein components of the oxidative phosphorylation system, as well as 2 ribosomal RNAs and 22 transfer RNAs required for their expression. Tissues can contain tens to thousands of copies of mtDNA per cell, depending on cell type. Variants in mtDNA can be maternally transmitted or arise somatically, and, when they co-exist with wild-type molecules, lead to a state called heteroplasmy. Notably, more than 99% of the mitochondrial proteome, including all proteins required for mtDNA maintenance, replication and transcription, is encoded by the nuclear DNA (nucDNA) and imported[4] into the organelle.

Defects in mtDNA are associated with a spectrum of human diseases. Since the first identification of pathogenic mtDNA mutations[5,6], scores of maternally inherited syndromes have been reported[7]. Mendelian forms of mitochondrial disease producing mtDNA deletion or depletion were later identified and mapped to nuclear genes involved in mtDNA replication, maintenance and nucleotide balance[8–10]. More subtle declines in mtDNA copy number (mtCN) and an accumulation of somatic mtDNA mutations have both long been associated with ageing and age-associated disease[11,12]. Mutations in mtDNA accumulate in many cancers and in a small subset are even considered to be 'drivers' of tumorigenesis[13].

The dynamics of heteroplasmy are complex and presumed to be shaped by mutation, drift and selection. The mtDNA mutation rate has been reported as 10–100× higher than for the nucDNA[14], with the non-coding region (NCR) containing three hypervariable regions thought to be mutational hotspots[15]. The high copy number, elevated substitution rate and lack of recombination have made mtDNA NCR variants a valuable genetic tool in forensics and anthropology, even leading to the African mitochondrial 'Eve' hypothesis[16,17]. Heteroplasmy can vary across siblings, attributed to germline bottleneck effects, and between cell types and tissues, thought to be due to random segregation and selection[18,19]. Detailed mechanisms underlying heteroplasmy dynamics in humans remain obscure, although mouse studies[20] predict a role for nuclear genetics.

[1]Howard Hughes Medical Institute and Department of Molecular Biology, Massachusetts General Hospital, Boston, MA, USA. [2]Broad Institute of MIT and Harvard, Cambridge, MA, USA. [3]Analytic and Translational Genetics Unit, Center for Genomic Medicine, Massachusetts General Hospital, Boston, MA, USA. [4]Department of Clinical Neurosciences, University of Cambridge, Cambridge, UK. [5]MRC Mitochondrial Biology Unit, University of Cambridge, Cambridge, UK. [6]Department of Systems Biology, Harvard Medical School, Boston, MA, USA. [7]These authors jointly supervised this work: Benjamin M. Neale, Vamsi K. Mootha. ✉e-mail: rahul_gupta@hms.harvard.edu; bneale@broadinstitute.org; vamsi_mootha@hms.harvard.edu

Here, we characterize the spectrum of mtCN and heteroplasmy across approximately 300,000 individuals spanning 6 ancestry groups in the UK Biobank (UKB) and AllofUs (AoU). We find that blood mtCN declines with age, is influenced by blood cell composition and is under the control of numerous nuclear genetic loci. We then turn to mtDNA variation, finding that about 1 in 192 individuals carries 1 of 10 well-known pathogenic mtDNA variants. We characterize the landscape of mtDNA variation across this population and find that nearly every human harbours heteroplasmic mtDNA variants. Whereas heteroplasmic mtDNA single nucleotide variants (SNVs) tend to be somatic in origin and to accumulate with age, we find that heteroplasmic indels tend to be quantitatively maternally inherited, with their relative levels influenced by nuclear genetic variation. These loci provide insights into the mechanisms by which the mitochondrial and nuclear genomes genetically interact to maintain mtDNA homeostasis.

## Calling mtCN and variants

We developed mtSwirl, a scalable pipeline for calling mtDNA variants and copy number from whole-genome sequencing (WGS) data (Methods and Supplementary Note 1). We extended a pipeline used to analyse mtDNA variation in gnomAD[21], now constructing self-reference sequences for each sample using homoplasmic and homozygous calls on the mtDNA and reference nucDNA regions of mtDNA origin (NUMTs; Extended Data Fig. 1a). mtSwirl shows improved mtDNA coverage, particularly among African haplogroups (Extended Data Fig. 1b–e), and reduced variant calls at very low heteroplasmy (Extended Data Fig. 1f), indicating reduced ancestry- and NUMT-specific mis-mapping. We observe high concordance of heteroplasmy estimates with the previous method used in gnomAD ($R^2 = 0.996$ for heteroplasmy > 0.05), with homoplasmies showing allele fractions now closer to 1, suggesting reduced NUMT artefact[21] (Extended Data Fig. 1g). We used mtSwirl to quantify mtDNA traits across 274,832 individuals of diverse ancestry across UKB and AoU (Extended Data Fig. 2 and Supplementary Table 1), generating more than 7,800,000 mtDNA variant calls across all samples.

## Determinants of variation in human mtCN

We began by identifying covariates of blood mtCN (mtCN$_{raw}$) in UKB, observing a strong influence of blood cell composition ($R^2 \approx 23\%$; Fig. 1a) as previously reported[22,23] (Extended Data Fig. 3c). We identified several more unexpected covariates including time of day, month of year and fasting duration ($R^2 \approx 2.5\%$; Fig. 1a and Extended Data Fig. 3e–j). Following adjustment for all identified covariates (Methods and Supplementary Notes 2 and 3), we found that covariate-adjusted mtCN (which we term mtCN$_{adj}$) was unimodal in UKB across 178,134 subjects with an average of 61.66 copies per diploid nuclear genome (Extended Data Fig. 3d). We observed a linear decline in mtCN$_{adj}$ with age (Fig. 1c) of approximately 2% per decade among both males and females.

We next assessed the degree to which variation in mtCN$_{adj}$ is under nuclear genetic control. Our genome-wide association study (GWAS) identified 92 linkage disequilibrium (LD)-independent nucDNA association signals across 46 loci (Fig. 1d) after cross-ancestry meta-analysis, with an estimated SNP-heritability of approximately 4% (Methods). By contrast, mtDNA haplogroup explained less than 0.5% of the variance in mtCN$_{adj}$, with only a few associations of small magnitude observed (Extended Data Fig. 4a,b). Thirty-three nuclear loci showed variants with a posterior inclusion probability (PIP) of 0.1 or greater after fine-mapping (Methods); 11 of these had protein-altering variants in the 95% credible set (CS) at PIP > 0.1 (Fig. 1e) and 7 showed expression quantitative trait locus (eQTL) colocalization with the assigned gene at PIP > 0.1, including *TFAM*, *MFN2*, *NDUFV3* and *RRM1*. Eight loci contained genes implicated in disorders of mtDNA maintenance, six of which harboured variants with PIP > 0.1. Prioritized genes (Methods) encoded proteins that participate in the mtDNA nucleoid and replisome

(*TFAM*, *POLG2*, *TWINKLE*, *TOP1MT*, *LONP1*), nucleotide metabolism (*RRM1*, *RRM2B*, *DGUOK*, *AK3*, *SLC25A5*) and mitochondrial fusion (*MFN1*, *MFN2*). The *PNP–APEX1* locus was notable as these adjacent genes encode proteins in nucleotide metabolism and mtDNA repair, neither of which has been implicated in mtCN control. Fine mapping implicated both genes, even identifying a missense variant in *APEX1* at PIP > 0.9 (Extended Data Fig. 5a). Several more loci included mitochondrial proteins with no previous links to mtDNA (*SLC25A10*, *MCAT*, *NDUFV3*). Telomerase (*TERT*) is in the vicinity of one locus; however, fine mapping did not provide further evidence for its causality (Supplementary Table 3).

We also performed a gene-based rare variant association study (RVAS) for mtCN$_{adj}$ in UKB (Methods and Supplementary Table 7). In several instances we find convergence with our GWAS, including associations with ultra-rare (minor allele frequency (MAF) < 0.0001) missense or loss of function (LoF) variation in *TWNK* and *TFAM* (Extended Data Fig. 5c). RVAS provided clarity to other GWAS loci with uncertain gene assignments (for example, highlighting *TOP3A* in a locus containing several genes; Fig. 1d) and identified several associations with genes not identified by GWAS. For instance, we found associations with the burden of rare protein-altering variation in genes previously linked to Mendelian mtDNA deletion or depletion disease (*OMA1*, *SAMHD1*), as well as associations with genes unlinked to mitochondria (for example, *MILR1*) (Extended Data Fig. 5d).

We next tested mtCN$_{adj}$ for heritability enrichment in genes associated with organelles or organs using stratified LD-score regression[24–26] (S-LDSC; Methods). The most significant organelle enrichment was seen for the mitochondrion (Extended Data Fig. 4c). Across organs, skeletal muscle and whole blood were top scoring (Extended Data Fig. 4d). Whole blood enrichment is expected given the sampling site, but skeletal muscle enrichment was unexpected and may be due to shared patterns of gene expression between blood and muscle, or could indicate non-cell autonomous control of blood mtCN.

## Blood composition influences bulk mtCN

Although many previous studies have reported associations between low blood mtCN and common diseases[27–30], we could not replicate these results using mtCN$_{adj}$ in UKB for type 2 diabetes, myocardial infarction, stroke, hypertension or dementia (Fig. 1f). When we repeated this analysis using mtCN$_{raw}$, that is, without adjusting for blood composition, we could recover these earlier associations (Fig. 1f). We extended these analyses to 24 more common diseases, finding that, in total, 20 showed significantly increased risk with reduced mtCN$_{raw}$; after correction for blood cell composition, the inverse correlations disappeared for all traits except osteoarthritis (Extended Data Fig. 3k). Associations with four cardiovascular disease traits even changed direction with mtCN$_{adj}$, now showing a positive correlation with increased risk. In all five cases, Mendelian randomization did not support a causal role for mtCN$_{raw}$ or mtCN$_{adj}$ after correcting for multiple tests (Extended Data Fig. 6). Even the oft-reported elevated mtCN in females[31] appears to be largely driven by blood composition (Fig. 1b,c). Our GWAS analyses also underscore the confounding effects of blood composition in previous work. Using mtCN$_{adj}$, we could replicate (at $P < 5 \times 10^{-5}$) 70 of the 96 previously reported mtCN GWAS loci[32], with 37 at genome-wide significance (GWS) (Methods). Using mtCN$_{raw}$, we could recover 12 more loci from this previous study at GWS including loci containing *HBS1L-MYOB*, *C2*, *HLA*, *GSDMC* and *CD226*, which are linked to blood cell types and inflammation (Extended Data Fig. 4f). By contrast, associations near *TFAM*, a well-known mtCN-controlling gene[33], encouragingly strengthen by about 40 orders of magnitude following blood composition adjustment.

It has long been known that inflammation is associated with cardiometabolic disease[34]; indeed, elevations in inflammatory blood cell indices predict elevated risk for 25 of 29 tested diseases in UKB (Fig. 1f and Extended Data Fig. 3l). Bidirectional Mendelian randomization

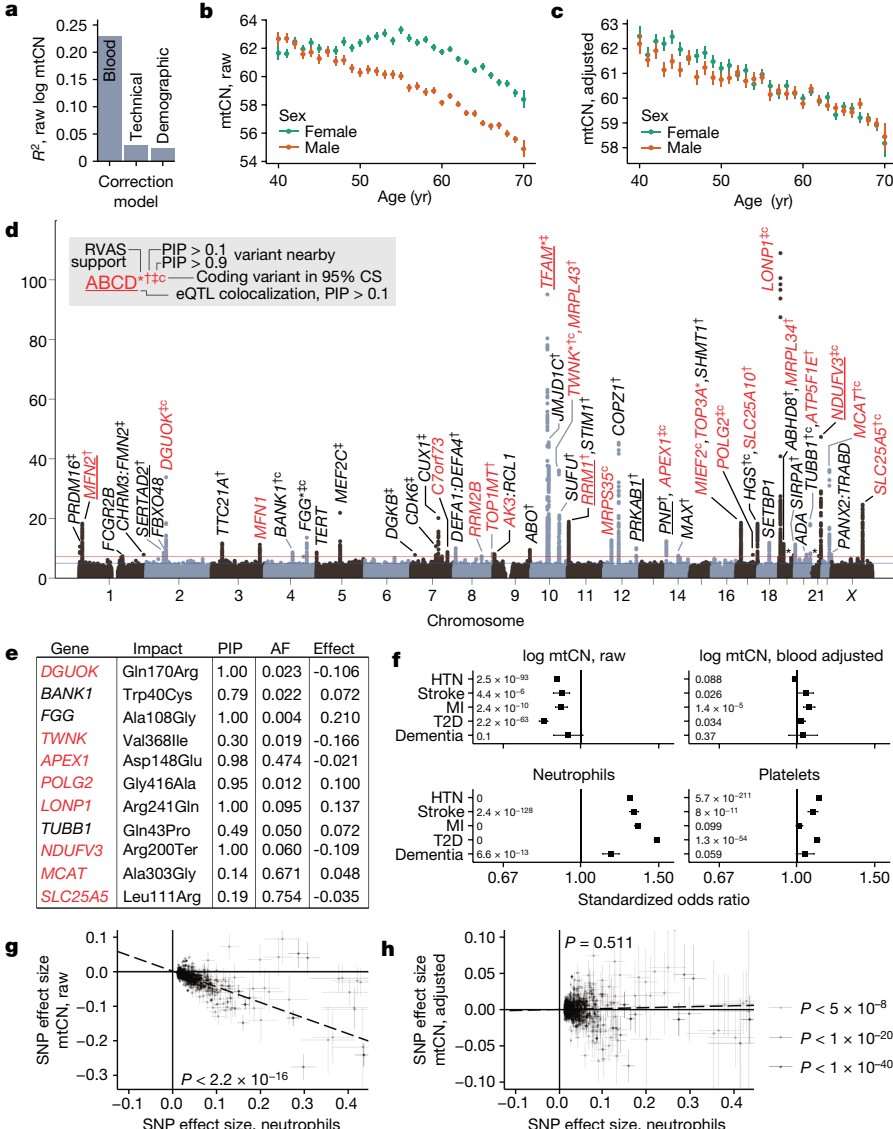

**Fig. 1 | Genetic and phenotypic determinants of mtCN in UKB. a**, Variance explained in mtCN by correction models. **b**,**c**, Mean mtCN$_{raw}$ (**b**) and mtCN$_{adj}$ (**c**) as a function of age and genetic sex. For **b** and **c**, mtCN is copies per diploid nuclear genome, error bars are mean ± 1 s.e.m., and total $n$ = 178,129 and 164,798, respectively. **d**, GWAS Manhattan plot from UKB cross-ancestry meta-analysis. Labelled genes were obtained using fine-mapping, rare variant evidence or nearest gene. Red genes encode mitochondrial proteins or are implicated in mtDNA disease; *gene at GWS for the Cauchy $P$ value from RVAS; †CS variants proximal to the gene with PIP > 0.1; ‡CS with PIP > 0.9; 'c', coding variant in the CS; underline, eQTL colocalization PIP > 0.1. Asterisks above peaks on chr 19 and 21 correspond to GP6 and RUNX1, respectively. **e**, Variants

in the 95% CS with PIP > 0.1 causing a protein-altering change. **f**, Standardized odds ratios for log mtCN$_{raw}$, log mtCN$_{adj}$ and major blood composition phenotypes in predicting risk of selected common diseases in UKB. Inset numbers are two-sided raw $P$ values with Bonferroni $P$ value cut-off = 0.0025; error bars are 95% confidence intervals (95% CIs) around odds ratios (ORs); sample sizes are in Supplementary Table 8. HTN, hypertension; MI, myocardial infarction; T2D, type 2 diabetes. **g**,**h**, Correlations between effect sizes for lead SNPs detected at GWS for neutrophil count between neutrophil count and mtCN$_{raw}$ ($P$ = 4.4 × 10$^{-73}$) (**g**) and mtCN$_{adj}$ ($P$ = 0.511) (**h**). Error bars represent 1 s.e., dotted line is weighted least squares regression line, inset corresponds to regression $P$ value. AF, allele frequency.

showed that effect sizes of loci at GWS for neutrophil count were strongly negatively correlated with corresponding mtCN$_{raw}$ effect sizes (Fig. 1g), whereas the converse did not convincingly hold (Extended Data Fig. 4g), suggesting that changes in blood cell composition cause mtCN$_{raw}$ changes rather than the reverse. Importantly, neutrophil count effect sizes did not predict corresponding mtCN$_{adj}$ effect sizes (Fig. 1h and Extended Data Fig. 4h).

The most parsimonious explanation for our observations is that previously reported associations between low blood mtCN and elevated common disease risk are, in many cases, secondary to blood composition changes. For the few associations that survive blood composition corrections (Extended Data Fig. 3k), other mechanisms may be involved.

Indeed, Mendelian randomization suggests reverse causation or shows high heterogeneity for these traits, arguing against simple forward causal relationships in these instances (Extended Data Fig. 6).

## Nuclear control over mtDNA 7S coverage

We next aimed to use variation in sequencing coverage across the 16,569 bases of the mtDNA to dissect specific molecular mechanisms of mtDNA replication. We observe a coverage dip by over 50% in the major NCR of the mtDNA (Fig. 2a), which contains the light strand promoter (LSP), three conserved sequence blocks (CSBs), the heavy strand origin of replication (O$_H$) and the D-loop, which contains a stable third

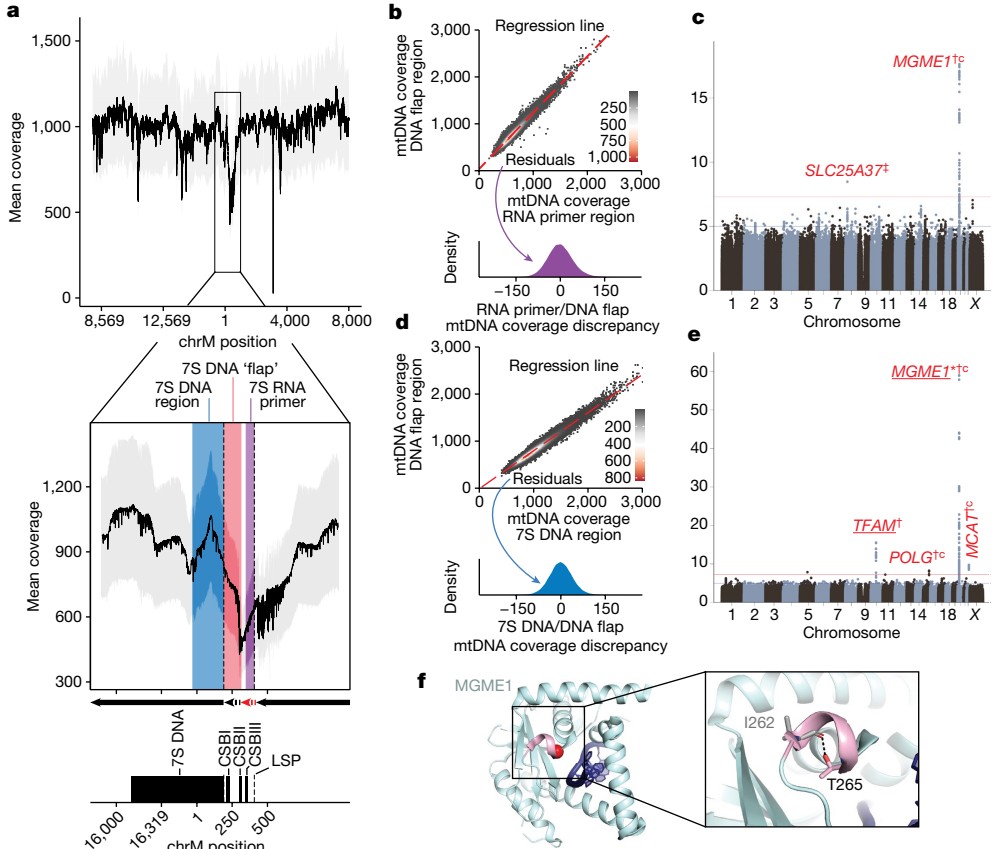

**Fig. 2 | Nuclear genetic control of relative mtDNA coverage in the NCR.**
**a**, Mean UKB mtDNA per-base coverage. Dropdown highlights coverage depression in the mtDNA NCR. Arrows refer to mtDNA replication products: red dashed arrow, RNA primer; black dashed arrow, transient DNA 'flap'; black solid arrow, replicated mtDNA. Grey ribbon is ±1 s.d. CSB, conserved sequence box. **b**, Two-dimensional (2D) histogram showing mtDNA coverage in the DNA flap region versus RNA primer region. Red line is linear fit, from which the residual is used as a 'coverage discrepancy'. The distribution of these residuals is shown in the lower panel. **c**, GWAS Manhattan plot of the discrepancy of mtDNA coverage in the DNA flap region versus RNA primer region (see **b**). **d**, 2D histogram showing mtDNA coverage in the DNA flap region versus 7S DNA

region. As in **b**, red line is linear fit, and the residual is shown as a density in the lower panel. **e**, GWAS Manhattan plot of the discrepancy of mtDNA coverage in the DNA flap region versus 7S DNA region (see **d**). Red genes are mitochondria-relevant; *gene with Cauchy $P$ value at GWS from RVAS; †CS variants proximal to the gene with PIP > 0.1; ‡proximal CS variants with PIP > 0.9; 'c', missense variant identified in the CS; underline, eQTL colocalization with PIP > 0.1. **f**, Structure of MGME1 (5ZYV from RSCB under CC0 license; https://doi.org/10.2210/pdb5zyv/pdb) with bound single-stranded DNA in dark blue, the $3_{10}$ helix in pink and the T265 alpha carbon as a red sphere. Inset shows the hydrogen bond between T265 and I262.

strand of DNA (7S DNA) (Extended Data Fig. 7). It is believed that mtDNA replication requires an 'RNA primer' which forms from the termination of LSP-initiated transcription at CSBII (red dashed arrow, Fig. 2a inset). Primed mtDNA synthesis begins at CSBII, with the nascent DNA between CSBII and $O_H$ forming a transient 'DNA flap' (black dashed arrow, Fig. 2a inset). Further replication can either continue to full-length or be terminated prematurely to produce the persistent 7S DNA (black solid arrow, Fig, 2a inset; see also ref. 35). In theory, we expect the highest local WGS coverage in the persistently triple-stranded 7S DNA, lower coverage in the transiently triple-stranded DNA flap region and lowest coverage in the RNA primer region. This is what we observe (Fig. 2a).

We hypothesized that genetic variation in nuclear-encoded mtDNA replication machinery might influence the tendency of replication intermediates in the NCR to persist. To attempt to quantify these intermediates, we computed the discordance in coverage between these three regions across individuals in UKB (that is, residuals; Fig. 2b,d and Methods). Upon performing GWAS and cross-ancestry meta-analysis for these traits, we find that nuclear genetic variants near *MGME1* associate with the degree of coverage discordance between the RNA primer and the DNA flap (Fig. 2c), whereas variants near *TFAM*, *POLG*, *MCAT* and *MGME1* associate with the discordance between 7S DNA and the DNA flap (Fig. 2e). All four genes encode mitochondrial-localized proteins,

and MGME1 and POLG work in concert to resolve flap intermediates (that is, the DNA flap) through exonuclease activity during mtDNA replication[36]. Missense variants in *POLG*, *MGME1* and *MCAT* all show PIP > 0.1 after fine-mapping, and the highest PIP variant at the *MGME1* locus causes p.Thr265Ile, which is in the MGME1 exonuclease domain (Fig. 2f). We also identify a variant causing p.Ala303Gly in *MCAT*, which has no previous connection to mtDNA maintenance and encodes a component of mitochondrial type II fatty acid synthase. RVAS identified further associations between the levels of missense or LoF variation in novel genes and the 7S DNA and DNA flap coverage discordance, including *OMA1* (Supplementary Table 7).

## Phenotypes caused by pathogenic mtDNA mutations

We next considered mtDNA sequence variation in UKB (Methods), with an initial focus on well-established, disease-associated mtDNA variants. We began by assessing the carrier rates for ten common pathogenic mtDNA variants associated with maternally inherited diseases, including Leber's hereditary optic neuropathy; mitochondrial encephalomyopathy, lactic acidosis and stroke-like episodes (MELAS); and aminoglycoside-induced ototoxicity (Fig. 3). We find that approximately 1 in 192 individuals in UKB carries at least one of the

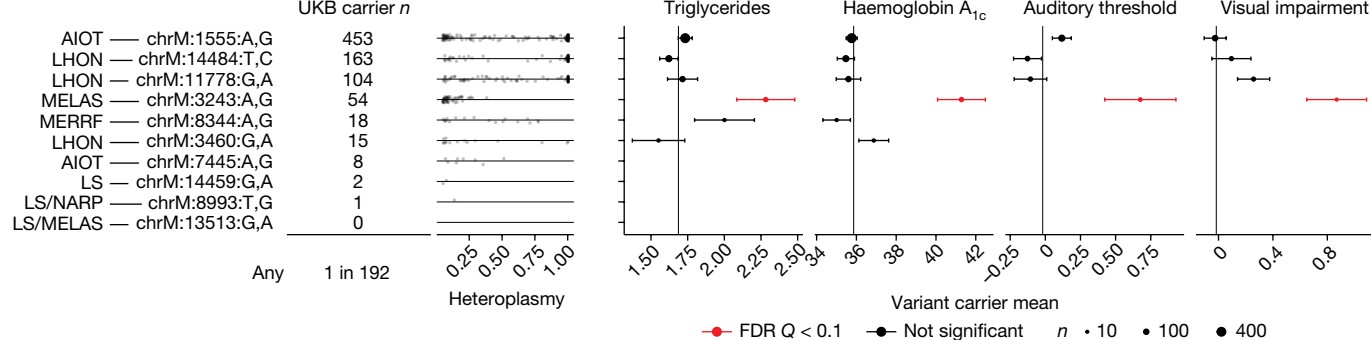

**Fig. 3 | Carrier frequencies and intermediate phenotypes for pathogenic mtDNA mutations assessed in UKB.** Carrier frequencies for ten pathogenic mutations in UKB, with heteroplasmy distributions plotted as jittered points and annotations corresponding to canonically associated disease(s). Panels show mean triglyceride levels, haemoglobin $A_{1c}$, auditory threshold (by means of speech-recognition threshold test) and visual impairment (logMAR, by means of vision test) among mtDNA variant carriers. Point size corresponds to number of carriers with available phenotype data ($n$); only points with more than 10 measurements are shown. Vertical lines represent trait means among individuals not carrying any of the ten variants. Error bars, ±1 s.e.m. AIOT, aminoglycoside-induced ototoxicity; LHON, Leber's hereditary optic neuropathy; MERRF, myoclonic epilepsy with ragged red fibres; LS, Leigh syndrome; NARP, neuropathy, ataxia, retinitis pigmentosa; FDR, false discovery rate.

ten pathogenic mtDNA variants, in agreement with a previous estimate of 1 in 200 (ref. 37).

An open question is whether individuals carrying rare pathogenic mtDNA variants in the population exhibit intermediate disease phenotypes. We can now address this thanks to the rich phenotyping in UKB. We tested four phenotypes traditionally associated with these mtDNA variants: haemoglobin $A_{1c}$ (chrM:3243:A,G), triglyceride levels (chrM:3243:A,G), hearing impairment (chrM:1555:A,G, chrM:3243:A,G, chrM:7445:A,G) and visual impairment (chrM:3460:G,A, chrM:11778:G,A, chrM:14484:T,C, chrM:14459:G,A). Individuals carrying the chrM:3243:A,G variant show elevated haemoglobin $A_{1c}$, elevated triglycerides, and hearing and vision impairment (Fig. 3 and Methods) relative to individuals carrying none of these ten mtDNA variants. Owing to their low frequency of detection in the UKB sample, we do not have the statistical power to exclude the presence of more subtle intermediate phenotypes among the other tested variants.

## mtDNA variation across 253,583 people

Next, we more broadly examined the entire spectrum of homoplasmic and heteroplasmic mtDNA variation. Our analysis across UKB and AoU yields the largest database of mtDNA SNVs and indels to date to our knowledge (Fig. 4a). Consistent with earlier gnomAD analyses[21], we find that the number of homoplasmies per individual is closely related to haplogroup, with haplogroup H (closest to GRCh38 reference) showing the fewest and haplogroup L0 showing the most (Extended Data Fig. 8a). Aggregate heteroplasmy distributions were highly consistent between UKB and AoU (Extended Data Fig. 8d), and most individuals carried 0–1 heteroplasmic SNVs and 0–2 heteroplasmic indels (Extended Data Fig. 8e). The hypervariable regions of the mtDNA, found in the NCR, contain an elevated heteroplasmic SNV rate and most heteroplasmic indel variants (Fig. 4a). Heteroplasmic indels primarily arise near poly-C stretches (for example, chrM:302, chrM:567, chrM:955, chrM:16182) in the non-protein-coding mtDNA, whereas coding mtDNA shows a low indel rate despite the presence of many poly-C tracts (Fig. 4a), consistent with negative selection. We tested the most common heteroplasmies in UKB for association with risk of 29 common diseases (Methods) and found no evidence of association, although sample sizes were limited (Extended Data Fig. 8g).

## Heteroplasmy transmission and age accrual

We next investigated the patterns of transmission and age-dependence for mtDNA heteroplasmies. For analysis of age, we focused on AoU given

the broader age range of participants (20–90 versus 40–70 for UKB). Although heteroplasmic SNVs tend to accumulate with age (particularly after age 70), this was not the case for indel heteroplasmies (Fig. 4b). Using siblings and parent–offspring pairs in UKB (Methods), we found that nearly all heteroplasmic indels were quantitatively maternally transmitted and shared between siblings, whereas most heteroplasmic SNVs were not (Fig. 4c). We also analysed WGS from 602 trios from the 1000 Genomes Project (1000G), finding a similar pattern (Fig. 4d). Unlike UKB blood samples, 1000G samples underwent Epstein-Barr virus transformation to create cell lines before WGS[38,39], implying that the maintenance of these heteroplasmic indels is robust and can be quantitatively maintained through both maternal transmission and cell culture, albeit with some added variance (Fig. 4d). The robust maternal transmission and stability across age leads us to conclude that most indel heteroplasmies are inherited as mixtures; by contrast, for heteroplasmic SNVs, the typical lack of transmission and accumulation with age strongly suggest that they typically arise by means of somatic mutagenesis. In contrast to earlier reports[40], we find no evidence of paternal transmission (Fig. 4c,d). Over 80% of heteroplasmic SNVs were transitions, which showed a sharp increase in frequency in older age, consistent with the somatic mtDNA mutational spectrum seen in ageing brains[41]. Curiously, we observed a decline in heteroplasmic transversions in older individuals (Extended Data Fig. 8f).

## Nuclear GWASs for mtDNA heteroplasmy

We then sought to determine the extent to which mtDNA heteroplasmy is influenced by nuclear genetic loci. To our knowledge, nuclear loci influencing individual mtDNA heteroplasmies have never been identified in humans. Given that most common heteroplasmies showed maternal transmission (Extended Data Fig. 9), we restricted to individuals carrying each heteroplasmy and performed GWAS with the heteroplasmy level as a quantitative trait (Fig. 4e and Extended Data Fig. 8h).

We identified 42 LD-independent associations across 39 heteroplasmies after cross-ancestry meta-analysis of UKB GWASs (Supplementary Note 7). Our results revealed a shared nuclear genetic architecture for heteroplasmies across mtDNA sites, with 9 of 20 unique nuclear loci associated with more than one heteroplasmic variant (Fig. 4f and Extended Data Fig. 10a). Cross-mtDNA heterogeneity was also observed: chrM:302:A,AC and chrM:302:A,ACC appeared most associated with loci near *SSBP1*, *TFAM*, *LONP1* and *MCAT*, whereas the other heteroplasmies were most strongly associated with loci containing *DGUOK*, *PNP* and *POLG2*. Although many genes implicated in heteroplasmy control were also identified in our mtCN GWAS, others were not

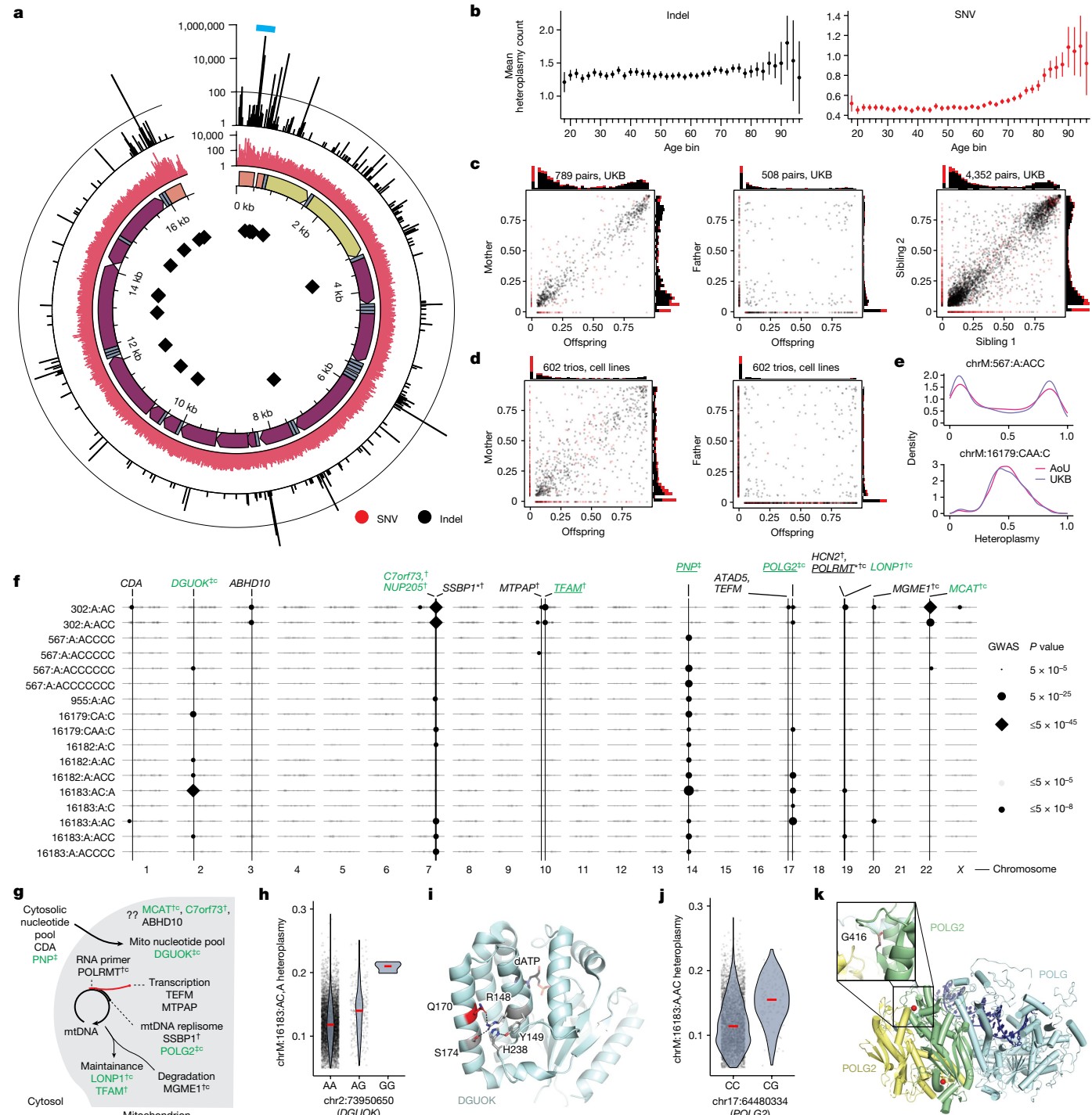

**Fig. 4 | Pervasive nuclear genetic control over common mtDNA heteroplasmies. a**, Quality control (QC)-passing mtDNA heteroplasmies in UKB and AoU. From the inside: mtDNA positions of poly-C tracts; genomic annotations (orange, HVR; yellow, rRNA genes; blue, tRNA genes; purple, coding genes); heteroplasmic SNV counts (red); heteroplasmic indel counts (black). The teal arc region is the focus of Fig. 5. Line in outermost track, 100 indels. **b**, Mean heteroplasmy count per individual across age groups in AoU. Error bars are 1 s.e.m.; total $n = 95,328$. **c**, Heteroplasmy transmission in mother versus offspring (left), father versus offspring (middle) and sibling versus sibling (right) for UKB heteroplasmic variants. **d**, Heteroplasmy transmission in 1000G cell lines in mother versus offspring (left) and father versus offspring (right) pairs. **e**, Selected heteroplasmy distributions among carriers. For panels **a**–**d**, red, SNV; black, indels. **f**, GWAS lead SNPs from common heteroplasmies with any signals at GWS. Point size corresponds to

lead SNP two-sided $P$ value; dark points are at GWS. Vertical lines, SNPs identified for multiple mtDNA variants or near genes of interest. Green, genes also nominated for mtCN; *has Cauchy $P$ value at GWS from RVAS; †CS variants with PIP > 0.1; ‡CS variants with PIP > 0.9; 'c', coding variant in CS; underline, eQTL colocalization with PIP > 0.1. **g**, Role of genes identified by heteroplasmy GWAS in mtDNA dynamics. **h**, chrM:16183:AC,A heteroplasmy versus *DGUOK* lead SNP genotype. **i**, Structure of DGUOK (2OCP from RSCB under CC0 license; https://doi.org/10.2210/pdb2ocp/pdb) with Q170 in red, nearby residues participating in hydrogen bonds or stacking interaction in pink, and dATP as black sticks. **j**, chrM:16183:A,AC heteroplasmy versus *POLG2* lead SNP genotype. **k**, Structure of polymerase gamma (4ZTU from RSCB under CC0 license; https://doi.org/10.2210/pdb4ZTU/pdb) with POLG in light blue and POLG2 subunits in green/yellow. Bound DNA is in dark blue; POLG2 residue G416 is shown as red spheres. In panels **h** and **j**, red lines, median.

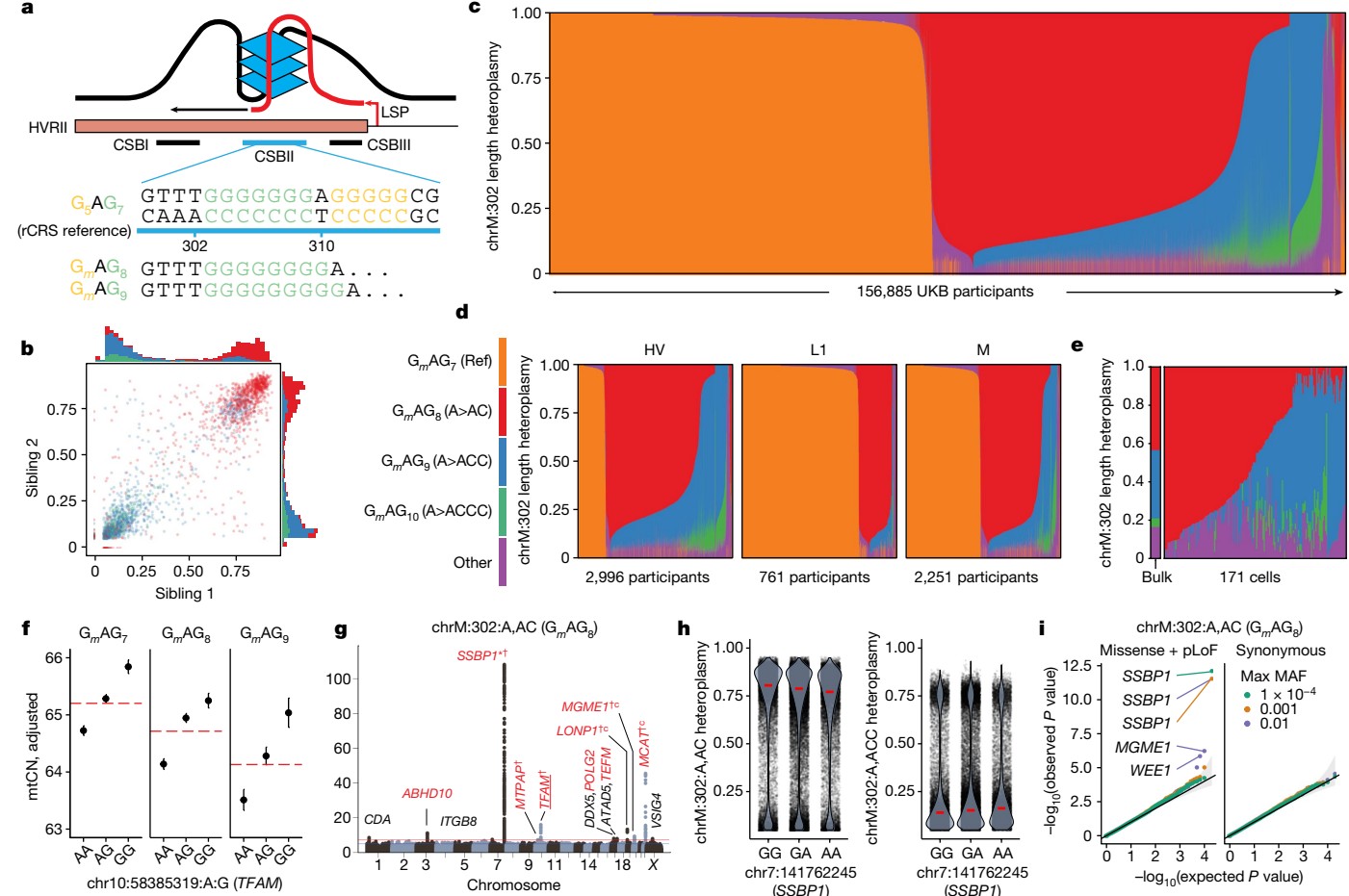

**Fig. 5 | Length heteroplasmies at chrM:302 are inherited maternally as mixtures, co-exist in single cells and are under the influence of variation in the nuclear genome. a**, Scheme of chrM:302 region with associated G-quadruplex and length heteroplasmy ($G_mAG_n$) nomenclature. **b**, Sibling–sibling transmission of chrM:302 length heteroplasmies. **c**–**e**, chrM:302 length heteroplasmy composition across UKB (**c**), within select UKB mtDNA haplogroups (**d**) and across 171 single cells in whole blood (**e**). For **c**–**e**, each vertical bar corresponds to a single individual (**c**,**d**) or cell (**e**). For **b**–**e**, colours correspond to the legend next to panel **d**. **f**, Mean $mtCN_{adj}$ as a function of major chrM:302 allele (red line) and *TFAM* allele

(black dot). Error bars, mean ± 1 s.e.m.; mtCN, copies per diploid nuclear genome; total $n = 121,816$. **g**, Case-only mtDNA heteroplasmy GWAS Manhattan plot for chrM:302:A,AC. Red genes are mitochondria-related; *gene with RVAS Cauchy *P* value at GWS; [†]CS variants proximal to the gene with PIP > 0.1; 'c', missense variant identified in the CS; underline, eQTL colocalization with PIP > 0.1. **h**, chrM:302 heteroplasmy as a function of highest PIP SNP genotype in *SSBP1* locus. Red line, median. **i**, Quantile–quantile plots of gene-based SKAT-O *P* values from RVAS for chrM:302:A,AC. Colours represent max MAF of included variants, black line is null expectation, error band is 95% CI under the null. Ref, reference.

(for example, *TEFM*, *MTPAP*, *SSBP1*, *ABHD10*; Fig. 4f). Many associated loci were near genes with established roles in mtDNA replication and maintenance (Fig. 4g), with missense variants identified in the 95% CS in *DGUOK*, *LONP1*, *POLRMT*, *MGME1* and *POLG2*, and eQTL colocalization PIP > 0.1 seen for *POLRMT*, *POLG2* and *TFAM*. Of the novel hits, we highlight a locus containing *C7orf73* (Fig. 4f and Extended Data Fig. 10f), which encodes a protein recently linked to complex IV (ref. 42), suggesting a moonlighting role for this short protein in mtDNA maintenance.

Zooming in, we see relatively large effect sizes from PIP > 0.9 variants in or near genes related to nucleotide metabolism (*PNP*, *DGUOK*) and DNA replication (*POLG2*). The probable causal variant for *PNP* (PIP 1, Extended Data Fig. 10g) is intronic and colocalizes with a strong negative cross-tissue eQTL[43] (multi-tissue $P \approx 0$; colocalization PIP 1; Extended Data Fig. 10h,i). *PNP* is not yet linked to mtDNA disease but performs an analogous reaction to *TYMP* (an mtDNA disease gene) on purines. The probable causal variant for *DGUOK* (PIP 0.99, Fig. 4h) results in a p.Gln170Arg missense change in the kinase domain, potentially affecting the tertiary structure of the protein as this glutamine side chain participates in a number of hydrogen bonds and stacking interactions (Fig. 4i). The putative causal variant for *POLG2* (PIP 1, Fig. 4j) results in p.Gly416Ala in a predicted anticodon binding domain. This amino acid

is highly conserved (Extended Data Fig. 10j) and the mutation affects a loop near the *POLG2* homodimer surface (Fig. 4k). These examples highlight protein-altering variants that appear to substantially affect the levels of specific heteroplasmic mtDNA variants.

To test whether heteroplasmy-associated nuclear loci act through mtDNA mutagenesis, we repeated our GWAS, re-coding heteroplasmy traits as 'case/control', in which for each mtDNA variant, cases showed detectable heteroplasmy and controls did not. We observed little signal (Extended Data Fig. 10b), arguing against a mutagenic origin influenced by nucDNA variation and supporting the notion that maternal transmission determines the presence of each tested heteroplasmy, whereas nuclear variation can influence the subsequent relative heteroplasmic fraction.

We took several steps to validate our genetic findings. We performed a replication analysis in AoU across 96,698 diverse individuals and observed high concordance between cross-ancestry meta-analysis effect sizes in UKB and AoU ($R^2 = 0.79$; Extended Data Fig. 10c and Supplementary Note 4) with limited attenuation (as expected with winner's curse[44]). We investigated potential technical and biological confounders, observing little correlation between these variables and heteroplasmies (Extended Data Fig. 11a–e and Supplementary Note 2).

We explicitly tested the robustness of our results to the contaminating effects of NUMTs (Supplementary Notes 5 and 6), finding that GWAS effect sizes were not sensitive to mtDNA coverage as would be expected for NUMT-derived signals (Extended Data Fig. 11j–m). We found strong correlations between UKB meta-analysis effect sizes and those from individual ancestry groups in AoU despite small $n$ ($R^2 = 0.49–0.78$; Extended Data Fig. 10d), reducing the likelihood of confounding by recent polymorphic NUMTs. We tested all GWAS hits for LD $R^2 > 0.1$ with variants within 20 kilobase (kb) windows of 4,736 reference and polymorphic NUMTs, finding only 1 potentially concerning locus—among the UKB EUR (European) group, the *SSBP1* locus had LD $R^2 \approx 1$ with variants in a reference NUMT. Importantly, this locus remained significant for chrM:302:A,AC among the AFR (African) group in AoU despite AFR showing much lower LD with NUMT variants (Extended Data Fig. 10k). Further, the levels of ultra-rare missense/LoF variation in *SSBP1* were significantly associated with chrM:302:A,AC heteroplasmy (Fig. 5i and Supplementary Table 7).

## CSBII variation across people and cells

The 'length heteroplasmy' at chrM:302, located in the CSBII region of the mtDNA NCR (Fig. 5a), is the most common heteroplasmic site we observed and occurs within a regulatory motif for mtDNA replication[2]. Although the reference genome corresponds to $G_mAG_7$ (nomenclature indicates the length of the poly-G stretch on the GRCh38 opposite strand, Fig. 5a), we frequently observe individuals harbouring $G_mAG_8$ (chrM:302:A,AC), $G_mAG_9$ (chrM:302:A,ACC) and $G_mAG_{10}$ (chrM:302:A,ACCC). The fractions of mtDNA carrying these variants are quantitatively shared between siblings (Fig. 5b), indicating maternal transmission of mixtures of multiple mtDNA haplotypes at position 302.

Most of the 156,885 individuals assessed in UKB harbour a mixture of these length heteroplasmies (Fig. 5c), with individuals from different haplogroups showing different distributions (Fig. 5d). The observed quantitative maternal transmission of heteroplasmy implies that mtDNA mixtures exist in individual cells, and we indeed find mtDNA mixtures at chrM:302 in 171 single cells from one individual (Fig. 5e) by re-analysing previously reported single-cell data (Methods).

We find multiple lines of evidence linking mtDNA replication and length variation at chrM:302. Longer alleles at this site are associated with declining mtCN$_{adj}$ with an effect size comparable to the *TFAM* locus (Fig. 5f, PIP $\approx 1$). Nuclear genetic analyses for chrM:302:A,AC, the most common length heteroplasmy, nominated several genes relevant for mtDNA replication and nucleotide balance (for example, *SSBP1*, identified by GWAS and corroborated by ultra-rare RVAS; Fig. 5g,i), including several genes not identified in GWASs for other heteroplasmic sites (*CDA*, *MTPAP*, *TFAM*, *TEFM*, *LONP1*, *MCAT*; Figs. 4f and 5g). mtCN and chrM:302:A,AC heteroplasmy even show colocalization at the two most significant mtCN loci: 10:60145079:A,G (a *TFAM* 5′ untranslated region (UTR) variant) and 19:5711930:C,T (a *LONP1* missense variant); both show a PIP $\approx 1$ for mtCN and have PIP $> 0.3$ for chrM:302:A,AC. It is notable that previous studies have suggested that the chrM:302 site serves as a 'rheostat' for mtDNA replication versus transcription, which are functionally linked in mitochondria[3,45]. The G-quadruplex at CSBII (Fig. 5a) is a tertiary RNA/DNA hybrid structure that promotes DNA replication by impairing RNA polymerase progression, promoting the formation of interrupted RNA fragments subsequently used for primed replication[2,46]. Prior in vitro studies have suggested that CSBII G-quadruplex strength is a function of chrM:302 allele, altering the degree to which RNA transcription switches to DNA synthesis[45] (Fig. 5a). We now report that nuclear variants in genes related to the mtDNA replisome can favour one length heteroplasmy over another—for example, variants near *SSBP1* favour chrM:302:A,ACC (Fig. 5h). Taken together, our results propose that nuclear genetic variation can influence the replication efficiency of mtDNA molecules based on chrM:302 allele.

## Discussion

Given that all protein machinery for mtDNA replication and maintenance is nucDNA-encoded, it is plausible that commonly occurring nuclear variants can influence mtDNA heteroplasmy, although this has never been demonstrated in humans. Here, by leveraging WGS across two large biobanks, we report pervasive nuclear genetic control of mtDNA abundance and heteroplasmy variation in humans. Many of these nuclear quantitative trait loci (QTLs) correspond to machinery responsible for mtDNA maintenance, which may influence heteroplasmy by directly acting on mtDNA and altering the relative replication efficiency of mtDNA molecules based on mtDNA sequence, whereas several others correspond to genes never before linked to mtDNA biology. High statistical resolution allows us to gain detailed molecular insights into the mechanisms underlying an entire battery of mito-nuclear interactions, with implications for basic physiology, human disease and evolution.

Our ability to dissect the genetic architecture of mtCN and heteroplasmy was possible both because of the statistical power afforded by the scale of large biobanks and because of careful attention given to technical and biological confounders. We analysed mtDNA sequences across 274,832 individuals of diverse ancestries from two biobanks. We were particularly attentive to contamination by mtDNA pseudogenes in the nuclear genome (NUMTs, Supplementary Notes 5 and 6). We explicitly tested many potential confounders of mtDNA traits, finding that correction of mtCN for blood cell composition had a profound effect on the observed association landscape. Many previously reported associations between blood mtCN and cardiometabolic traits[27,28] disappear or reverse direction after adjustment for blood cell composition (Fig. 1f). Our corrections reduce and even eliminate certain recently reported GWAS hits[32] near genes suspiciously related to blood cell composition and inflammation (for example, *HLA*, *HBS1L*). Our data suggest that, in many cases, an inflammatory state in cardiometabolic disease influences blood cell composition, driving the previously observed decline in mtCN.

The resulting GWASs of mtCN$_{adj}$ and mtDNA heteroplasmies provide molecular insights into mtDNA maintenance. The nuclear loci we identify, including those with fine-mapped missense variation (for example, *MGME1*, *POLG*, *POLG2*, *DGUOK*, *LONP1*), are enriched for roles in the mtDNA nucleoid, mtDNA replication and nucleotide balance. We show how population-level genetic analysis can produce detailed, mechanistic insights into mtDNA replication: GWAS of the relative mtDNA coverage in the 7S DNA 'flap' region highlights missense variants in both *MGME1* and *POLG*, whose products have exonuclease activity that can resolve this replication 'flap' intermediate. We speculate that the putatively causal variant in *MGME1*, p.Thr265Ile, may act by directly affecting DNA binding by disrupting a hydrogen bond within a helix-forming part of the DNA binding pocket of the *MGME1* exonuclease domain (Fig. 2f). We observe notable differences in the genetic architecture of mtCN$_{adj}$ versus heteroplasmy: although *TFAM*, *LONP1*, *DGUOK* and *PNP* are associated with both traits, the former two (encoding components of the mtDNA nucleoid) were the most significant associations for mtCN$_{adj}$, whereas the latter two (involved in nucleotide balance) were among the strongest associations across many heteroplasmies. QTLs corresponding to *TWNK* were identified only for mtCN$_{adj}$, whereas associations near *SSBP1*, *TEFM* and *POLRMT* were specific to heteroplasmy, suggesting that genetic variation in different mtDNA replication genes can have effects specific to mtCN or heteroplasmy. We spotlight many loci with no previous links to mtDNA biology, such as *C7orf73*, *MCAT*, *ABHD10*, *NDUFV3*, *CDA* and *ADA*, implying new roles for their protein products. Future studies are required to evaluate the specific impacts of the candidate causal variants on the function of proteins involved in mtDNA replication and maintenance.

Our study has implications for rare mitochondrial diseases. First, our GWAS nominates candidate genes for unsolved mitochondrial

disease. *PNP* is an excellent example: it has not previously been linked to mtDNA disease; however, we now show that it is associated with $mtCN_{adj}$ and the levels of 13 length heteroplasmic variants at 3 mtDNA sites. It participates in purine catabolism, and defects in analogous steps in pyrimidine catabolism are linked to mtDNA deletion/depletion syndromes. Second, we confirm an earlier estimate that about 1 in 200 individuals carries a known pathogenic mtDNA variant[37], but now also report intermediate phenotypes in such individuals – for example, the MELAS A3243G variant is associated with an increased risk for diabetes. Interestingly, the heteroplasmy distribution observed for the MELAS variant appears to be left-shifted, potentially suggesting negative selection as previously observed[18]. Third, because the number of wild-type mtDNA molecules is key for healthy physiology, it is tempting to speculate that individuals with a higher mtCN polygenic score may be more resilient to pathogenic, heteroplasmic mtDNA mutations, helping to explain some of the striking phenotypic variability observed between family members that carry the same maternally transmitted pathogenic mtDNA mutations[47]. Larger, rare disease-focused studies will be required to determine the extent to which the nuclear variants we have identified can modify the penetrance of mtDNA mutations.

A striking finding from our work is that nearly every human harbours heteroplasmic mtDNA variants obeying two key principles: (1) heteroplasmic SNVs are typically somatic and accrue with age sharply after age 70, whereas (2) heteroplasmic indels are found in more than 60% of individuals, do not accrue with age and are usually inherited as mixtures in the same maternal lineage. The accrual of point mutations with age has been reported[11]; however, to our knowledge the stability of indels with age has not previously been appreciated. Consistent with earlier work[15], heteroplasmic SNVs tend to occur more in the mtDNA hypervariable regions, but we find that most heteroplasmies detected here are actually inherited indels. Most heteroplasmic indels appear to occur next to poly-C stretches in the non-protein-coding mtDNA; heteroplasmic indel rates are orders of magnitude lower next to poly-C stretches in coding regions, suggesting negative selection in these regions. Strikingly, for any given common indel, we find that maternal heteroplasmy levels quantitatively predict offspring heteroplasmy levels, suggesting neutral transmission. We show that these heteroplasmy levels are also under nuclear genetic control, with associated loci enriched for genes involved in mtDNA biology and nucleotide balance. These loci are similar across heteroplasmies at multiple mtDNA sites, suggesting a shared genetic architecture.

In theory, the nuclear QTLs we identify for mtDNA length heteroplasmies could operate by one of two mechanisms: (1) the associated nuclear variants are 'mutagenic' and impair mtDNA copying fidelity resulting in somatic indels due to slippage in poly-C tracts[48], or (2) these nuclear variants confer a replicative advantage to maternally inherited mtDNA molecules carrying certain length variants. Our data favour the latter. Case/control GWAS showed very little signal compared with case-only analysis; in concert with the observed maternal transmission this strongly suggests that the identified nuclear QTLs modify existing indel heteroplasmy levels rather than acting through mutagenesis, potentially by altering the replicative efficiency of the mtDNA molecules carrying different alleles.

Our work provides insight into mechanisms by which the nuclear genotype may be able to confer a replicative advantage to specific mtDNA variants. This is perhaps best illustrated by length heteroplasmy at chrM:302. This heteroplasmy occurs within the G-quadruplex in CSBII in the mtDNA NCR, which may induce switching from transcription to replication by blocking transcription progression. Previous in vitro studies have shown that the chrM:302 length polymorphism affects the strength of this G-quadruplex, hence modifying the transcription/replication switch[3,45]. We find that mixtures of mtDNA with different chrM:302 length variants are found in over half of the population and are maternally inherited. Once inherited, we show that chrM:302 alleles influence mtDNA abundance (acting in *cis*), and we find that the resulting heteroplasmy levels are influenced (in-*trans*) by nuclear QTLs (for example, *SSBP1*, *POLG2*, *TEFM*) whose protein products are thought to directly operate this regulatory switch[45]. In sum, our results indicate that the associated nuclear variants alter chrM:302 heteroplasmy by influencing factors that interact with the chrM:302 G-quadruplex, thus privileging the replication of mtDNA templates carrying a particular chrM:302 genotype. Recent experiments in embryonic stem cells led to speculation that CSBII length variants may contribute to mtDNA reversion after mitochondrial replacement therapy[49] owing to replicative advantage of carryover mtDNA from the intending mother. Our results may provide mechanistic insight into nuclear genetic control of this reversion.

An open question is why mtDNA heteroplasmy is so common in humans, and whether a selective advantage preserves this variation and the observed mito-nuclear interactions. In the current paper, we have shown that quantitative mtDNA traits in the population can be under both *cis*-acting control (through mtDNA variation) and *trans*-acting control (through nucDNA variation), and it is possible that these effects balance each other to maintain stable heteroplasmy across generations. As the mtDNA has high mutation rates with little or no recombination, it is prone to the accumulation of disabling mutations that could lead to its 'meltdown' through Mueller's ratchet[50]. However, mtDNA mutation followed by heteroplasmy is a requisite step in evolutionary adaptation. Nuclear QTLs for mtDNA heteroplasmy may represent mechanisms by which a reservoir of such variation can be tolerated and harnessed over evolutionary time-scales.

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

## Methods

### Overview of mtSwirl

Here we develop mtSwirl, a scalable pipeline for mtCN and variant calling which makes calls relative to an internally generated per-sample consensus sequence before mapping all calls back to GRCh38. In addition to GRCh38 reference files and WGS data, the mtSwirl pipeline takes as input nuclear genome reference intervals that represent regions with high homology to the mtDNA (reference NUMTs). We constructed a set of 385 putative NUMTs by using a BLAST-based inventory of reference NUMTs published previously[51], extending the boundaries of each interval by 500 bases, and merging any overlapping intervals. Initial variant calls in the mtDNA and reference NUMT regions are made from mapped WGS data using Mutect2 and HaplotypeCaller, respectively (using GATK v.4.2.6.0), and haplogroup inference is performed using Haplogrep[52]. Consensus sequences are subsequently constructed using homoplasmies (mtDNA) and homozygous alternative (nucDNA) calls. Reads are realigned to the new consensus sequence and variants are called on the mtDNA using Mutect2. To avoid the artificial coverage depression at the ends of the mtDNA reference genome, we call variants in the control region after alignment to a shifted mtDNA molecule. All variant calls and per-base coverage estimates are then returned to GRCh38 coordinates and output from the pipeline. See Supplementary Note 1 for more details. We release two versions of our pipeline on GitHub (https://github.com/rahulg603/mtSwirl): mtSwirlSingle, a single-sample pipeline intended for use with Cromwell and on platforms with high worker limits such as Terra and the AoU Workbench, and mtSwirlMulti, a multi-sample version that processes multiple samples serially per machine, intended for use on platforms with a smaller parallel worker limit such as the UKB Research Analysis Platform.

### Cohorts

**UKB.** The UKB is a large prospective cohort study of approximately 500,000 individuals in the UK[53], about 200,000 of whom had WGS performed at the time of this study. Samples were selected for the first round of WGS using a pseudorandom approach to ensure that included samples were representative of the full cohort. Sequencing data were generated using DNA extracted from buffy coat obtained from participants; more details have been reported previously[54]. All UKB data were accessed under application 31063 and mtDNA variant calling was performed on the UKB Research Analysis Platform.

**AoU.** AoU is a large longitudinal cohort study based in the USA, with a central goal of enroling a diverse cohort of participants providing electronic health record data over time, specimens for genetic analysis, survey responses and standardized biometric measurements[55]. At the time of this study, 98,590 individuals had completed WGS on samples obtained from whole blood. DNA extraction was completed at the Mayo Clinic, and sequencing was performed at three sequencing centres (Baylor College of Medicine, Broad Institute and University of Washington) using harmonized protocols. Post-sequencing variant and sample QC was performed by the AoU Data and Research Center (DRC). All mtDNA analyses were performed using the AoU Researcher Workbench in the Controlled Tier v6 workspace: 'Genetic determinants of mitochondrial DNA phenotypes', using data from the Q2 2022 release. See https://support.researchallofus.org/hc/en-us/article_attachments/7237425684244/All_Of_Us_Q2_2022_Release_Genomic_Quality_Report.pdf for more details on genomics QC and preprocessing.

**gnomAD v.3.1 subset.** gnomAD v.3.1 is a database aggregating WGS data from 76,156 samples from several experiments and projects around the world, as part of which an mtDNA variant call-set was recently produced[21]. Samples were sourced from several study designs including case–control studies for common diseases, population-based cohorts and observational studies. Individuals with inborn severe paediatric disease were excluded. Most data were sourced from sequencing performed on either blood samples extracted using study-specific methodologies or cell lines[21]. We made use of a subset of the gnomAD v.3.1 samples to prototype our pipeline (mtSwirl) and compare its performance with previous mtCN and variant calls ('Vanilla'). We excluded samples with very high mtCN as done previously[21], as these are most likely to be cell line samples rather than whole blood samples; we used a more stringent threshold of 350 as we wanted to maximally enrich for whole blood samples for this analysis. We also removed samples with mtCN < 50 due to elevated NUMT contamination in these samples[21] (Extended Data Fig. 8c). We selected approximately 6,300 samples from gnomAD v.3.1 to maximize inclusion of diverse haplogroups including those underrepresented in UKB (Extended Data Fig. 2a). We specifically supplemented samples belonging to the L haplogroups and enforced a cap on the number of samples assigned to either NFE (Non-Finnish European) or FIN (Finnish). For other larger haplogroups we performed random subsampling proportional to the original composition of the gnomAD dataset to achieve our final sample size. All analyses were performed using Terra (https://app.terra.bio/), and all analyses were performed using the mtSwirl pipeline deployed using Cromwell in Terra.

**1000G.** The expanded 1000G cohort is a foundational collection of 3,202 diverse samples from 26 populations with recently completed high-coverage WGS and 602 trios[38,39]. Unlike the other cohorts, for which sequencing was performed directly on whole blood or whole blood subfractions, sequencing for 1000G was performed on lymphoblastoid cell cultures which were established from peripheral blood mononuclear cells at the Coriell Cell Repositories[39]. The expanded 1000G cohort, which includes the full set of unrelated samples from 1000G phase 3 as well as additional samples to complete 602 trios, was recently sequenced with more details elsewhere[38]. All data were accessed through the '1000G-high-coverage-2019' workspace in Terra, and all analyses were performed using mtSwirl deployed using Cromwell in Terra.

### Computing mean nucDNA coverage in UKB

As mean nucDNA coverage was not available for UKB, we used samtools v.1.9 idxstats[56], samtools flagstat and GATK v.4.2.6.0 CollectQualityYieldMetrics as part of the mtSwirlMulti pipeline to efficiently and economically estimate mean coverage on the nucDNA. Idxstats-based counts of total mapped reads were computed over autosomes with the subsequent formula applied to get average nucDNA coverage after removing contributions from duplicate reads:

$$\text{Mean coverage} = \frac{(\text{total mapped reads} - \text{singletons} - \text{reads with discordant mate} - \text{duplicates}) \times \text{read length}}{\text{genome length}}$$

### Computing mtCN

Across all cohorts we used the following formula to compute mtCN:

$$2 \times \frac{\text{mean or median mtDNA coverage}}{\text{mean nucDNA coverage}}$$

We defaulted to use of mean mtDNA coverage for the main mtCN-related analyses.

### Post-calling mtDNA phenotype QC

To integrate our variant calls and perform sample and variant QC, we extended a previously developed pipeline[21]. Single-sample variant call format files (VCFs) emitted from mtSwirl were merged into a single

Hail MatrixTable (v.0.2.98 (ref. 57)) upon which all downstream steps were conducted.

For sample QC, any samples showing homoplasmic variant overlap (Supplementary Note 1) were removed. We observed a significant elevation in heteroplasmic SNV calls among samples with mtCN below 50, with a stabilization of heteroplasmic calls above 50 mtDNA copies per cell (Extended Data Fig. 8c), highly suggestive of elevated NUMT contamination in the low copy number samples. Thus, to avoid contamination of our results, all samples with mtCN < 50 were removed. Finally, all samples with evidence of contamination more than 2% were removed, as estimated by (1) mtDNA contamination using Haplocheck 0124 (ref. 58) in mtSwirl, (2) nucDNA contamination or (3) the presence of multiple haplogroup-defining variants at abnormally low allele fraction. Given the small count of samples processed in 2006 and abnormally elevated mtCN estimates in these samples (Extended Data Fig. 3e), we excluded these samples from all UKB analyses.

For variant QC, (1) variants with a very low heteroplasmy (less than 0.01) were called as reference with a heteroplasmy of 0, (2) variants with heteroplasmy below 0.05 were flagged and removed as these are at high risk of being enriched for NUMT-derived signals and (3) all variant calls flagged by Mutect2 were removed (Supplementary Note 1). For all sites, a minimum coverage threshold of 100 was used to distinguish between homoplasmic reference calls and sites without variant calls due to low variant calling confidence as done previously[21]. mtDNA variants were annotated using the Variant Effect Predictor v.101 (ref. 59) and dbSNP v.151 (ref. 60). Variants with at least 0.1% of samples passing filters and showing a heteroplasmy between 0 and 0.5 were annotated as 'common low heteroplasmy'. Variant calls failing QC were coded with a missing heteroplasmy.

For mtCN, we removed the samples identified during variant call-set sample QC as showing signs of contamination or abnormal overlapping homoplasmy calls, or which were processed in 2006. Because we expect mtDNA-wide coverage measures, such as mtCN, to be robust to NUMTs, we do not enforce hard cut-offs on mtCN measurements.

## Construction of mtDNA heteroplasmy phenotypes

We defined our set of common heteroplasmies in UKB as 'common low heteroplasmy' variants (Methods) which are present as heteroplasmies in at least 500 individuals, resulting in 39 variants. We produced two main sets of phenotypes: (1) a 'case-only' dataset consisting of heteroplasmy values for these variants in which any individuals without the variant detected were coded as missing and (2) a 'case–control' dataset in which cases consisted of those with any detectable heteroplasmy and controls consisted of those with the variant not detected. In both phenotype schemes, samples identified as homoplasmic for each variant were always coded as missing. For the case–control dataset, only samples that could be accurately inferred as a reference for each variant were labelled as controls—specifically, the sample was coded as missing for a variant if it had a coverage less than 100 at the site or showed the variant call as QC-fail (Methods).

For sensitivity analyses, we produced several further case-only heteroplasmy datasets: (1) in which any variant calls supported by an alternative allele depth (AD alt) of less than the mean nucDNA coverage of the sample were made missing; (2) in which heteroplasmy estimates were corrected for the depth of mtDNA coverage at the variant site after re-alignment; and (3) in which length heteroplasmy estimates at chrM:302 were corrected for median coverage at CSBII. All corrections were performed by obtaining residuals from the linear regression of the heteroplasmy onto the covariate for each variant across all samples before genetic analysis.

## mtDNA phenotype covariate adjustment approach

We investigated time of day of blood draw, fasting time, assessment date and assessment centre as technical covariates for mtDNA traits. As draw time and assessment date are continuous, we used natural splines in the correction model to flexibly model nonlinear relationships between these covariates and the mtDNA phenotype. For assessment date, we used knots placed roughly seasonally to model seasonal variation in mtDNA phenotypes—these corresponded to 3 month increments starting on 1 July 2007 and ending on 1 July 2010. For draw time, we used a natural spline basis with 5 degrees of freedom. Assessment month and assessment centre were modelled as indicator variables. Fasting times were provided in increments of 1 h and thus were modelled as indicator variables; fasting times of more than 18 h were labelled as 18 and fasting times of 0 were labelled as 1. All terms were included in a joint model for correction.

We also investigated the relationship between mtDNA phenotypes and blood cell type percentages and mean blood cell volumes. We selected all non-redundant traits available: white blood cell leucocyte count, haematocrit percentage, platelet crit, monocyte percentage, neutrophil percentage, eosinophil percentage, basophil percentage, reticulocyte percentage, high light scatter reticulocyte percentage, immature reticulocyte fraction, mean corpuscular volume, mean reticulocyte volume, mean sphered cell volume, mean platelet thrombocyte volume. We did not include nucleated red blood cell percentage as only approximately 1% of the entire UKB cohort has non-zero values for this measure, and we excluded lymphocyte percentage given collinearity with neutrophil percentage ($r = 0.92$) and the sum-to-1 property of the white blood cell differential measurements. To avoid excess leverage from outlying blood cell measurements, we removed any blood measurements with a $Z$-score > 4. All terms were included in a joint model for correction.

For both the technical covariate and blood cell type models, $F$-test $P$ values were obtained for each of the 40 mtDNA phenotypes (39 case-only heteroplasmies and mtCN). For any phenotypes that showed $F$-test $P < 0.05/40$ (Bonferroni corrected), we produced corrected versions of the phenotype by obtaining the residuals from the regression of the mtDNA phenotype onto covariates of interest before genetic analysis. For mtCN, adjustments were performed with log(mtCN) as the response variable. For heteroplasmy estimates, adjustments were performed with case-only heteroplasmies as the response variable. The specific corrections implemented were (where 'ns' refers to the natural spline function):

$$\log(\text{mtCN}) \sim \text{ns(blood draw time, 5)} + \text{assessment centre} \\ + \text{fasting time} + \text{ns(assessment date, SEASONAL KNOTS)} \\ + \text{month of assessment} + \text{blood cell variables}$$

As sensitivity analyses for case-only heteroplasmy phenotypes, residuals from the following models were produced:

$$\text{chrM:567:A,ACCCCCC} \sim \text{ns(blood draw time, 5)} + \text{assessment centre} \\ + \text{fasting time} + \text{ns(assessment date, SEASONAL KNOTS)} \\ + \text{month of assessment}$$

$$(\text{chrM:16093:T,C; chrM:16182:A,ACC; chrM:16183:A,AC}) \sim \text{blood cell} \\ \text{variables}$$

For each response variable, residuals were generated using residuals (lm(model)) as implemented in R v.4.2.1. In all visualizations of covariate-adjusted variables (for example, $\text{mtCN}_{\text{adj}}$), we rescaled the residualized variable by adding the pre-adjustment mean. In the case of $\text{mtCN}_{\text{adj}}$, we rescaled and exponentiated the residualized variable to return adjusted values back to an absolute scale. See Supplementary Notes 2 and 3 for more details.

## mtDNA principal component analysis and predictive power for mtDNA haplogroups

To construct a high-quality variant genotype matrix for principal component analysis (PCA), we obtained the set of homoplasmic variants

(heteroplasmy ≥ 0.95) passing QC identified at a MAF ≥ 0.001 in UKB. Any samples with a QC-pass homoplasmy detected were coded as 1 for each respective variant; all others were coded as 0. This binary genotype matrix was subsequently filtered to the set of unrelated samples upon which we computed the first 50 principal components after centring and scaling using the efficient truncated singular value decomposition algorithm implemented in the irlba v.2.3.5.1 package in R. Related samples were projected onto these principal components (PCs) to produce a set of mtDNA PC coordinates for each sample. The set of related samples were defined previously in the Pan UK Biobank (Pan UKBB) project[61]. In brief, PC-relate was used as implemented in Hail within each assigned genetic ancestry group in UKB and the maximal set of unrelated samples were identified using the maximal independent set algorithm implemented in Hail.

To assess the goodness of fit of mtDNA PCs for the prediction of top-level mtDNA haplogroups, we fit a multinomial model with top-level haplogroup as the response variable and the first 30 mtDNA PCs as explanatory variables as implemented in the nnet v.7.3-17 package in R[62]. We included only samples belonging to haplogroups with at least 30 samples in UKB. For assessment of the predictive power of mtDNA PCs for 'level 2' haplogroups, we fit multinomial models using a similar approach for each top-level haplogroup, with 'level 2' haplogroups as the response variable. In all cases, a null model was fit in parallel with the same response variable with only an intercept term. We computed McFadden's pseudo $R^2$ for each model with the following formula:

$$\text{Pseudo } R^2 = 1 - \frac{\text{log likelihood}}{\text{null model log likelihood}}$$

## Correlations between mtCN, mtCN$_{adj}$, blood cell composition, heteroplasmies and disease phenotypes

We obtained 29 common disease diagnoses from UKB from a previously curated set of phecodes and International Classification of Disease–10 (ICD10) codes corresponding to major common diseases[61] along with demographic variables (age, sex) and blood cell composition phenotypes (Methods). We obtained mtCN$_{raw}$, mtCN$_{adj}$, common ($N > 500$) case-only heteroplasmies (Methods) and three major blood cell composition traits (platelet crit, monocyte count and neutrophil count), and performed $Z$-score transformation for each. To test for associations with disease phenotypes, we implemented a logistic regression model using the glm function in R, including age, sex, age$^2$, age$^2$ × sex, age × sex, top-level haplogroup and genetic ancestry group assignment as covariates:

$$\text{Disease phenotype} \approx \text{trait} + \text{age} + \text{sex} + \text{age}^2 + \text{age}^2 \times \text{sex} + \text{age} \times \text{sex} + \text{population} + \text{top level haplogroup}$$

We included haplogroups with at least 30 individuals represented in UKB. Haplogroup was included in the model only when the trait was mtDNA-derived (for example, it was not included for blood composition phenotypes). Odds ratios were obtained as $\exp(\beta_{trait})$, and the 95% CI was obtained as $\exp(\beta_{trait} \pm 1.96 \times \text{s.e.}_{trait})$.

## Derivation of mtDNA coverage discrepancy phenotypes

We obtained mtDNA intervals corresponding to the 7S DNA, heavy strand origin, CSBII, CSBIII and the LSP[45,63,64]. We computed per-individual median mtDNA coverages in the regions corresponding to the first third of the 7S DNA (termed '7S DNA'), the region between CSBII and the heavy strand origin ('7S DNA flap'), and the region between CSBIII and the LSP ('7S RNA primer'). To generate coverage discrepancy phenotypes, we regressed DNA flap coverage onto either 7S DNA coverage or 7S RNA primer coverage. To avoid coverage discrepancies attributable to inherited mtDNA variation in the regions of interest, we included indicator variables for all top-level haplogroups with at

least 30 samples as well as their interactions with 7S DNA or 7S RNA primer coverage. We also included terms corresponding to the same blood cell composition and technical variables used for adjustment of mtCN (Methods and Supplementary Note 2) to reduce the degree of variation attributable to these factors. The residuals from the following model were used as the coverage discrepancy phenotype for GWAS:

$$\begin{aligned}\text{7S DNA flap coverage} &\approx \text{(7S RNA primer or 7S DNA coverage)} \\ &+ \text{haplogroup} + \text{(7S RNA primer or 7S DNA coverage)} \\ &\times \text{haplogroup} + \text{blood cell composition} + \text{technical variables}\end{aligned}$$

## Relatedness analyses in UKB

Relatedness was computed and sibling–sibling and parent–offspring pairs were inferred as previously described in UKB[65]. For the assessment of transmission of all QC-pass mtDNA variants, we restricted to only variants found in five or more samples.

## Determination of chrM:302 length heteroplasmy composition

To construct length heteroplasmy compositional profiles, we obtained all pre- and post-QC variant calls made at position chrM:302. We generated a 'QC-fail' heteroplasmy estimate at position 302 for each individual by summing pre-QC heteroplasmies that failed post-calling QC; all other alleles included in the composition passed QC (Methods). We defined a 'reference' call at chrM:302 for each sample as 1 − sum(heteroplasmy of any allele at chrM:302), in which the sum included all QC-pass alleles as well as the 'QC-fail' estimate. All samples without variant calls at chrM:302 were assigned a reference fraction of 1, and samples with a depth of less than 100 at chrM:302 (after local re-alignment during variant calling) were excluded. For each sample, we combined all heteroplasmies from calls other than reference, chrM:302:A,AC, chrM:302:A,ACC and chrM:302:A,ACCC into an 'Other' category. The 'QC-fail' fraction was included in the 'Other' category. Any calls with a missing value for a chrM:302 allele (that is, because the allele was removed due to filtering) were imputed as a heteroplasmy of 0 for the purposes of visualizations and analyses. As a final step, any calls with a heteroplasmy fraction less than 0.05 were labelled 'Other' as we use this heteroplasmy cut-off throughout our study to avoid contamination from potential NUMT-derived artifact.

## Associations between pathogenic variant carrier status and continuous phenotypes in UKB

We obtained continuous phenotypes available in UKB corresponding to classic symptoms of MELAS—diabetes-like symptoms (elevated triglycerides (ID 30870), elevated haemoglobin A$_{1c}$ (ID 30750)) and hearing impairment (by means of the speech-reception threshold assessment (IDs 20019 and 20021))—as well as the results from the visual acuity test for analysis of known pathogenic variants for Leber's hereditary optic neuropathy (logMAR from visual acuity test (IDs 5201 and 5208)). All obtained phenotypes were filtered to samples with available mtDNA variant calls and corrections were applied for age, sex, age$^2$, age$^2$ × sex, age × sex and genetic ancestry group assignment by obtaining residuals from the following linear regression model using residuals (lm(model)) in R:

$$\text{Measurement} \approx \text{age} + \text{sex} + \text{age}^2 + \text{age}^2 \times \text{sex} + \text{age} \times \text{sex} + \text{population}$$

As blood biomarkers tend to have log-normal distributions, corrections were applied after log transformation of haemoglobin A$_{1c}$ and triglyceride levels. Post-adjustment, all measurements were returned to their original scale by adding the pre-adjustment dataset-wide means for each measurement modality. Final estimates for the speech-recognition threshold and vision logMAR were generated by averaging measurements for the left and right ear and eye, respectively.

Carriers of known pathogenic mtDNA variants were defined as individuals carrying the variant post-QC at any fraction. We defined a set of controls as individuals with none of the ten known pathogenic mtDNA variants tested. Only samples that could be accurately inferred as reference for all ten variants were labelled as controls—the sample was excluded if, for any of the ten variants, it had a coverage of below 100 at the site or showed a QC-fail variant call (Methods).

Comparisons between residual phenotype values among variant carriers versus global controls were performed only for variant–phenotype pairs with more than ten defined phenotype values among variant carriers. $P$ values were obtained by performing a two-sample $t$-test between phenotype values among variant carriers and the set of global controls, and $Q$ values were obtained by applying the Benjamini–Hochberg procedure.

### Creation of mutational spectrum categories

Heteroplasmic SNV mutation types in AoU were constructed using the set of QC-pass heteroplasmic SNVs. For each SNV type, the set of individuals without any heteroplasmic variants was identified as those with no QC-pass variant call of that type; these individuals were included as zeros in estimates of the mean SNV count of each type.

### chrM:302 length heteroplasmy inference in single cells

Single-cell mitochondrial single-cell assay for transposase-accessible chromatin with sequencing (mtscATAC-seq) data[18] were obtained and analysed with Massachusetts General Hospital Institutional Review Board (IRB) approval under protocol no. 2016P001517. We used the BedTools[66] intersect tool (v.2.29.2) to identify read alignments completely spanning the chrM:300–318 locus in the mtscATAC-seq data. We then iterated over these reads and classified their chrM:302 length variant by extracting the poly-C/G tracts using a regular expression, 'AA(CCC+[CT]CC+)GC', anchored on the two constant base pairs on either side of the variant region to detect the canonical variant structure of two poly-C/G tracts with or without a single intervening A/T. Alleles in matching reads were classified based on the length of their poly-C/G tracts, whereas alleles in the reads that did not match the regular expression were classified as missing. Next, we filtered out any reads with cell barcodes that were not in the published list of cell calls, and further restricted our analysis to only the cells with at least 20 reads at the chrM:300–318 locus. For each of these high-coverage cells, we calculated the fraction of reads showing each of the top three most common length variants ($G_6AG_8$, $G_6AG_9$ and $G_6AG_{10}$) and aggregated any other detected alleles into the remainder (Other) for display as a stacked bar plot. We also estimated bulk heteroplasmy by summing the allele counts from the high-coverage cells and re-calculating the fractions for the top three length variants, again with all other alleles being aggregated into the remainder 'Other' category.

### UKB GWAS approach

All GWASs were performed in UKB using approaches as performed in the Pan UKBB initiative[61]. In brief, ancestry assignment was performed by first projecting UKB samples into genotype PC-space constructed from reference samples from 1000G phase 3 and the Human Genome Diversity Project (HGDP), and subsequently using a random forest classifier to assign continental labels trained on the 1000G + HGDP reference data. In each ancestry group, PCA was performed among unrelated samples with related samples projected onto this PC-space. Further sample QC was performed as described as part of the Pan UKBB initiative[61], including removal of ancestry outliers using a centroid-based metric, and filtering of individuals with high genotype missingness, sex discordance and sex chromosome aneuploidies. Variant QC was also performed on UKB-provided imputed v3 variants (GRCh37) as part of the Pan UKBB initiative[61], including only those with INFO scores greater than 0.8 on autosomes and the X-chromosome. Association tests were performed only on variants with a minor allele count (MAC) > 20. We have constructed and released a mapping from our QC-pass UKB GRCh37 variants to GRCh38 coordinates, built using the bcftools +liftover tool (https://github.com/freeseek/score) with default parameters.

For GWAS, SAIGE v.1.1.5 (ref. 67) was used to perform association tests for each assigned ancestry group using the first ten per-population PCs, age, age × sex, age² and age² × sex as covariates (referred to as 'baseline'). Ancestry groups were included only if at least 50 individuals had the phenotype defined. The use of the SAIGE GRM-based approach allowed for the inclusion of related samples in the GWAS, and we enabled leave-one-chromosome-out fitting in all steps. For all continuous phenotype GWASs (case-only mtDNA heteroplasmy traits and mtCN), phenotypes were inverse rank normalized before genetic analysis.

For all main mtDNA heteroplasmy analyses, top-level mtDNA haplogroup was included as an extra set of covariates in the GWAS model as a set of 24 indicator variables with haplogroup A as reference. Any samples belonging to top-level haplogroups with fewer than 30 samples represented were excluded. The same GWAS model was used for sensitivity analysis of case-only heteroplasmies after removing calls with AD alt less than mean nucDNA coverage, after correction for local variant coverage, after correction for CSBII coverage, and after correction for technical or blood trait covariates (Methods). For the main mtCN analyses, we used only the baseline covariates to perform genetic associations with $mtCN_{raw}$ and $mtCN_{adj}$.

We performed two extra sensitivity analyses for case-only heteroplasmy GWASs: (1) inclusion of 30 mtDNA PCs as covariates in the GWAS model instead of top-level haplogroup for seven variants which showed relatively high heterogeneity across level two haplogroups, and (2) inclusion of $mtCN_{adj}$ as a covariate in the GWAS model for all case-only heteroplasmies in addition to top-level haplogroup. We also tested the effects of including top-level haplogroup indicator variables as extra covariates in GWASs for $mtCN_{raw}$ and $mtCN_{adj}$.

### AoU GWAS approach

We performed a GWAS in AoU as a replication for our main case-only heteroplasmy analyses in UKB. Ancestry inference was performed upstream by the AoU DRC. In brief, AoU samples were projected into the PCA space of genotypes from chromosomes 20 and 21 from HGDP and 1000G, and a random forest classifier trained to identify ancestry labels in 1000G + HGDP was used to assign continental ancestry labels to AoU samples.

We performed sample and variant QC after WGS variant calls (GRCh38) were imported into Hail. Multi-allelic sites were split and sites with very low precomputed allele frequency were removed (MAF > 0.0001 retained). For sample QC, samples flagged by the DRC as population outliers for several metrics or identified as related by the DRC were excluded. For variant QC, we removed any variants filtered by the DRC, which occurred in brief because of no high-quality genotypes for the variant (defined as $GQ \geq 20$, $DP \geq 10$, $AB \geq 0.2$ for heterozygotes), excess heterozygotes or a low-quality score for the variant. We further removed any variants not in Hardy–Weinberg equilibrium (one-sided $P \leq 1 \times 10^{-10}$) and variants with a call rate $\leq 0.95$. Finally, we removed any variants with MAC < 20 in each assigned ancestry group.

We next extracted covariates relevant for our GWAS model. We used an SQL query to obtain date of birth in the controlled data repository and used the provided QC flat files to obtain sex assigned at birth. As date of sample collection was not provided, approximate age was constructed for all analyses by subtracting the year of birth from the year 2021. To address residual stratification in assigned ancestry groups, we produced PCs in each ancestry group using a very similar approach as used in UKB (Methods) as we found that the provided PCs did not appropriately handle stratification among positive control phenotypes such as height, blood glucose, diastolic blood pressure and systolic blood pressure (Supplementary Note 4). We included 20 recomputed PCs, in addition to approximate age, age², age × sex and age² × sex as

covariates in the final GWAS model. We did not perform genetic association analysis for the MID (Middle Eastern) group as fewer than 400 samples with available WGS data were assigned MID.

We used Hail with the hl.linear_regression_rows() method to perform GWAS after all QC. As described in the Methods, we performed genetic analysis for all QC-pass case-only mtDNA heteroplasmies with homoplasmic calls set to missing. As this analysis is intended for replication, we included any mtDNA variants found in 300 or more samples across any ancestry group, resulting in 41 variants for genetic analysis. Of these, 36 were also analysed in UKB; 3 UKB variants were not sufficiently common in AoU for genetic analysis. As in UKB, for the analysis of case-only mtDNA heteroplasmies, top-level mtDNA haplogroup was included as covariates in the GWAS model as a set of 27 indicator variables in addition to age, sex and PC covariates. Samples belonging to top-level haplogroups with fewer than 30 samples in AoU were excluded. All case-only mtDNA heteroplasmy phenotypes were inverse rank normalized before analysis.

See the AoU genotype quality report for more information on upstream genotype data and sample QC, ancestry inference and relatedness inference (https://support.researchallofus.org/hc/en-us/article_attachments/7237425684244/All_Of_Us_Q2_2022_Release_Genomic_Quality_Report.pdf).

### UKB rare variant analysis approach
Gene-based and single-variant testing of rare variants was performed using SAIGE-GENE+ (ref. 68) as implemented in SAIGE v.1.1.5. Given the analysis of low-frequency variants and the small sizes of the other populations, we focused on the EUR (European) genetic ancestry group for this analysis. Covariates and phenotypes were identical to those used for the common variant GWASs in all cases (Methods). Genetic data were obtained from the UKB OQFE 450k exomes release. We enabled leave-one-chromosome-out fitting in all steps, with default parameters used for estimation of categorical variance ratios. SKAT-O[69] was used for set-based testing, with burden and SKAT[70] P values reported for each test. Gene- and variant-consequence annotations were used as constructed elsewhere[68]. For each gene, synonymous, missense, LoF, missense + LoF and synonymous + missense + pLoF variants with maximum MAF $1 \times 10^{-4}$, $1 \times 10^{-3}$ and $1 \times 10^{-2}$ were included in combinatorial sets (12 variant sets per gene) with aggregate P values combined per gene using the Cauchy combination test[71]. Rare variant associations from first assessed using P values from the Cauchy test which combines information across all evaluated categories, with subsequent evaluation of associated variant groups (for example, missense versus synonymous, MAF cut-offs) performed only for results at GWS from the Cauchy test. Thus, for a given phenotype, we defined our GWS threshold based on the primary assessment of the singular Cauchy test (that is, $\frac{0.05}{\approx 18000 \text{ genes}}$).

### Heritability estimation and enrichment analyses for mtCN
S-LDSC[25] was used for heritability estimation and enrichment analyses for mtCN in UKB as performed previously[24]. In brief, we analysed EUR summary statistics in UKB, restricting variants to those in HapMap3 (HM3). We estimated overall SNP-heritability, controlling for 97 annotations corresponding to coding regions, enhancer regions, MAF bins and others[72] (referred to as baselineLD v.2.2). For enrichment analyses, we obtained gene-sets corresponding to (1) the top 10% of genes specifically expressed in major tissues from GTEx[26] and (2) genes producing protein products that localize to each major organelle with high confidence using COMPARTMENTS[73]. Variants were mapped to each gene with a 100 kb symmetric window and LD scores for each gene-set annotation were computed using the 1000G EUR reference panel (https://alkesgroup.broadinstitute.org/LDSCORE/). Heritability enrichment for all gene-sets was tested using S-LDSC atop the baseline v.1.1 model, controlling for 53 annotations including coding regions and 5′ and 3′ UTRs[25].

### Cross-ancestry meta-analysis in UKB and AoU
We conducted a fixed-effect meta-analysis across ancestries in each cohort (UKB and AoU) based on inverse-variance weighted betas and standard errors[74]. For each ancestry, we excluded low-confidence variants defined as MAC ≤ 20 in either biobank. We computed effect size heterogeneity P values across ancestries using Cochran's Q-test[75]. All computation was done using Hail v.0.2.

All visualizations of main GWASs (for example, mtCN, coverage discrepancy traits, heteroplasmy traits) are of cross-ancestry meta-analyses after restriction to the set of 'high-quality' variants as defined previously[61].

### Identification of LD-independent lead SNPs and locus definitions
Clumping was performed using Plink v.1.90 (ref. 76) in Hail Batch for GWAS results obtained in UKB after filtering to high-quality variants. We used significance thresholds of 1 for both the index and clumped SNPs, set the LD threshold for clumping at 0.1 and set the distance threshold at 500 kb. We used single-ancestry and multi-ancestry LD reference panels corresponding to the ancestry groups included in the final multi-ancestry meta-analyses for each mtDNA phenotype as well as for blood cell traits. Reference panels were constructed by randomly sampling 5,000 individuals from all samples in any given set of ancestry groups in the UKB. For single-ancestry LD panels corresponding to ancestry groups with fewer than 5,000 individuals assigned (EAS (East Asian) and MID), the full sample available for each ancestry group was used. More details on the LD reference panels can be found as part of the Pan UKBB project[61]. Clumping output files from Plink were converted to Hail Tables and then combined into MatrixTables using the multi-way-zip-join method as implemented in Hail.

We defined distinct loci conservatively by starting with LD-independent lead SNPs at GWS and merging any SNPs within 2 megabases (Mb) of one another.

### Replication of previous mtCN GWAS with our study
We performed a comparison of significant loci identified in a previous GWAS of mtCN in UKB[32] with our own by performing LD clumping on previously released summary statistics as described (Methods) using 1000G phase 3 EUR reference data for LD. We defined distinct loci as described (Methods), merging any SNPs within 2 Mb of one another, arriving at 96 loci previously identified. We defined a replicated locus with mtCN_raw or mtCN_adj as one in which our GWAS showed a signal at $P < 5 \times 10^{-5}$ or $5 \times 10^{-8}$ within 2 Mb of the most significant variant identified in the previous study at each locus.

### Bidirectional Mendelian randomization between UKB mtCN and selected traits
GWAS effect sizes and LD-independent loci from the UKB cross-ancestry meta-analysis for mtCN_raw and mtCN_adj were obtained. Summary statistics and LD-independent loci from GWAS among EUR for neutrophil count (ID 30140) and case/control disease traits that showed correlation with mtCN_adj: osteoarthritis (categorical_20002_both_sexes_1465), angina (categorical_20002_both_sexes_1473), myocardial infarction (phecode_411.2_both_sexes), ischaemic heart disease (phecode_411_both_sexes) and high cholesterol (categorical_20002_both_sexes_1473), were obtained from the Pan UKBB project[61]. Loci for effect-size comparison were restricted to those passing variant QC as performed in UKB (Methods). For each mtCN phenotype, neutrophil count and disease trait, GWAS effect sizes were obtained for all variants at GWS in the mtCN GWAS, and, vice versa, mtCN, neutrophil count and disease trait GWAS effect sizes were obtained for all neutrophil count and disease trait variants at GWS. We assessed the relationship between pre- and post-adjustment mtCN GWAS effect sizes and neutrophil count/disease trait GWAS effect sizes using inverse-variance weighted linear regression using weights corresponding to

$\frac{1}{\text{s.e.(mtCN)}^2} \times \frac{1}{\text{s.e.(trait of interest)}^2}$, in which effect size standard errors were obtained from the respective GWAS.

## Fine-mapping in UKB

To identify putative causal variants in associated loci, we conducted statistical fine-mapping of mtDNA traits in UKB using cross-ancestry meta-analysis summary statistics. Although we previously showed that fine-mapping a meta-analysis is often miscalibrated due to heterogeneous characteristics of constituent cohorts (for example, genotyping or imputation)[77], a within-cohort cross-ancestry meta-analysis such as the present study is a notable exception given no such heterogeneity systematically exists across ancestries.

We used FINEMAP-inf and SuSiE-inf, which model infinitesimal effects[78], with cross-ancestry meta-analysis summary statistics (Methods) and a covariate-adjusted in-sample dosage LD matrix[79]. We defined fine-mapping regions based on a 3 Mb window around each lead variant and merged regions if they overlapped as described previously[79]. We excluded the major histocompatibility complex (MHC) region (chr 6: 25–36 Mb) from analysis due to extensive LD structure in the region. For each method, we allowed up to ten causal variants per region and derived PIPs of each variant using a uniform prior probability of causality. To achieve better calibration, we computed min(PIP) across the methods and derived up to 10 independent 95% CSs from SuSiE-inf as described elsewhere[79]. All reported PIPs are min(PIP) values between the two methods.

## Enrichment of functional categories among fine-mapped variants

We computed functional enrichment of fine-mapped variants across the mtDNA traits in UKB. We first annotated each variant with seven functional categories (pLoF, missense, synonymous, 5′ UTR, 3′ UTR, promoter, cis-regulatory element (CRE) and non-genic) as described previously[79]. We then estimated functional enrichment for each category as a relative risk (that is, a ratio of proportion of variants) between being in an annotation and fine-mapped (PIP ≤ 0.01 or PIP > 0.1). That is, a relative risk = (proportion of variants with PIP > 0.1 that are in the annotation)/(proportion of variants with PIP ≤ 0.01 that are in the annotation). The 95% CIs were calculated using bootstrapping with 5,000 replicates. We note that, to increase statistical power, we combined pLoF/missense and 5′/3′ UTR into single categories, respectively, and used a more lenient threshold (PIP > 0.1 versus >0.9) compared with our previous analysis[79].

## Gene- and variant-prioritization

To nominate genes using GWAS results for each phenotype, we used the following approach to balance clarity with confidence in the gene assignment.

1. If the locus had a CS, for each CS:
   a. Filter to variants in the CS and retain variants from the CS that are either minimal PIP or coding, or have PIP > 0.7.
   b. If the variant has PIP > 0.9 and is a coding variant for a gene, assign that gene to the CS.
   c. Otherwise, assign genes within 3 kb of the variant or, if no genes are within 3 kb, assign the nearest gene to the CS.
2. If the locus had multiple CSs and at least one had a variant with PIP > 0.1, we retained assignments corresponding only to variants with PIP > 0.1.
3. If the locus did not have a CS, we assigned the gene with a boundary nearest to the most significant variant in the locus.
4. We also used RVAS to nominate additional, or support existing, gene assignments for all GWAS loci containing genes with SKAT-O Cauchy RVAS P values at GWS for the same phenotype.

If a variant is inside a gene body (but is non-coding), we considered that gene to be nearest. For case-only heteroplasmy GWASs, when the same locus was significant across multiple heteroplasmy phenotypes, we performed manual integration to arrive at a set of genes supported by the most compelling genetic evidence across variants for each locus. The SSBP1 locus was particularly complex, so we assigned SSBP1 (which harbours the max PIP variant) and provided visualization of the full locus (Extended Data Fig. 10k). We did not use fine-mapping evidence from variants with PIP > 0.1 that are not assigned to a CS. All assignments were manually reviewed. In all GWAS visualizations, we labelled the strength of evidence supporting the gene assignment (for example, if supported by moderate- or high-PIP fine-mapped variants, coding variants, RVAS gene-based test association).

## Colocalization with eQTLs

We conducted colocalization of fine-mapped variants of mtDNA phenotypes and cis-eQTL associations from GTEx v.8 (ref. 43) and eQTL catalogue release 4 (ref. 80) as described previously[79]. Briefly, we retrieved fine-mapping results of cis-eQTL associations that were fine-mapped using SuSiE[81] with covariate-adjusted in-sample dosage LD-matrices[79]. We then computed a PIP of colocalization for a variant as a product of PIP for GWAS and for cis-eQTL (CLPP = $PIP_{GWAS} \times PIP_{cis\text{-}eQTL}$)[82]. When displaying colocalization across heteroplasmy traits, we indicate colocalization if we see a colocalization PIP > 0.1 for the assigned gene and any variant in the CS for any tissue and for any heteroplasmy trait.

## Replication of UKB heteroplasmy results in AoU

To perform replication analysis in AoU, we used LD-independent lead SNPs from all case-only heteroplasmy GWASs originally performed in UKB (Methods). We filtered association statistics from AoU (Methods) to these lead variants and compared effect sizes when the variants were identified in AoU with MAC > 20. We used Deming regression implemented in the deming v.1.4 package in R to assess the relationship between effect sizes for these lead SNPs in cross-ancestry meta-analyses in the two biobanks while accounting for standard errors in both[83,84]. We coded alleles such that effect sizes were always positive in UKB.

## Assessment of LD with known polymorphic and reference NUMTs

We collated an extensive database of polymorphic and reference NUMT intervals using BLAST, known reference NUMTs[51,85] and published polymorphic NUMTs inferred using mate-pair mapping discordance[86,87]. To search for regions of homology to the mtDNA within the reference nucDNA, we used BLASTn 2.13.0 with the GRCh37 reference genome with a word size of 11, an expected threshold of 0.05, short queries enabled and default values for the other parameters. In total, we obtained 4,736 overlapping reference and polymorphic NUMT intervals. We constructed a 20 kb window around each nucDNA NUMT region (10 kb up, 10 kb down) and then conservatively tested for LD $R^2 > 0.1$ between any SNP in the window and each lead variant at GWS for our UKB case-only heteroplasmy GWAS using in-sample genome-wide EUR LD-matrices generated in UKB[61]. All LD values used to examine individual loci in either biobank were computed in-sample—for example, in AoU we computed LD using the post-QC genotype MatrixTable (Methods) used for GWAS with the Hail function hl.ld_matrix().

## Multiple alignment of POLG2 protein sequence

POLG2 homologues were detected using the best bidirectional BlastP hit (expected < $1 \times 10^{-3}$) from humans and were aligned using MUSCLE[88].

## Reporting summary

Further information on research design is available in the Nature Portfolio Reporting Summary linked to this article.

## Data availability

In terms of data processed or generated as part of this study, we provide per-population mtDNA heteroplasmic and homoplasmic allele

frequencies and counts in UKB and AoU (Supplementary Tables 5 and 6), genetic association statistics for LD-independent lead SNPs and fine-mapped variants in UKB in addition to colocalization results (Supplementary Tables 2–4) and gene-based RVAS association statistics for genes at GWS for the Cauchy test (Supplementary Table 7). All GWAS sample sizes for each genetic ancestry group, meta-analysis and phenotype can be found in Supplementary Table 1. All GWAS summary statistics from UKB cross-ancestry meta-analyses (used here in discovery analyses) have been deposited in the GWAS Catalog (ID: GCP000614). Summary statistics containing all per-ancestry association statistics as well as cross-ancestry meta-analyses can be accessed through the Google Cloud Platform (bucket: gs://mito-wgs-public-2023). Full GWAS summary statistics from AoU (used here as a replication cohort) have been deposited in a workspace available on the AoU workbench (titled 'Nuclear genetic control of mtDNA copy number and heteroplasmy in humans'; https://workbench.researchallofus. org/workspaces/aou-rw-3273c7f0/nucleargeneticcontrolofmtd- nacopynumberandheteroplasmyinhumans/data). Individual-level data generated as part of UKB (mtCN and mtDNA variant calls) have been returned to UKB to enable utilization of the full individual-level data by the broader scientific community through the UKB data showcase. Individual-level data generated as part of AoU have been deposited in the same workspace containing summary statistics on the AoU Research Workbench. Please see our GitHub repository (https://github.com/rahulg603/mtSwirl) for more information on accessing these data. At the time of publication, access to the AoU workbench controlled tier is restricted to US-based academic institutions, government entities, health care institutions and non-profit organizations. Please also note that as of the time of publication, the only method to gain access to the AoU workspace containing the data generated here is to contact us to be added to the workspace. For information about access to the Researcher Workbench as a registered researcher, please visit https://www.researchallofus.org. In terms of external data used in this study, we leveraged GWAS summary statistics, and ancestry-specific LD-matrices, and a curated list of 29 common, high-quality disease phenotypes generated as part of the Pan UKBB project[61]. Paths for these summary statistics (https://pan. ukbb.broadinstitute.org/docs/per-phenotype-files) and LD-matrices (https://pan.ukbb.broadinstitute.org/docs/ld) can be found on the Pan UKBB project website (https://pan.ukbb.broadinstitute.org); these were accessed through the Google Cloud Platform as part of this study. UKB phenotype and whole-genome sequencing data can be accessed through the UKB Research Analysis Platform after completing a UKB access application (https://ukbiobank.dnanexus. com/landing). AoU phenotype and genotype data can be accessed through the Controlled Tier v6 on the AoU researcher workbench (https://workbench.researchallofus.org). gnomAD v.3.1.2 (https:// gnomad.broadinstitute.org) WGS was accessed through a custom Terra workspace (titled 'gnomad_subsampled_mitopipeline_head_ to_head'). High-coverage WGS data from 1000G were accessed using the public '1000G-high-coverage-2019' workspace in Terra. Published mtscATAC-seq data used for chrM:302 analysis can be obtained with dbGaP approval. Gene-sets for enrichment analyses can be obtained using COMPARTMENTS (https://compartments. jensenlab.org) and MitoCarta 2.0 (https://www.broadinstitute.org/ files/shared/metabolism/mitocarta/human.mitocarta2.0.html) as described previously[24]. The GRCh37 and GRCh38 reference genomes as well as other standard reference data are available through the GATK resource bundle (https://gatk.broadinstitute.org/hc/en-us/articles/ 360035890811-Resource-bundle). Annotations for the baseline v.1.1 and BaselineLD v.2.2 models for S-LDSC as well certain other relevant reference data, including the HapMap3 SNP list, can be obtained from https://alkesgroup.broadinstitute.org/LDSCORE/. Known reference and polymorphic NUMTs were obtained from supplemental data as provided in published work[51,85–87].

## Code availability

We release the full WDL pipelines and associated input files for mtDNA analysis from whole-genome sequencing data on GitHub (https://github.com/rahulg603/mtSwirl; https://doi.org/10.5281/ zenodo.8067503). We also provide the code we used to run the pipeline on the UKB Research Analysis Platform, AoU and Terra; consolidate all data; perform mtDNA sample and variant QC; and run GWAS. See the Methods and the README in the GitHub repository for more information on how to use the pipeline. Several tools were used as part of mtSwirl, including GATK v.4.2.6.0 (https://gatk.broadinstitute.org/), samtools v.1.9 (https://github.com/samtools/samtools) and bcftools v.1.16 (https://github.com/samtools/bcftools), Haplochecker 0124 https://github.com/genepi/haplocheck), R v.3.1.1 (https://r-project.org), Hail v.0.2.84 (https://hail.is) and UCSC kent tools source v.430 (genome-source.soe.ucsc.edu/kent.git and https://hgdownload.soe. ucsc.edu/admin/exe/linux.x86_64/). We used several published tools and scripts to perform downstream analysis of the mtDNA call-set in this study. All data wrangling, statistical analysis and figure generation was performed using Hail v.0.2.98 (https://hail.is), python v.3.7.10 (https://www.python.org) or R v.4.2.1 (https://r-project.org). Parallelization of tasks in UKB was performed using Hail Batch (in Hail v.0.2.98) (https://batch.hail.is) and in AoU using Cromwell v.77 (https://crom- well.readthedocs.io). GWAS was performed in UKB using SAIGE v.1.1.5 (https://saigegit.github.io). For scaling of UKB GWAS, a custom modification of the GWAS pipeline from the Pan UKBB pan-ancestry GWAS was implemented (https://github.com/atgu/ukbb_pan_ancestry). Linear regression GWAS was performed in AoU using Hail. We release the code used for GWAS on both UKB and AoU on GitHub (https://github. com/rahulg603/mtSwirl). mtDNA PCA was performed in R using the irlba v.2.3.5.1 package (https://cran.r-project.org/web/packages/irlba/ index.html). Multinomial models were trained using the nnet v.7.3-17 package in R (https://cran.r-project.org/web/packages/nnet/index. html). Circos plots were made using the circlize package v.0.4.15 in R (https://jokergoo.github.io/circlize_book/book/). For analysis of chrM:302 in single-cell data, we used BedTools v.2.29.2 (https://bed- tools.readthedocs.io). LD clumping was performed using Plink v.1.90 (https://www.cog-genomics.org/plink/). Fine-mapping was performed using FINEMAP-inf v.1.3 and SuSiE-inf v.1.2 (https://github.com/Finu- caneLab/fine-mapping-inf). eQTL data were obtained from GTEx v.8 (https://gtexportal.org) and the eQTL catalogue release 4 (https://www. ebi.ac.uk/eqtl/). For replication analysis effect size comparisons, the deming v.1.4 package was used in R (https://cran.r-project.org/web/ packages/deming/index.html). Heritability estimates and enrichment analyses were performed using stratified LD-score regression (https:// github.com/bulik/ldsc). BLASTn v.2.13.0 was used as available from the NCBI (https://blast.ncbi.nlm.nih.gov/Blast.cgi). MUSCLE v.3.8.31 was used for protein sequence alignment (https://drive5.com/muscle/ downloads_v3.htm).

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

**Acknowledgements** We thank M. Falkenberg, C. Gustafsson, T. Barton-Owen and C. Winter for discussions; M. Walker for facilitating access to published data; T.-L. To for technical analyses; and K. Laricchia, K. Chao, G. Tiao, S. Chapman and N. Gupta for help with gnomAD data and pipelines. This project was supported in part by grants no. 5R35GM122455 (V.K.M.), no. 5R37MH107649 (B.M.N.), no. 5R01MH101244 (B.M.N.), no. 5F30AG074507 (R.G.) and no. K99HG012222 (W.Z.) from the National Institutes of Health. This work is supported by the Novo Nordisk Foundation, no. NNF21SA0072102 (B.M.N.). A.V.K. is supported by The Jane Coffin Childs Memorial Fund for Medical Research fellowship. P.F.C. is a Wellcome Trust Principal Research Fellow (grant no. 212219/Z/18/Z), and a UK NIHR Senior Investigator, who receives support from the Medical Research Council (MRC) Mitochondrial Biology Unit (grant no. MC_UU_00028/7), the MRC International Centre for Genomic Medicine in Neuromuscular Disease (grant no. MR/S005021/1), the Leverhulme Trust (grant no. RPG-2018-408), an MRC research grant (no. MR/S035699/1) and an Alzheimer's Society Project Grant (no. AS-PG-18b-022). V.K.M. is an Investigator of the Howard Hughes Medical Institute. This research was supported by the NIHR Cambridge Biomedical Research Centre (grant no. BRC-1215-20014). The views expressed are those of the authors and not necessarily those of the NIHR or the Department of Health and Social Care. This research has been conducted using the UK Biobank Resource under Application Number 31063. The AllofUs Research Program is supported by the National Institutes of Health, Office of the Director: Regional Medical Centers: 1 OT2 OD026549; 1 OT2 OD026554; 1 OT2 OD026557; 1 OT2 OD026556; 1 OT2 OD026550; 1 OT2 OD026552; 1 OT2 OD026553; 1 OT2 OD026548; 1 OT2 OD026551; 1 OT2 OD026555; IAA no.: AOD 16037; Federally Qualified Health Centers: HHSN 263201600085U; Data and Research Center: 5 U2C OD023196; Biobank: 1 U24 OD023121; The Participant Center: U24 OD023176; Participant Technology Systems Center: 1 U24 OD023163; Communications and Engagement: 3 OT2 OD023205; 3 OT2 OD023206; and Community Partners: 1 OT2 OD025277; 3 OT2 OD025315; 1 OT2 OD025337; 1 OT2 OD025276. In addition, the AllofUs Research Program would not be possible without the partnership of its participants.

**Author contributions** R.G., B.M.N. and V.K.M. conceived of the project. R.G. developed mtSwirl, produced mtDNA call-set and performed bioinformatic and statistical analyses with contributions from M.K. for fine-mapping and colocalization, T.J.D. for analysis of mtscATAC-seq data, K.T. for LD clumping and J.G.M. for structural analyses. K.J.K. and S.E.C. oversaw mtSwirl implementation. R.G., M.K., A.V.K., W.Z., P.F.C., K.J.K., S.E.C., B.M.N. and V.K.M. designed analyses. B.M.N. and V.K.M. supervised the study. R.G. and V.K.M. wrote the manuscript with input and feedback from all other authors.

**Competing interests** V.K.M. is a paid advisor to 5am Ventures. B.M.N. is a member of the scientific advisory board at Deep Genomics and Neumora and a consultant of the scientific advisory board for Camp4 Therapeutics. K.J.K. is a consultant for Vor Biopharma. The remaining authors declare no competing interests.

**Additional information**
**Correspondence and requests for materials** should be addressed to Rahul Gupta, Benjamin M. Neale or Vamsi K. Mootha.

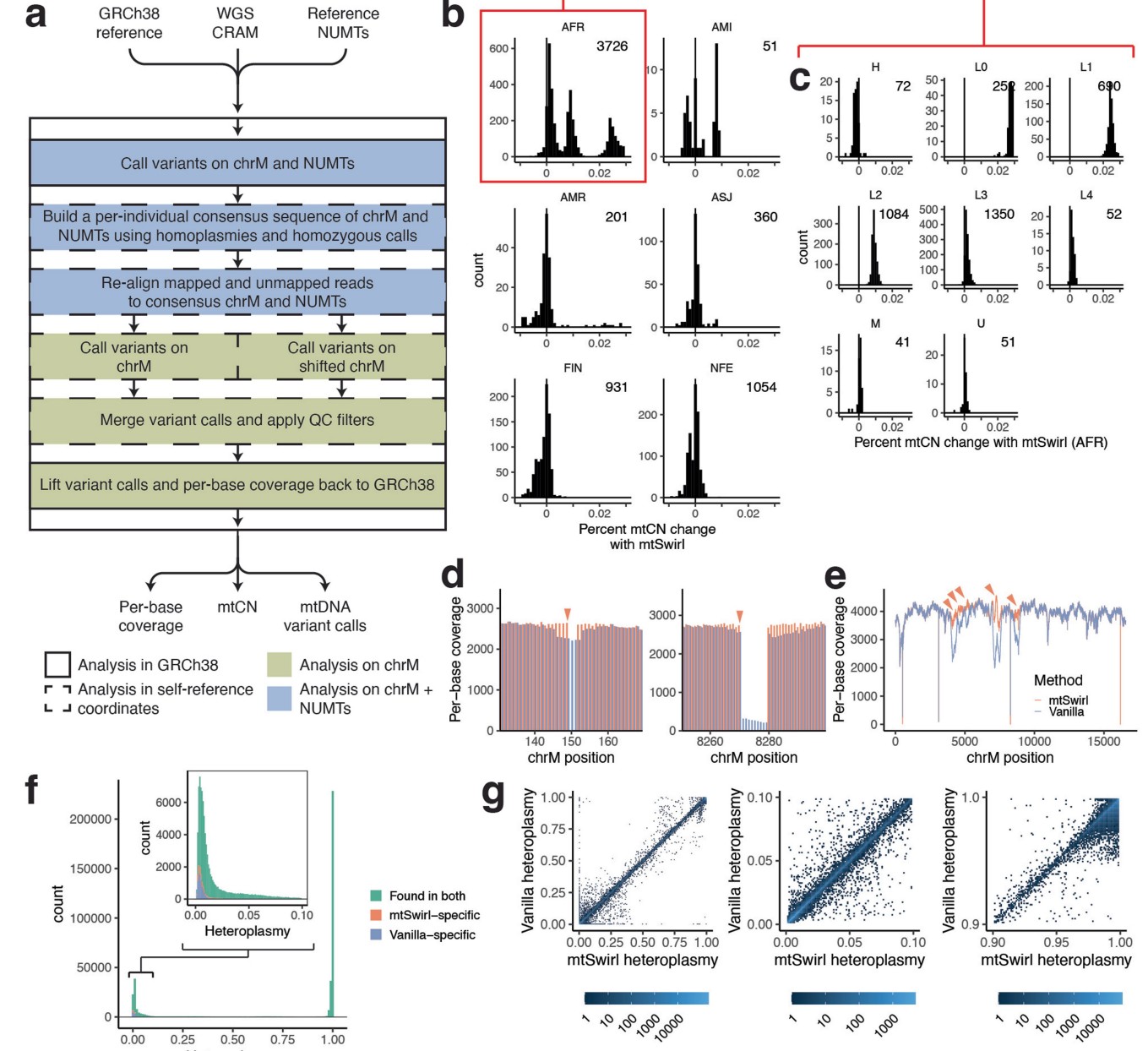

**Extended Data Fig. 1 | Copy number and heteroplasmy estimation improvements using mtSwirl pipeline. a**. Overview of mtSwirl pipeline workflow. Colors represent genomic region analyzed (blue = chrM and NUMTs; yellow = chrM only); border style represents coordinate system (solid = GRCh38; dashed = self-reference coordinates). All output is in GRCh38. **b**. Percent change in mtCN estimated using "vanilla" pipeline versus mtSwirl as a function of inferred nuclear ancestry. **c**. Percent change in mtCN among AFR individuals as a function of mtDNA haplogroup. **d**. Example of per-base coverage improvement with mtSwirl near a homoplasmic indel, likely due to use of mtDNA self-reference

sequence. Arrows highlight homoplasmic indels. **e**. Example per-base coverage improvement likely due to reduced mis-mapping to nucDNA. Arrows highlight coverage improvements. **f**. Variant calls found using both pipelines (green), only in mtSwirl (red), and only in "vanilla" (blue). Inset corresponds to zoomed view of low heteroplasmy variants. **g**. 2D histogram showing relationship between heteroplasmy estimates using mtSwirl with "vanilla". Left panel corresponds to overall heteroplasmy space; middle is zoomed to low heteroplasmy variants; right is zoomed to high heteroplasmy variants.

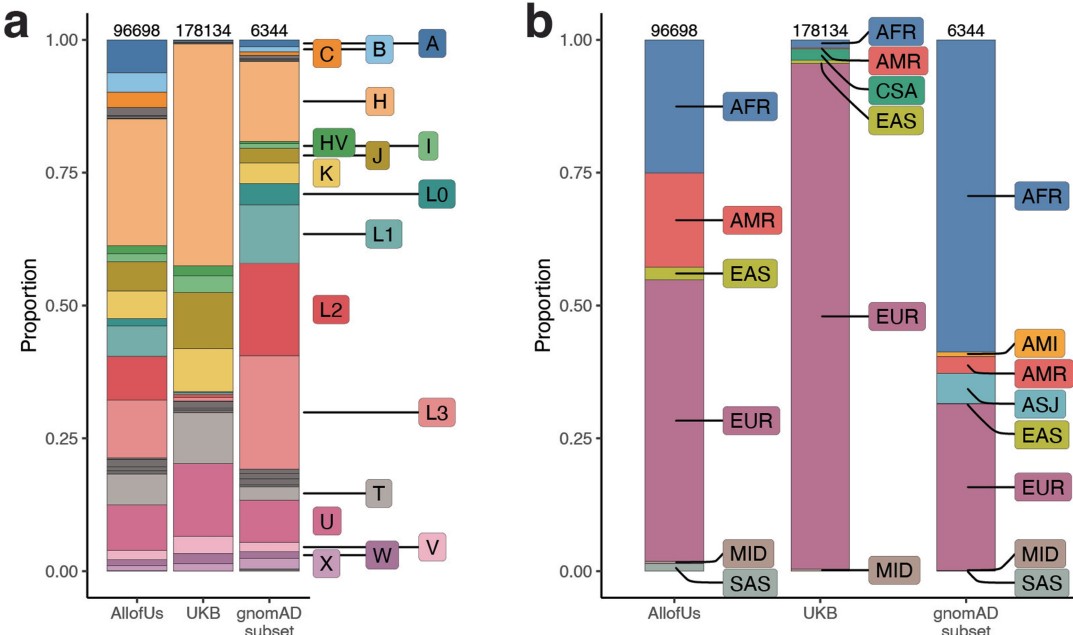

African: L0, L1, L2, L3, L4, L5, L6
Asian: A, B, C, D, E, F, G, M, N, O, P, Q, S, Y, Z
European: H, HV, I, J, K, R, T, U, V, W, X

**Extended Data Fig. 2 | Composition of cohorts used in this study. a**. Top-level haplogroups represented in each analyzed cohort. Labeled haplogroups comprise >1.5% of the samples in at least one cohort. Haplogroups are mapped to broad ancestral categories as indicated by the text below the plot with colors corresponding to the colors of the labeled haplogroups. **b**. Inferred nuclear genetic ancestry groups in each analyzed cohort. Ancestry group assignment was completed within each cohort. The NFE and FIN groups in gnomAD were combined under "EUR" for the purposes of this comparison. In both panels, numbers at the top of each bar indicate the number of samples with generated mtDNA callsets passing QC with completed genetic ancestry assignment.

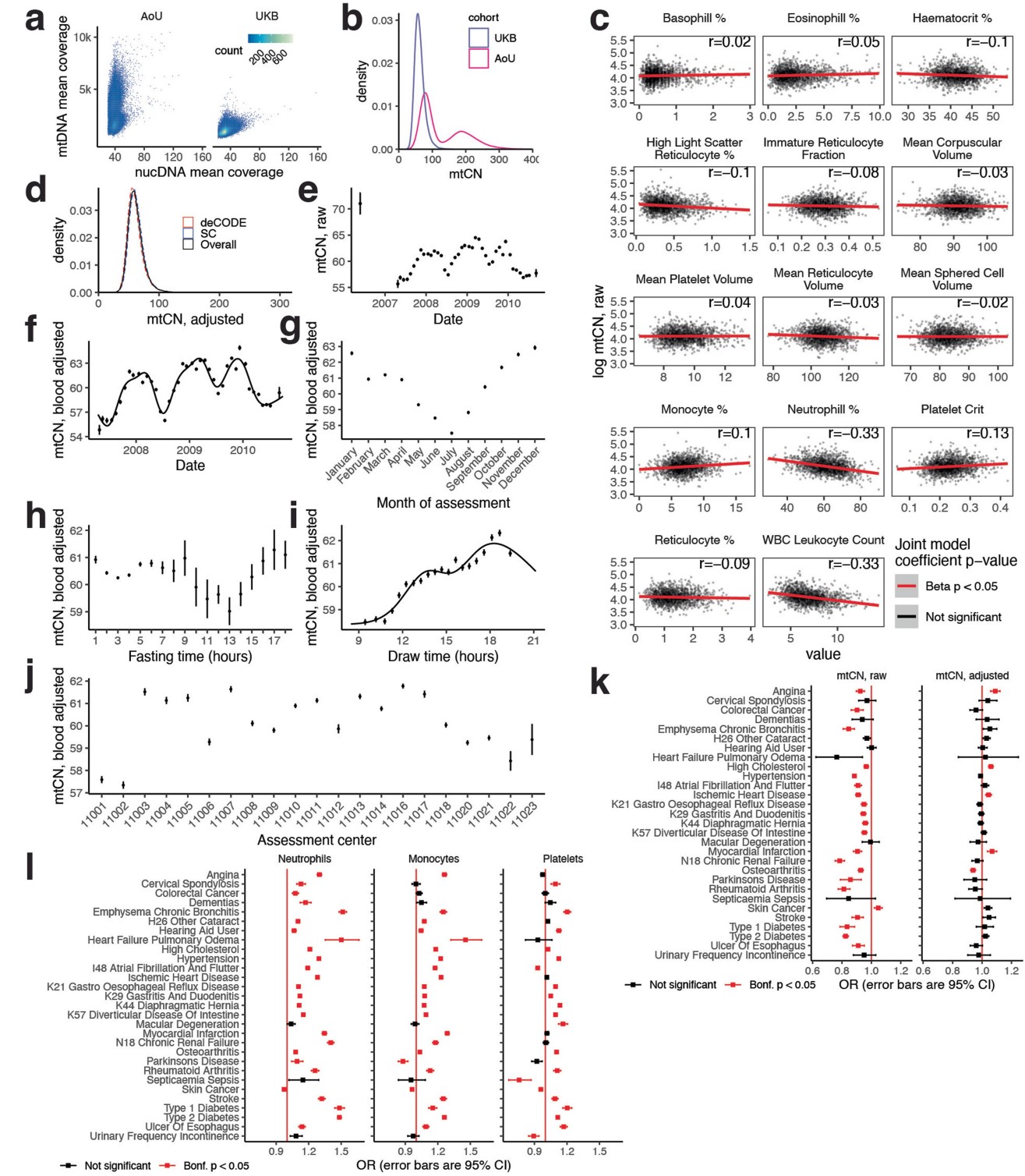

**Extended Data Fig. 3 | mtCN shows substantial correlations with technical and biological covariates. a.** Bivariate mean coverage distributions across nucDNA and mtDNA in AoU and UKB. **b.** Distributions of mtCN per diploid nuclear genome across AoU and UKB. **c.** Correlations between log mtCN and blood cell traits in UKB. Line corresponds to ordinary least squares fit; line color corresponds to the raw coefficient p-value of a joint model regressing log mtCN$_{raw}$ on all blood cell phenotypes. Inset is Pearson correlation coefficient. **d.** Distribution of mtCN$_{adj}$ in UKB. Color corresponds to sequencing center; black is the combined density. **e.** Mean mtCN$_{raw}$ versus assessment date, binned into months. Total N = 196,372. Pilot month samples are removed from subsequent analyses. **f.** Mean blood-corrected mtCN as a function of assessment date; line is

a natural spline with knots positioned seasonally; total N = 179,626. **g.** Mean blood-corrected mtCN as a function of assessment month; total N = 179,626. **h.** Mean blood-corrected mtCN as a function of self-reported fasting time; total N = 179,623. **i.** Mean blood-corrected mtCN as a function of draw time; line corresponds to natural spline with 5 knots; total N = 179,601. **j.** Mean blood-corrected mtCN as a function of assessment center; total N = 179,626. **k.** OR of raw and corrected mtCN in predicting 29 common diseases in UKB. **l.** OR of top blood cell composition traits in predicting any of 29 common diseases in UKB. For **e-j** error bars correspond to mean +/− 1 s.e.m.. For **k-l** error bars correspond to 95% CI around the OR, and sample sizes for each comparison can be found in Supplementary Table 8. All tests are two-sided.

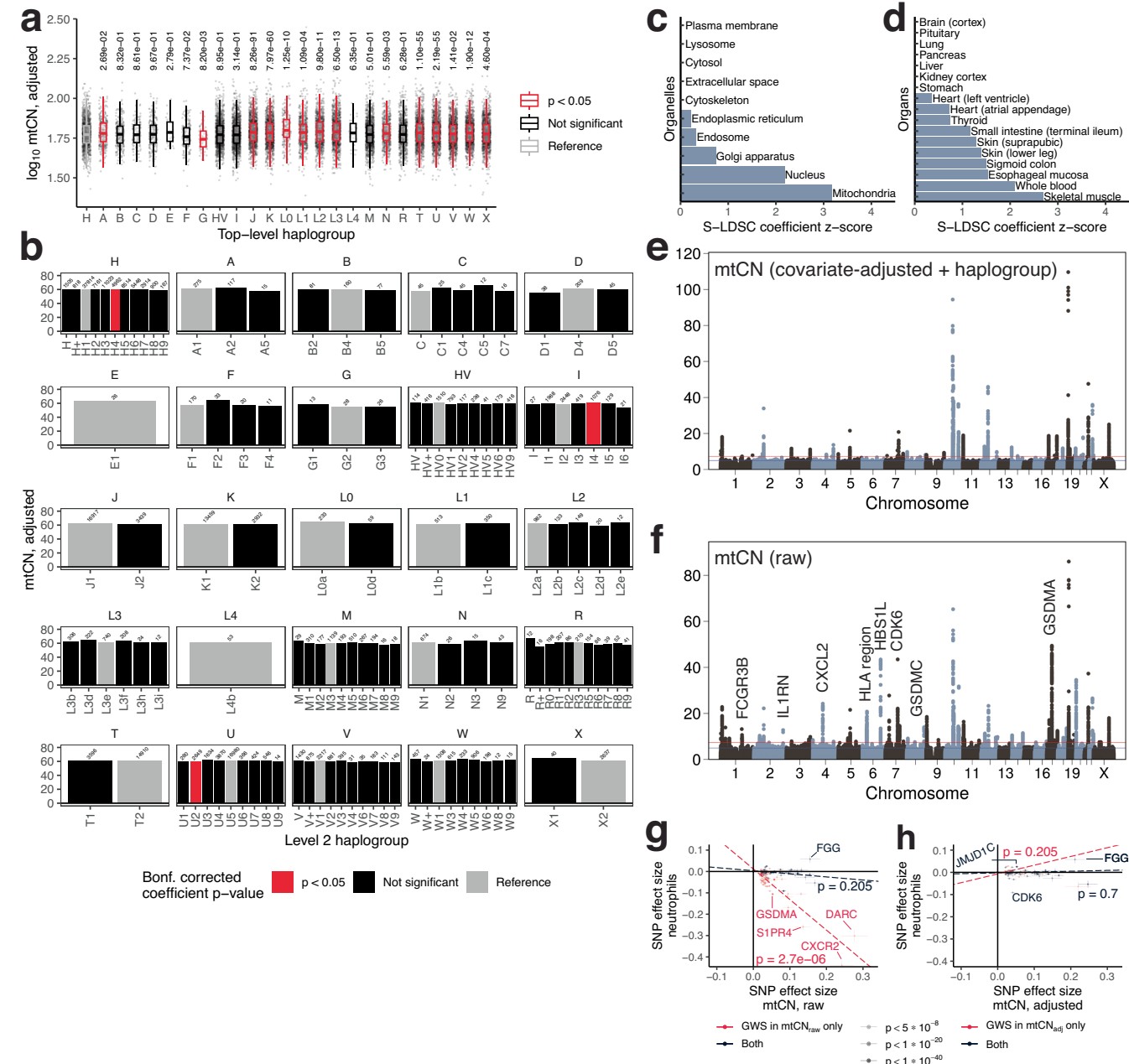

**Extended Data Fig. 4 | The genetic architecture of mtDNA copy number is influenced by blood cell traits but not haplogroup. a**. log$_{10}$ mtCN$_{adj}$ per diploid nuclear genome as a function of major top-level haplogroup. Points have been downsampled to at most 1,000 per haplogroup. Color and inset represents two-sided raw regression coefficient p-value from a joint linear model regressing log mtCN onto top-level haplogroup. **b**. Mean mtCN$_{adj}$ as a function of "level 2" haplogroup. Colors correspond to two-sided coefficient p-values for a joint model regressing log mtCN onto level 2 haplogroups within each top-level haplogroup, corrected for multiple testing using the Bonferroni approach across 25 top-level haplogroups. **c**. Enrichment of genome-wide signal near genes annotated to localize to each organelle and **d**. near genes highly expressed in each tissue. **e**. GWAS Manhattan plot of mtCN$_{adj}$ additionally

corrected for top-level haplogroup. **f**. GWAS Manhattan plot of mtCN$_{raw}$. Labels indicate genes proximal to a non-exhaustive set of selected loci with substantially less-significant p-values in the corrected analysis. **g**. Correlation between effect sizes for lead SNPs at GWS detected for raw mtCN between mtCN$_{raw}$ and neutrophil count. **h**. Correlation between effect sizes for lead SNPs at GWS detected for mtCN$_{adj}$ between mtCN$_{adj}$ and neutrophil count. In panels **g-h**, error bars represent effect size +/- 1 s.e., dotted line corresponds to inverse variance weighted least squared regression line; inset corresponds to regression p-value. Regression fits were performed separately for loci genome-wide significant for both mtCN$_{raw}$ and mtCN$_{adj}$ (black) and for loci specific to each (red).

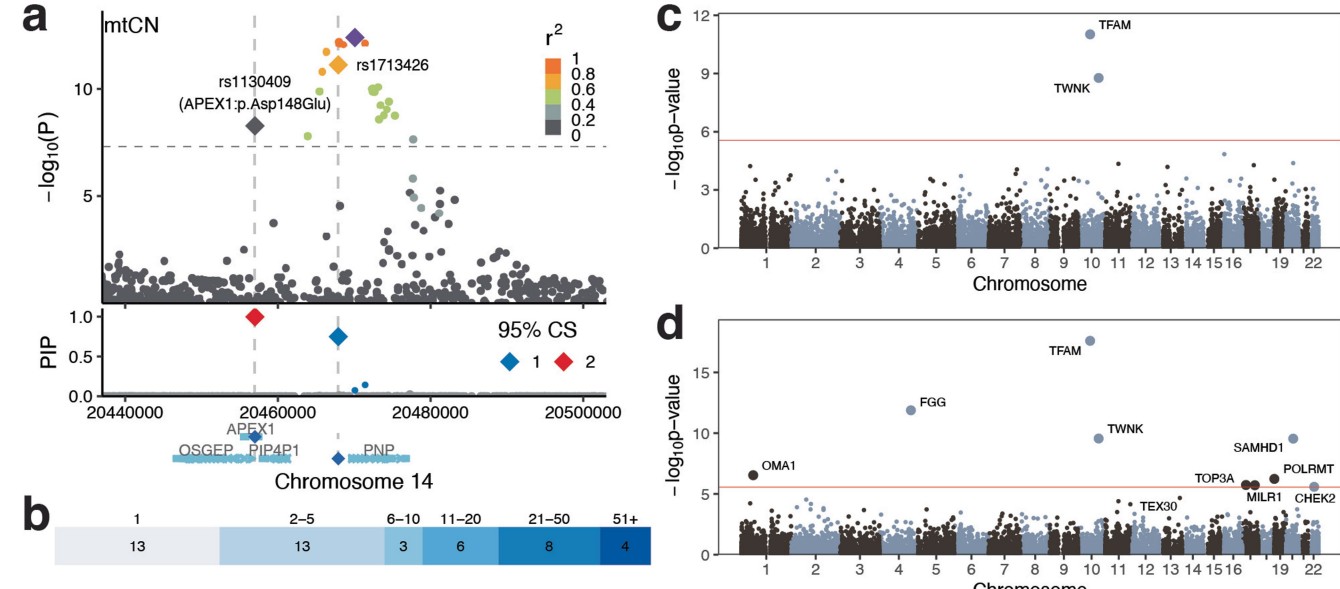

**Extended Data Fig. 5 | Fine-mapping and RVAS of UKB mtCN_adj. a.** Upper panel shows UKB mtCN_adj GWAS meta-analysis p-values at the chromosome 14 locus, visualized in GRCh38. Middle panel shows variants in the two 95% credible sets identified at this locus, with large diamonds corresponding to the highest PIP variants in each credible set. Bottom panel shows protein-coding gene annotations at this locus. Variant overlapping APEX1 is a missense variant in APEX1. **b.** Distribution of sizes of credible sets identified via fine-mapping for mtCN_adj. Numbers atop shaded region correspond to size of CS; numbers within shaded region corresponds to the count of credible sets of that size. RVAS gene-based Manhattan plot showing SKAT-O p-values using missense + LoF variation restricted to variants with MAF **c.** < 0.0001 and **d.** < 0.01. Red line is genome-wide significant at 0.05/number of genes tested.

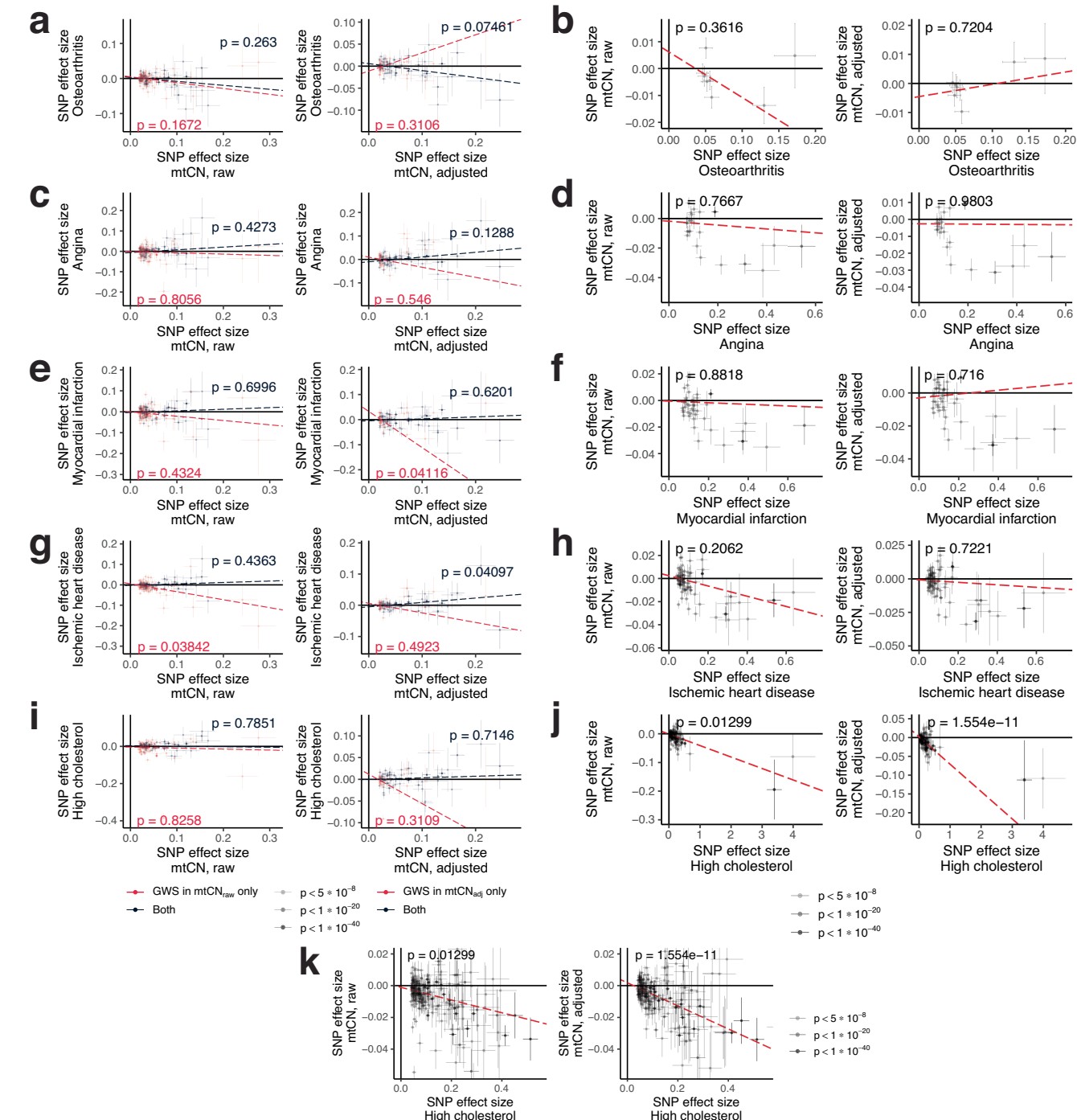

**Extended Data Fig. 6 | Bidirectional Mendelian randomization within UKB between mtCN and associated disease traits.** Correlation between effect sizes for lead SNPs detected for raw (left) and adjusted (right) mtCN between the respective mtCN phenotype and **a**. Osteoarthritis, **c**. Angina, **e**. Myocardial infarction, **g**. Ischemic heart disease, **i**. High cholesterol. Correlation between effect sizes for lead SNPs detected for **b**. Osteoarthritis, **d**. Angina, **f**. Myocardial infarction, **h**. Ischemic heart disease, **j**. High cholesterol and raw (left) and adjusted (right) mtCN. A zoomed-in version of **j** is shown in panel **k**. In all panels, points are GWAS effect sizes, error bars represent effect sizes +/− 1 s.e., dotted line corresponds to inverse variance weighted least squared regression line; inset corresponds to regression p-value. Regression fits were performed separately for loci genome-wide significant for both $mtCN_{raw}$ and $mtCN_{adj}$ (black) and for loci specific to each (red) for the analysis of mtCN effect on disease traits. Overall GWAS sample sizes are: Osteoarthritis – 420,473, Angina – 420,473, Myocardial infarction – 397,117, Ischemic heart disease – 419,724, High cholesterol – 420,473, mtCN raw – 163,372, mtCN adjusted – 163,372.

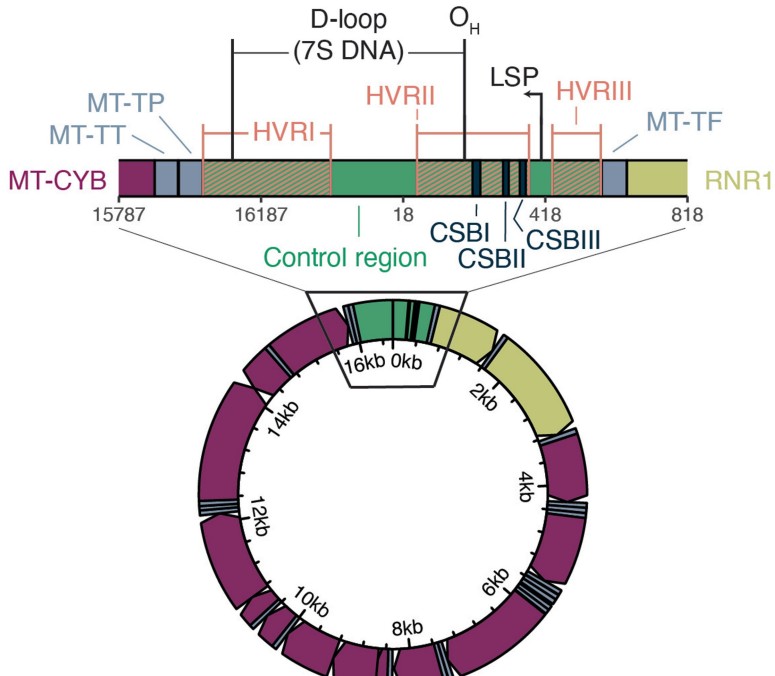

**Extended Data Fig. 7 | Organization of the mtDNA non-coding region.**
Colors indicate annotation type. Yellow, rRNA gene; steel, tRNA gene; purple, coding genes; green, non-coding region (also referred to as the control region); midnight, conserved sequence boxes (CSB); salmon stripe pattern, hyper- variable regions (HVR). The mtDNA D-loop refers to the region within the non-coding region often showing triple-stranded DNA due to the persistence of the 7S DNA. Annotations are oriented with the rCRS reference genome.

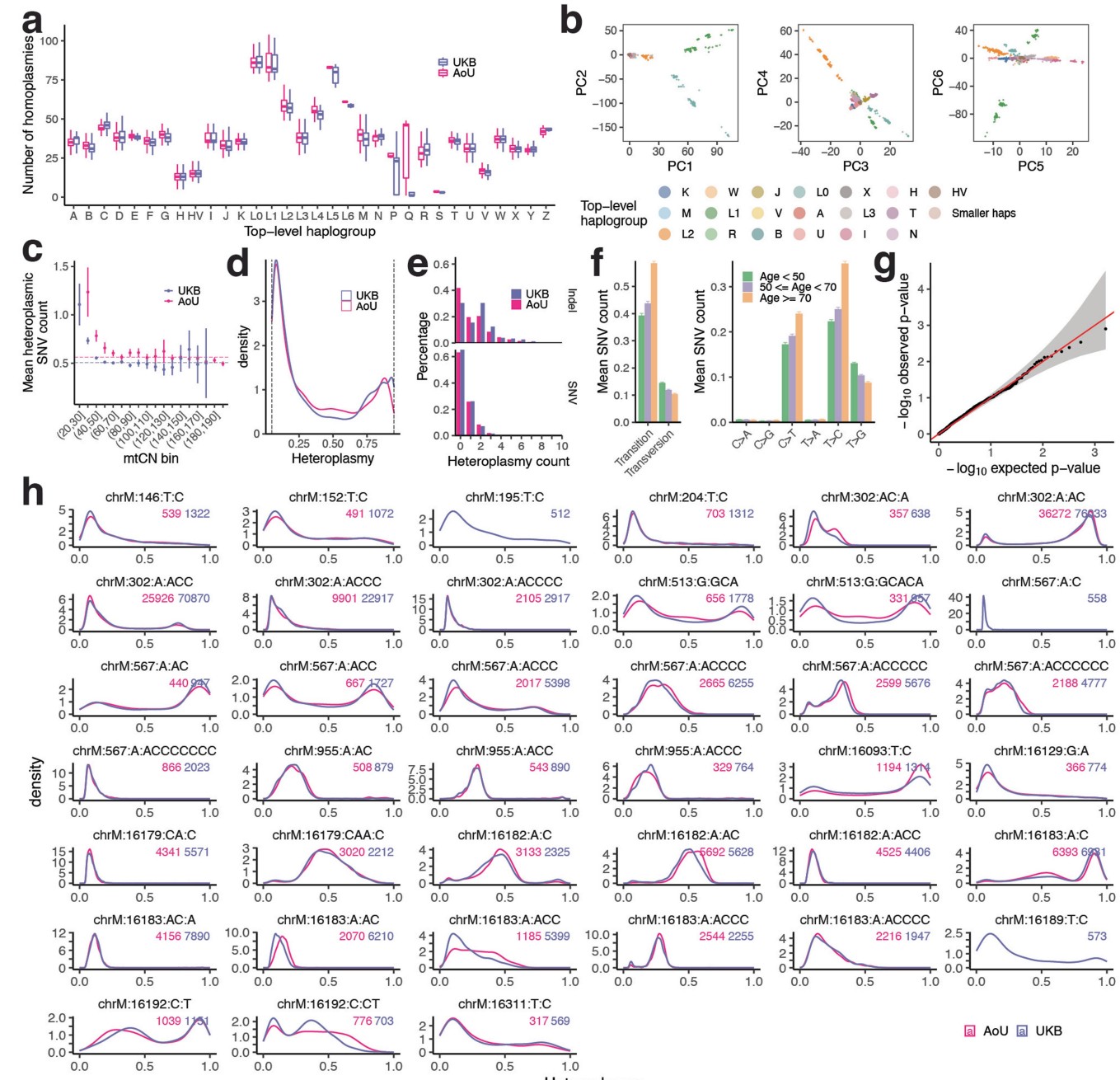

**Extended Data Fig. 8 | Overview of mtDNA variation across >250,000 individuals. a**. Box-and-whisker plots of homoplasmies per mtDNA haplogroup. Colors correspond to biobank. Outliers are suppressed to prevent visualizing AoU individual-level data. Total N = 95,343 (AoU) and 156,822 (UKB). **b**. Projection of UKB samples into mtDNA PC space computed using homoplasmies (MAF > 0.001). **c**. Mean heteroplasmic SNV count as a function of mtCN in UKB and AoU. Dotted lines correspond to mean number of heteroplasmic SNVs per person for individuals with mtCN > 50. Plot is truncated at mtCN < 200 for viewability. Error bars correspond to +/−1 s.e.m. Total N = 79,873 (AoU) and 199,832 (UKB). **d**.

Heteroplasmy distributions restricted to between 0.05 and 0.95 across UKB and AoU. **e**. Histogram of heteroplasmy counts per person for indels (top) and SNVs (bottom). **f**. Mean SNV count identified per-person in AoU as a function of variant type and age group. Error bars are +/−1 s.e.m. **g**. Quantile-quantile plot of p-values from logistic regression tests predicting case/control status of 29 common diseases in UKB using each of 39 common case-only heteroplasmies (see panel **h**). Black line is null expectation, ribbon is 95% CI around null expectation. **h**. Case-only heteroplasmy distributions of 39 variants detected in >500 UKB samples.

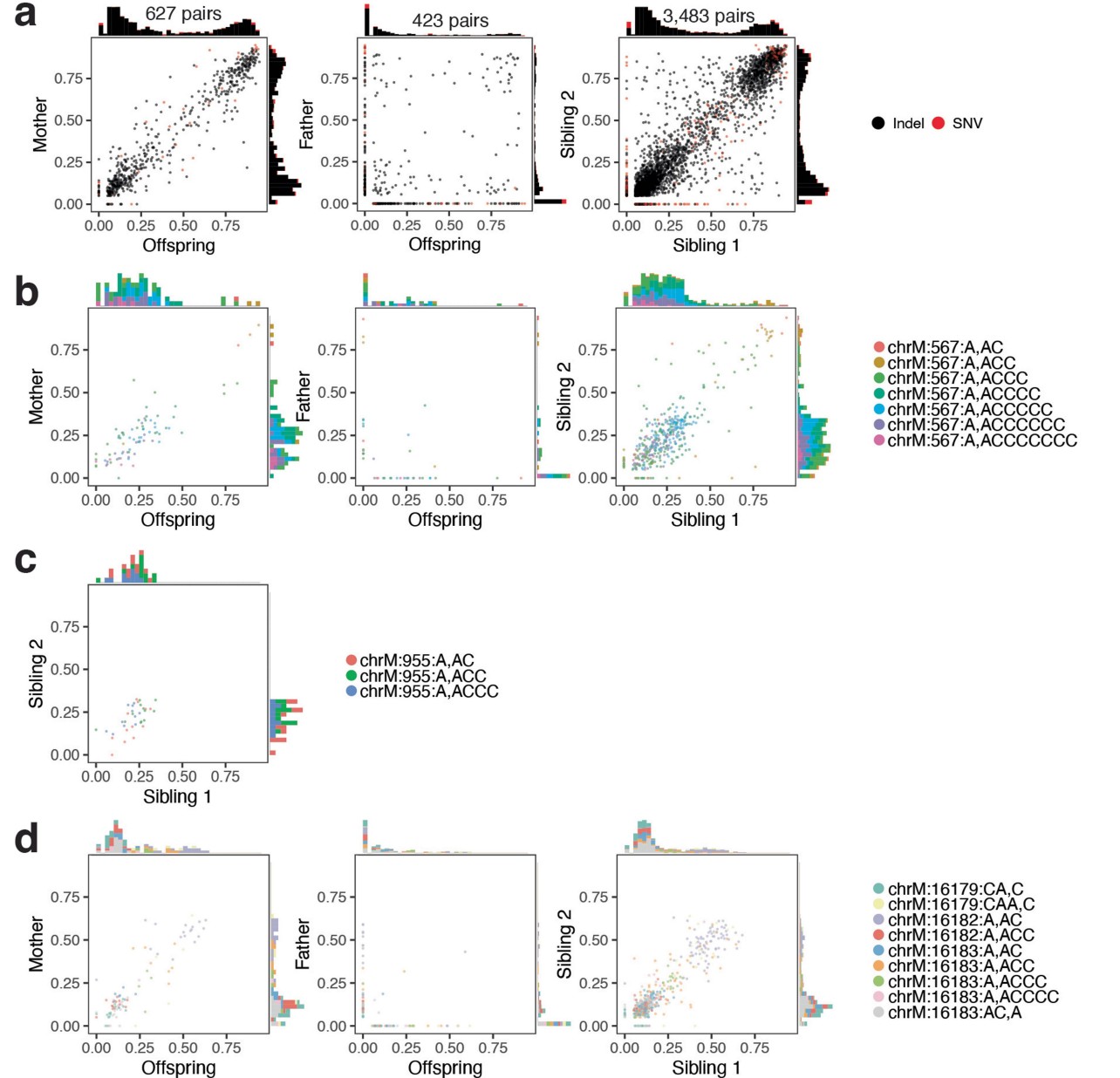

**Extended Data Fig. 9 | Transmission patterns of mtDNA heteroplasmic variants used for nuclear genetic analysis. a.** Heteroplasmy correlations for 39 common heteroplasmies (see Extended Data Fig. 8h). Inset text corresponds to the number of familial pairs included in the analysis. **b.** Heteroplasmy correlations for all tested variants at position 567. **c.** Heteroplasmy correlations

for all tested variants at position 955. **d.** Heteroplasmy correlations for all tested variants at positions 16179–16183. For panels **a**, **b**, and **d**, individual plots correspond to mother-offspring (left), father-offspring (middle), and sibling-sibling pairs (right). For panel **c**, the single plot corresponds to sibling-sibling pairs. For all panels, corresponding legend is on the right.

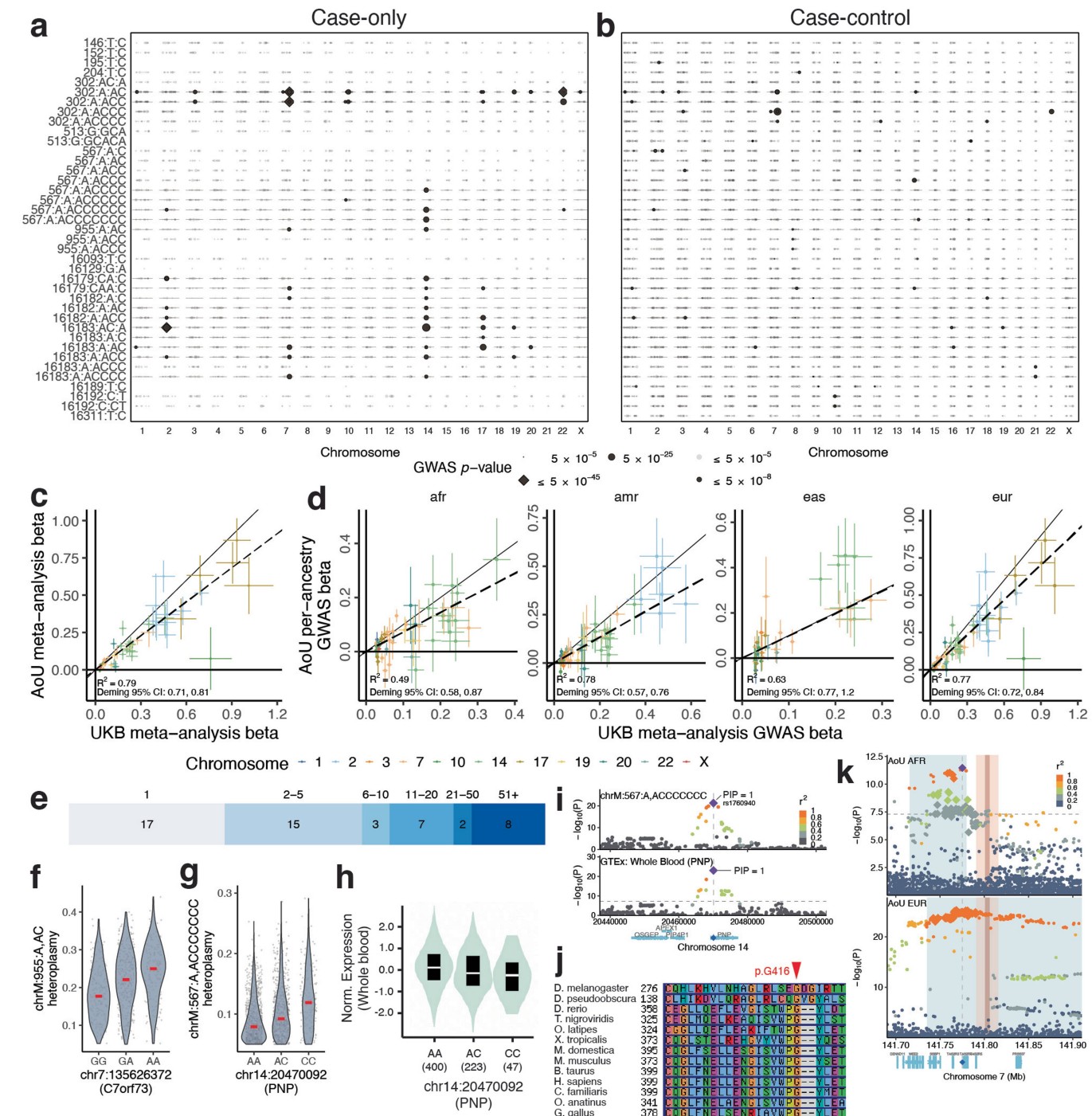

**Extended Data Fig. 10 | The full landscape of nuclear genetic associations to common mtDNA heteroplasmies. a**. Lead SNP p-values across all 39 tested case-only mtDNA heteroplasmies in the style of Fig. 4f. **b**. Lead SNP p-values across all 39 tested mtDNA heteroplasmies when coded as case-control phenotypes. **c**. Replication of lead SNP-variant pairs tested in both the UKB meta-analysis and AoU meta-analysis for case-only heteroplasmy. **d**. Replication of lead SNP-variant pairs tested in the UKB meta-analysis with each AoU continental ancestry group. For **c-d**, error bars correspond to effect size +/− 1 s.e.; colors correspond to nuclear chromosome; overall GWAS sample sizes can be found in Supplementary Table 1. **e**. The distribution of 95% credible set sizes from all heteroplasmy GWAS. Numbers atop shaded region correspond to size of CS; numbers within shaded region corresponds to the count of credible

sets of that size. **f**. chrM:955:A,AC heteroplasmy as a function of lead SNP genotype near C7orf73. **g**. chrM:567:A,ACCCCCCC heteroplasmy as a function of highest PIP SNP genotype in PNP. **h**. Whole blood PNP expression as a function of the same highest PIP SNP genotype in GTEx. **i**. Colocalization between chrM:16183:AC,A at the PNP locus and PNP eQTL in whole blood, shown in GRCh38. **j**. Multiple sequence alignment across vertebrates of best bidirectional hits for POLG2 (BLASTP E<1e-3) displayed with ClustalW colors with effect of putative causal variant labeled. **k**. GWAS results in AoU for AFR and EUR in the vicinity of SSBP1 for chrM:302:A,AC, shown in GRCh38. Large points correspond to 95% CS from UKB meta-analysis, blue ribbon is region with LD R² > 0.8 to lead SNP, dark red ribbon is a reference NUMT, light red ribbon is a 20kb window around the reference NUMT.

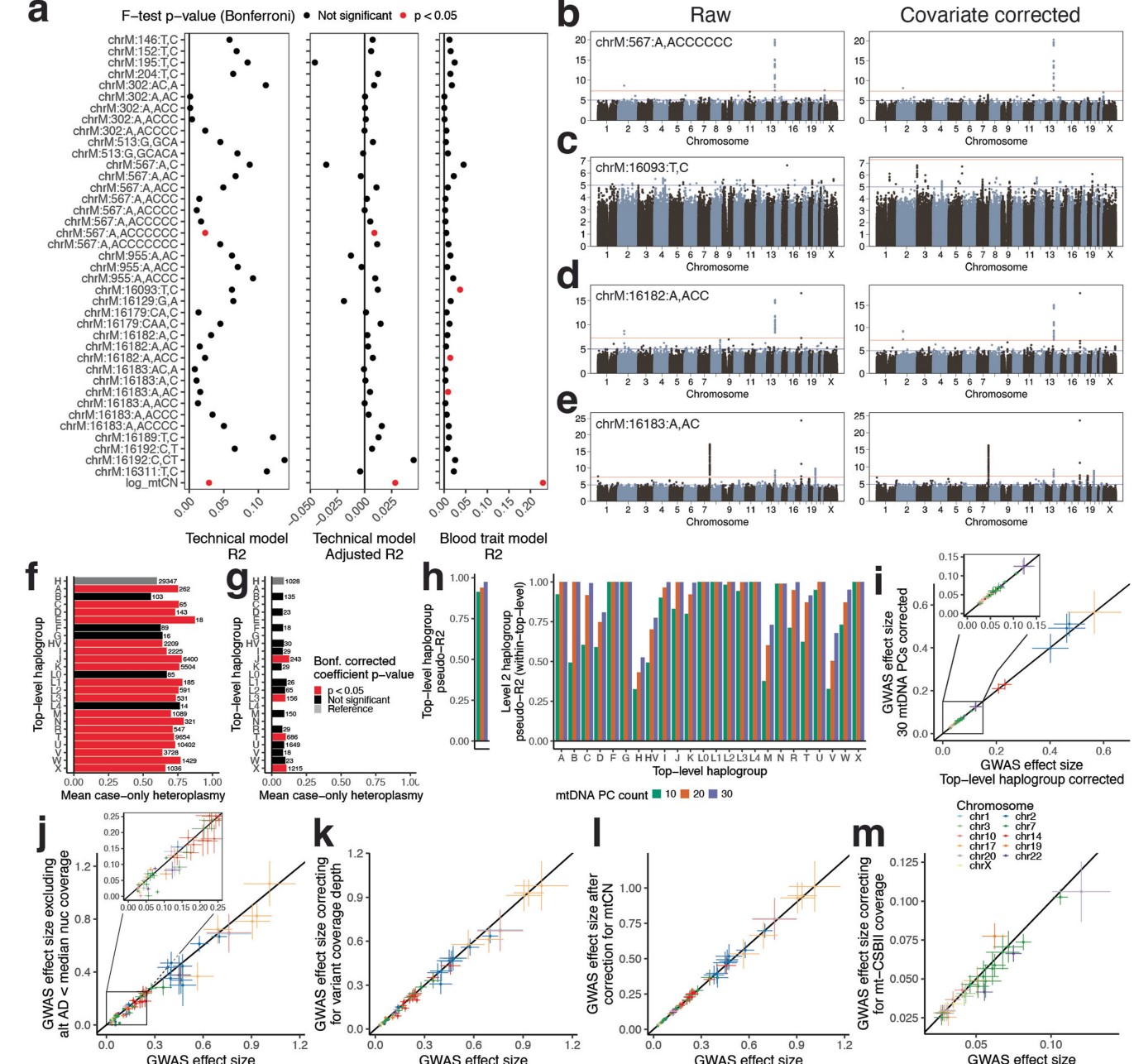

**Extended Data Fig. 11 | mtDNA heteroplasmy estimates and genetic associations are robust to potential confounders. a.** $R^2$ and adjusted $R^2$ for technical covariate model and $R^2$ for blood trait model for common mtDNA heteroplasmies and log mtCN$_{raw}$. Color corresponds to model F-test p-value < 0.05 (df = N-14 for blood, N-67 for technical; N in Supplementary Table 1) after Bonferroni correction. Sensitivity analyses for the GWASs of **b.** chrM:567:A, ACCCCCCC before and after technical covariate correction, **c.** chrM:16093:T,C before and after blood trait correction **d.** chrM:16182:A,ACC before and after blood trait correction **e.** chrM:16183:A,AC before and after blood trait correction. Mean case-only heteroplasmy as a function of top-level haplogroup for **f.** chrM:302:A,AC and **g.** chrM:16179:CA,C. Bar color corresponds to two-sided coefficient p-value for the regression of heteroplasmy onto top-level haplogroup, Bonferroni corrected for 39 tested heteroplasmies. **h.** McFadden's pseudo-$R^2$ for a multinomial model of top-level haplogroup versus mtDNA PCs (left) and "level 2" haplogroup versus mtDNA PCs within each top-level haplogroup. **i.**

GWAS lead SNP effect size estimate correlation when correcting for 30 mtDNA PCs vs correcting for only top-level haplogroup for selected variants showing high haplogroup heterogeneity (302:A,AC; 302:A,ACC; 302:A,ACCCC; 567:A, ACCCCCC; 955:A,ACC; 16179:CA,C; 16183:A,C). GWAS lead SNP effect size estimate correlation between case-only GWASs at baseline and **j.** GWASs after removing heteroplasmy calls supported by allele depth < median nuclear coverage, **k.** GWASs after correcting for variant coverage depth, **l.** GWASs after correcting for mtCN, **m.** length heteroplasmy GWASs after correcting for CSBII median coverage. For panels **i-m**, colors correspond to nuclear chromosome, points correspond to GWAS effect sizes for lead SNPs from baseline case-only GWASs with top-level haplogroup covariates, error bars represent effect sizes +/− 1 s.e., main GWAS sample sizes (x-axis) are found in Supplementary Table 1 (EUR), and sensitivity analysis GWAS sample sizes (y-axis) can be found in Supplementary Table 9.

# Reporting Summary

## Statistics

For all statistical analyses, confirm that the following items are present in the figure legend, table legend, main text, or Methods section.

| n/a | Confirmed | |
|---|---|---|
| ☐ | ☒ | The exact sample size (*n*) for each experimental group/condition, given as a discrete number and unit of measurement |
| ☐ | ☒ | A statement on whether measurements were taken from distinct samples or whether the same sample was measured repeatedly |
| ☐ | ☒ | The statistical test(s) used AND whether they are one- or two-sided *Only common tests should be described solely by name; describe more complex techniques in the Methods section.* |
| ☐ | ☒ | A description of all covariates tested |
| ☐ | ☒ | A description of any assumptions or corrections, such as tests of normality and adjustment for multiple comparisons |
| ☐ | ☒ | A full description of the statistical parameters including central tendency (e.g. means) or other basic estimates (e.g. regression coefficient) AND variation (e.g. standard deviation) or associated estimates of uncertainty (e.g. confidence intervals) |
| ☐ | ☒ | For null hypothesis testing, the test statistic (e.g. *F*, *t*, *r*) with confidence intervals, effect sizes, degrees of freedom and *P* value noted *Give P values as exact values whenever suitable.* |
| ☐ | ☒ | For Bayesian analysis, information on the choice of priors and Markov chain Monte Carlo settings |
| ☒ | ☐ | For hierarchical and complex designs, identification of the appropriate level for tests and full reporting of outcomes |
| ☐ | ☒ | Estimates of effect sizes (e.g. Cohen's *d*, Pearson's *r*), indicating how they were calculated |

*Our web collection on statistics for biologists contains articles on many of the points above.*

## Software and code

Policy information about availability of computer code

| | |
|---|---|
| Data collection | No data was collected in this study as we analyzed existing whole genome sequencing data. |
| Data analysis | We release the full WDL pipelines and associated input files for mtDNA analysis from whole genome sequencing data on GitHub (https://github.com/rahulg603/mtSwirl; DOI: 10.5281/zenodo.8067503). We also provide the code we used to run the pipeline on the UKB Research Analysis Platform, AoU, and Terra, consolidate all data, perform mtDNA sample and variant QC, and run GWAS. See Methods and the README in the GitHub repository for more information on how to use the pipeline. Several tools were used as part of mtSwirl, including GATK v4.2.6.0 (https://gatk.broadinstitute.org/), samtools v1.9 (https://github.com/samtools/samtools) and bcftools v1.16 (https://github.com/samtools/bcftools), Haplochecker 0124 https://github.com/genepi/haplocheck), R v3.1.1 (https://r-project.org), Hail v0.2.84 (https://hail.is), and UCSC kent tools source version 430 (genome-source.soe.ucsc.edu/kent.git and https://hgdownload.soe.ucsc.edu/admin/exe/linux.x86_64/). |
|  | We used several published tools and scripts to perform downstream analysis of the mtDNA callset in this study. All data wrangling, statistical analysis, and figure generation was performed using either Hail v0.2.98 (https://hail.is), python v3.7.10 (https://www.python.org), or R v4.2.1 (https://r-project.org). Parallelization of tasks in UKB was performed using Hail Batch (in Hail v0.2.98) (https://batch.hail.is) and in AoU using Cromwell v77 (https://cromwell.readthedocs.io). GWAS was performed in UKB using SAIGE v1.1.5 (https://saigegit.github.io). For scaling of UKB GWAS, a custom modification of the GWAS pipeline from the Pan UKBB pan-ancestry GWAS was implemented (https://github.com/atgu/ukbb_pan_ancestry). Linear regression GWAS was performed in AoU using Hail. We release the code used for GWAS on both UKB and AoU on GitHub (https://github.com/rahulg603/mtSwirl). mtDNA PCA was performed in R using the irlba v2.3.5.1 package (https://cran.r-project.org/web/packages/irlba/index.html). Multinomial models were trained using the nnet v7.3-17 package in R (https://cran.r-project.org/web/packages/nnet/index.html). Circos plots were made using the circlize package v0.4.15 in R (https://jokergoo.github.io/circlize_book/book/). For analysis of chrM:302 in single cell data, we used BedTools v2.29.2 (https://bedtools.readthedocs.io). LD clumping was performed using Plink v1.90 (https://www.cog-genomics.org/plink/). Finemapping was performed using FINEMAP-inf v1.3 and SuSiE-inf v1.2 (https:// |

github.com/FinucaneLab/fine-mapping-inf). eQTL data was obtained from GTEx v8 (https://gtexportal.org) and the eQTL catalogue release 4 (https://www.ebi.ac.uk/eqtl/). For replication analysis effect size comparisons, the deming v1.4 package was used in R (https://cran.r-project.org/web/packages/deming/index.html). Heritability estimates and enrichment analyses were performed using stratified LD-score regression (https://github.com/bulik/ldsc). BLASTn v2.13.0 was used as available from the NCBI (https://blast.ncbi.nlm.nih.gov/Blast.cgi). MUSCLE v3.8.31 was used for protein sequence alignment (https://drive5.com/muscle/downloads_v3.htm).

For manuscripts utilizing custom algorithms or software that are central to the research but not yet described in published literature, software must be made available to editors and reviewers. We strongly encourage code deposition in a community repository (e.g. GitHub). See the Nature Portfolio guidelines for submitting code & software for further information.

# Data

Policy information about availability of data

All manuscripts must include a data availability statement. This statement should provide the following information, where applicable:
- Accession codes, unique identifiers, or web links for publicly available datasets
- A description of any restrictions on data availability
- For clinical datasets or third party data, please ensure that the statement adheres to our policy

In terms of data processed or generated as part of this study, we provide per-population mtDNA heteroplasmic and homoplasmic allele frequencies and counts in UKB and AoU (Supplementary tables 5, 6), genetic association statistics for LD-independent lead SNPs and fine-mapped variants in UKB in addition to colocalization results (Supplementary tables 2-4), and gene-based RVAS association statistics for genes passing genome-wide significance for the Cauchy test (Supplementary table 7). All GWAS sample sizes for each genetic ancestry group, meta-analysis, and phenotype can be found in Supplementary table 1. All GWAS summary statistics from UKB cross-ancestry meta-analyses (used here in discovery analyses) have been deposited in GWAS Catalog (ID: GCP000614). Summary statistics containing all per-ancestry association statistics as well as cross-ancestry meta-analyses can be accessed via Google Cloud Platform (bucket: gs://mito-wgs-public-2023). Full GWAS summary statistics from AoU (used here as a replication cohort) have been deposited in a workspace available on AoU workbench (titled "Nuclear genetic control of mtDNA copy number and heteroplasmy in humans"; https://workbench.researchallofus.org/workspaces/aou-rw-3273c7f0/nucleargeneticcontrolofmtdnacopynumberandheteroplasmyinhumans/data). Individual level data generated as part of UKB (mtDNA copy number and mtDNA variant calls) have been returned to UKB to enable utilization of the full individual-level data by the broader scientific community via the UKB data showcase. Individual level data generated as part of AoU have been deposited in the same workspace containing summary statistics on the AoU Research Workbench. Please see our Github repository (https://github.com/rahulg603/mtSwirl) for more information on accessing these data. At the time of publication, access to the AoU workbench controlled-tier is restricted to US-based academic institutions, government entities, health care institutions, and non-profit organizations. Please also note that as of the time of publication, the only method to gain access to the AoU workspace containing the data generated here is to contact us to be added to the workspace. For information about access to the Researcher Workbench as a registered researcher, please visit https://www.researchallofus.org.

In terms of external data used in this study, we leveraged GWAS summary statistics, and ancestry-specific LD-matrices, and a curated list of 29 common, high-quality disease phenotypes generated as part of the Pan UKBB project 62. Paths for these summary statistics (https://pan.ukbb.broadinstitute.org/docs/per-phenotype-files) and LD-matrices (https://pan.ukbb.broadinstitute.org/docs/ld) can be found on the Pan UKBB project website (https://pan.ukbb.broadinstitute.org); these were accessed via Google Cloud Platform as part of this study. UKB phenotype and whole genome sequencing data can be accessed via the UKB Research Analysis Platform after completing a UKB access application (https://ukbiobank.dnanexus.com/landing). AoU phenotype and genotype data can be accessed via access to the Controlled Tier v6 on the AoU researcher workbench (https://workbench.researchallofus.org). gnomAD v3.1.2 (https://gnomad.broadinstitute.org) WGS was accessed via a custom Terra workspace (titled "gnomad_subsampled_mitopipeline_head_to_head"). High coverage WGS from 1000G was accessed using the public "1000G-high-coverage-2019" workspace in Terra. Published mtscATACseq data used for chrM:302 analysis can be obtained via approval from dbGaP. Gene-sets for enrichment analyses can be obtained using COMPARTMENTS (https://compartments.jensenlab.org) and MitoCarta 2.0 (https://www.broadinstitute.org/files/shared/metabolism/mitocarta/human.mitocarta2.0.html) as described previously 24. The GRCh37 and GRCh38 reference genomes as well as other standard reference data are available via the GATK resource bundle (https://gatk.broadinstitute.org/hc/en-us/articles/360035890811-Resource-bundle). Annotations for the baseline v1.1 and BaselineLD v2.2 models for S-LDSC as well certain other relevant reference data, including the HapMap3 SNP list, can be obtained from https://alkesgroup.broadinstitute.org/LDSCORE/. Known reference and polymorphic NUMTs were obtained from supplemental data as provided in published work 51,86–88.

# Human research participants

Policy information about studies involving human research participants and Sex and Gender in Research.

| Reporting on sex and gender | We did not perform new recruitment in this study and used existing cohorts and datasets. Sex was determined based on variables provided by UKB or via genetic inference in AllofUs. Sex is an important correlate with several observed phenotypes and has been used as a covariate for most analyses in this study. |
|---|---|
| Population characteristics | We did not perform new recruitment in this study and used existing cohorts and datasets. We did not filter on any relevant population characteristics in our analysis.<br><br>Regarding the characteristics of the datasets used in this study, briefly: UKB is a population-based cohort comprising ~500,000 individuals from ages 40-69 in the UK, recruited from sites across the country to cover a variety of socioeconomic settings and ensure an urban-rural mix (Sudlow et al. 2015 PLOS Medicine). AllofUs (AoU) is a population-based longitudinal cohort study in the US, enrolling participants age 18 or greater. AoU attempts to represent individuals otherwise underrepresented in biomedical research, and thus incorporates variables such as race, ethnic group, age, sex, gender identity, sexual orientation, disability status, income, and more in the recruiting strategy ("The 'All of Us' Research Program," 2019 NEJM). 1000G sampled participants across 26 populations around the world, assessing ~5 subgroups within each of 5 major continental populations to build a reference of genetic variation ("The 1000 Genomes Project Consortium" 2015 Nature, Byrska-Bishop et al. 2022 Cell). Phenotype data from this cohort was not used in this study. gnomAD v3 comprised opportunistically collected WGS data primarily from case-control studies of adult-onset common diseases, including cardiovascular disease, type 2 diabetes, and psychiatric disorders. Individuals with severe pediatric disease, or those with |

known first degree relatives of those with severe pediatric disease, were excluded from the cohort (Chen et al. 2022 bioRxiv, Karczewski et al. 2020 Nature). Phenotype data from this cohort was not used in this study. mtscATAC-seq data was obtained from one individual sequenced as part of recently analyzed MELAS cases (Walker et al. 2020 NEJM). Individuals were selected on the basis of a known diagnosis of carrying the m.3243A>G pathogenic variant, and not for population. All individuals from Walker et al. 2020 NEJM were male.

Neither AllofUs nor UKB select explicitly on diagnoses/treatment characteristics. Both sexes were represented in all cohorts except in mtscATAC-seq data. No filtering was performed on genotype information. The populations represented in our primary analyses are available in Supplementary table 1 and in Extended data figure 2. All datasets contained whole genome sequencing data, which was the focus of this study. More details can be found in Methods and in the relevant publications.

| Recruitment | We did not perform new recruitment in this study and used existing cohorts and datasets. See previously published literature for more information on the recruitment of individuals for the published datasets in this work: UK Biobank – Sudlow et al. 2015 PLOS Medicine; UK Biobank WGS – Halldorsson et al., 2022 Nature; AllofUs – "The 'All of Us' Research Program," 2019 NEJM; mtscATAC-seq data – Walker et al. 2020 NEJM. |
|---|---|
| Ethics oversight | We did not perform new recruitment in this study and used existing cohorts and datasets. Analysis of UK Biobank data was performed under UKB Application 31063. Analysis of AllofUs data was performed under Controlled Tier authorization in the workspace "Genetic determinants of mitochondrial DNA phenotypes". Institutional Review Board authorization of analysis of previously published single cell data was provided by Massachusetts General Hospital under protocol #2016P001517. |

Note that full information on the approval of the study protocol must also be provided in the manuscript.

# Field-specific reporting

Please select the one below that is the best fit for your research. If you are not sure, read the appropriate sections before making your selection.

☒ Life sciences ☐ Behavioural & social sciences ☐ Ecological, evolutionary & environmental sciences

For a reference copy of the document with all sections, see nature.com/documents/nr-reporting-summary-flat.pdf

# Life sciences study design

All studies must disclose on these points even when the disclosure is negative.

| Sample size | This was a biobank-scale analysis making use of all available whole genome sequencing samples in UK Biobank and AllofUs after quality control. Thus, no a priori sample size calculation was completed. This study comprises the largest analysis of mtDNA to date. |
|---|---|
| Data exclusions | Data were excluded during the study procedure for quality control or power purposes only. In brief, we excluded samples with evidence of contamination and with abnormal overlapping homoplasmy variant calls from all analysis. The former could produce incorrect mtDNA phenotype estimation or nucDNA genotype calls; the latter may indicate abnormalities in variant calling. We also excluded samples collected in the UKB pilot in 2006 as these samples had abnormal mtDNA copy number estimates. For variant analysis, we additionally excluded samples with low mtDNA copy number due to an established risk of NUMT contamination. We developed the "overlapping homoplasmy" filter and the "UKB pilot" filter during the study; the others were used previously (Laricchia et al. 2022 Genome Res) and pre-established. For genetic analyses, we restricted to samples with high confidence continental ancestry assignments in UKB (see http://pan.ukbb.broadinstitute.org) and AllofUs, and only performed GWAS for ancestry groups with enough measurements for interpretability. See Methods for more details. |
| Replication | We attempted replication for the two major components of the study: nuclear genetic analyses of (1) mtDNA copy number and (2) heteroplasmy. For (1), we obtained loci identified by the largest GWAS of mtCN previously completed by Longchamps et al. 2022; we successfully identified the vast majority of previously identified loci in our study (>85% at a stringent threshold of $p < 5e-5$). For (2), we used AoU to perform independent replication of heteroplasmy associations identified in UKB. We saw strong effect size concordance between cross-ancestry meta-analyses performed in either biobank ($R2 = 0.79$). |
| Randomization | We did not perform experimental group assignment in this study and performed genetic analysis using data from all samples that passed QC. Thus, as most of this work is population-based, this is not relevant for most of our work. In general, we extensively address potential confounders by leveraging the deep phenotyping in UK Biobank, testing and correcting for confounders such as blood cell composition, blood draw time, blood draw season, haplogroup, and others – see Methods. All genetic analyses included further corrections for population stratification by including genotype PCs computed within each genetic ancestry group, as well as age, sex, age2, age*sex, and age2*sex. Finally, for case/control disease trait associations with mtDNA phenotypes, we corrected for haplogroup, genetic ancestry group, and the aforementioned age and sex covariates. See Methods for more details. |
| Blinding | Blinding was not relevant for this study as experimental group assignment was not performed. |

# Reporting for specific materials, systems and methods

We require information from authors about some types of materials, experimental systems and methods used in many studies. Here, indicate whether each material, system or method listed is relevant to your study. If you are not sure if a list item applies to your research, read the appropriate section before selecting a response.

## Materials & experimental systems

| n/a | Involved in the study |
|-----|----------------------|
| ☒ | Antibodies |
| ☒ | Eukaryotic cell lines |
| ☒ | Palaeontology and archaeology |
| ☒ | Animals and other organisms |
| ☒ | Clinical data |
| ☒ | Dual use research of concern |

## Methods

| n/a | Involved in the study |
|-----|----------------------|
| ☒ | ChIP-seq |
| ☒ | Flow cytometry |
| ☒ | MRI-based neuroimaging |

