## [Peer Review File · Nature]

Manuscript Title: Nuclear genetic control of mtDNA copy number and heteroplasmy in humans

Reviewer Comments & Author Rebuttals

Reviewer Reports on the Initial Version:

Referees' comments:

Referee #1 (Remarks to the Author):

In the manuscript by Gupta, Neale, Mootha, and colleagues, the authors investigate covariation between germline nuclear DNA genotypes and mtDNA copy number/genotypic variation. The manuscript is extensive and covers significant territory, producing new results that challenge pre-existing literature on the determinants of mtDNA copy number variation in whole blood, patterns of inheritance in mtDNA polymorphisms, and nuclear determinants of mtDNA heteroplasmy. It is well-written, and at moments reads as the definitive manuscript on the topic. The manuscript reads longer than typical Nature papers, and I hope it stays that way. Below, I elaborate on some concerns and questions related to their findings that I hope the authors will find constructive.

1. I noted that the approach taken by MtSwirl of building self-reference sequences addresses potential issues in the mapping of mtDNA-derived reads from ancestries with high dissimilarity to the reference genome. I have two related questions: (a) do the changes in mtCN shown in EDF1b/c have any affect on the findings in the GWAS? Would the significant loci in Figure 1 have been identified without this approach? (b) Are some (probably low-heteroplasmy if answering in the affirmative) variants discovered by MtSwirl which would have been missed using a more conventional approach? I want to emphasize that it's OK to answer in the negative to both. But, if in the affirmative, it would be interesting to describe these at least briefly in the text, as they may have broader implications for understanding mtDNA genotypic variation in underrepresented groups.

2. What are the units in Fig 1C? Are these the number of mtDNA copies per haploid nuclear genome, or the absolute number of mtDNA copies? In either case, the number is slightly lower than I would have expected, although the single cell copy number data I'm familiar with is in solid tissues which of course are expected to have higher copy numbers (e.g. PMC6836731)

3. I enjoyed the analysis in Figure 2, especially as it blended a detailed mechanistic understanding of mtDNA replication with its implications on sequencing and sequencing depth. I was, however, confused about the inset in Panel B and the analysis that was done. As I understood it, the residuals from the regression immediately preceding "relatedness analysis in UKB" were used in the GWAS presented in Figure 2. This seems a bit strange, as the size of the residuals appears to increase as a function of coverage (and therefore, unless I'm wrong, absolute mtDNA copy number). In other words, mtDNA copy number and the residual are positively correlated. If this is the case, then the GWAS being performed is somehow (I'm imagining in a complicated way) confounded by the absolute mtDNA copy number of each sample. There are approaches to ameliorate this, which I think basically amount to standardizing the residuals around the baseline mtDNA copy number. I may be wrong on this, and if so I apologize, but if not, then it would be interesting to see what the results look like when correcting for this issue, and if the hits are preserved.

4. Have the authors considered making an MGME Thr265Ile construct and investigating if such a mutation disrupts DNA binding and processing of the intermediates in the CR? The structural

analysis is interesting but speculative.

5. The analysis presented in Figure 3 is, like the preceding analyses, very interesting. There is a confusing aspect to the presentation which I understand (the authors are plotting confidence intervals, but only coloring instances where the result survives FDR correction). This leads to a slightly puzzling visualization, where some significant hits are not colored. One way to address this is simply to comment on whether statistical power limits your ability to confidently detect these associations. Some certainly look like they would make the cut with just a few more samples.

6. The observation in Figure 4c is interesting, and although I really shouldn't be asking, I'd like to know how this stands with respect to other observations in the field. Has this been described before? In either case, I was hoping the authors could comment on the biological factors which might be driving the accumulation of SNPs but not indels with age. I'm not sure that the data in-hand is suitable for this sort of question, and I understand if their response is limited to informed speculation.

7. It is relatively well-appreciated in the literature that sequencing of homopolymers can be subject to artifacts and sequencing errors. It is somewhat obscured in Figure 4d, but my question relates to whether the heteroplasmies of indels specifically at long homopolymers such as 567, 16183 are preserved across siblings (it's already demonstrated for 302 in 5b). I suspect they will be and, if so, then this is a valuable piece of corroborating evidence that the heteroplasmy quantification is not subject to artifacts associated with sequencing homopolymers. If they are not associated across siblings, are there any replicate samples of the same individual that can be investigated to demonstrate reproducibility of homopolymer heteroplasmies? Related to this question, I wonder if the compositional form of the data (i.e. that an increase in 302A>ACC will necessarily decrease 302A or 302A>C) artificially influences any results. I suspect not, but perhaps the authors could comment on this.

8. I'm intrigued but also puzzled about the model the authors propose for the regulation of 302 heteroplasmies in Figure 5. If one of the 302 alleles (let's say A>AC) promotes additional DNA replication relative to the reference allele, should that not drive that mtDNA haplotype to fixation (and fast?, perhaps even over the lifetime of a single cell?). What prevents fixation of an mtDNA allele which has a replicative advantage?

9. As I said in my introduction, I enjoyed how much ground the manuscript covered. There were moments, though, where I felt the narrative flow or organization suddenly shift in an unexpected way. The best example of this was when the "Spectrum of mtDNA sequence variation across 253,583 individuals" section started. I don't have a specific suggestion, but it feels as if the key results of this section could be introduced earlier or even first without significantly disrupting the flow of the manuscript.

10. Perhaps I missed it: are detailed counts of snps/indels at each position from UKB/AllofUs available as a data supplement or in a repository? This would obviously be a significant resource for the community.

Referee #2 (Remarks to the Author):

In this study, Gupta and colleagues took advantage of existing large whole-genome DNA sequencing datasets (UKBB and AoU) to characterize variation between humans in terms of mitochondrial DNA (mtDNA) copy number and mtDNA sequence (heteroplasmy). Mitochondria plays essential roles in physiology, and it is therefore not surprising that defects in mitochondrial functions can result in human diseases (both rare and common). Here, the authors processed whole-genome sequence data to test previous reports that have connected mitochondria with human diseases, and to identify nuclear DNA sequence variants that associate (via GWAS) with

mtDNA copy number, mtDNA coverage and heteroplasmy. The novelty of the study resides in the size of the datasets, the breadth of the results, as well as the technical and analytical robustness of the analyses. Findings in this study will have many (immediate?) implications for cell biologists working on mitochondria, and also potentially investigators working on human diseases in which mitochondria are involved. The manuscript is clear, and the details provided in the Methods section are sufficient.

1. My only major comment is that I would have liked to see more links between the discoveries made in this paper and human phenotypic variation (beyond the mitochondria phenotypes described here). I appreciate that this may be the next step, but there is at least one simple analysis that could be done: the authors could take the GWAS variants identified here (one-by-one or in a polygenic score) and perform pheWAS analyses in available biobanks. If it associates with diseases or traits, what types (metabolic, etc.)? And if it does not associate, why?
2. A second related idea, which is more complicated but also more exciting, would be to test if these nuclear DNA sequence variants modify severity (penetrance, expressivity) in rare mitochondrial diseases. Probably outside the scope for this paper, but the idea could be discussed.
3. Cross-ancestry GWAS results are presented throughout the manuscript. Is it because the authors did not find ancestry-specific genetic associations?
4. Were gene-based tests to identify rare variants also performed? If not, why? In particular, would rare coding variants help confirm some of the genes implicated (or prioritized) by the GWAS fine-mapping or co-localization results?
5. Fig.5e. It looks like the one sample analyze in (e) does not have mitochondria with the reference allele (GmAG7, orange). In c and d, are they individuals with a similar profile (it does not seem, although maybe it is a super rare profile)? In other words, are they individuals in the large WGS datasets with heteroplasmy at chrM:302 that do not have the reference (orange?) allele?
6. (Minor) Provide complete legends for the supplementary tables.

Signed: Guillaume Lettre

Referee #3 (Remarks to the Author):

In "Nuclear genetic control of mtDNA copy number and heteroplasmy in humans" Gupta, Neale, Mootha and colleagues perform a large scale human genetics analysis to define common genetic variants that alter mtDNA copy number, mtDNA heteroplasmy and their phenotypic consequences. For this analysis they develop a new approach for calling mitochondrial DNA mutations that relies on a per individual consensus sequence rather than a reference genome. They apply this pipeline to whole genome sequencing (WGS) data from ~175,000 individuals from the UK Biobank and ~100,000 individuals from the All of Us cohort. They make several striking observations from this dataset including:

- (1) that much of the prior literature on mtDNA copy number in blood is confounded by a failure to fully account for blood cell counts
- (2) that the sequencing depth of coverage at the 7s DNA/DNA primer/RNA primer region can provide insights into mechanisms underlying mtDNA replication
- (3) that population scale datasets can be used to refine genotype/phenotype associations for mitochondrial disorders like MELAS
- (4) that many MTDNA heteroplasmy have both shared and distinct nuclear genetic determinants

Overall this analysis represents a landmark in the field of mitochondrial genetics both because of the scale of the data and the rigor of the analyses undertaken in particular the author's attention to minimizing confounding in the WGS derived mtDNA phenotypes and the potential artifacts introduced by nucDNA regions of mtDNA origin (NMUTs) in their genetic analyses.

The work could be strengthened in several ways:

(1) The authors derive mtDNA phenotypes from WGS data that is aligned to the hg38 reference genome however they do not take advantage of the dataset for their UK Biobank genetic analyses. Specifically, the genetic association studies in the UK Biobank are based on the hg19 imputed data from UKB-provided imputed v3 variants. As recently described by Halldorsson et al (Nature 2022) An analysis based on genetic variants identified with the whole genome sequencing data would likely identify more associated genetic variants (eg those that are poorly imputed) and also enhance discovery in non-European ancestry individuals where imputation panels are less accurate (particularly in rare variants). This analysis would perhaps identify germline genetic variants that are ancestry specific.

(2) Full use of genetic sequencing data would also enable rare variant association analyses in addition to the common variant association analyses. Such analyses would perhaps identify a convergence of common and rare genetic variants at a subset of loci and the resulting allelic series would further strengthen the conclusion that specific genes are causal.

(3) The variation in mtCN by month of assessment and time of day presented in Extended Data 3 is fascinating! Are there any germline genetic loci that associate with the residual between these two factors and the median value? It would be quite novel to identify genetic determinants of circadian rhythm or seasonal disorders identified through analyses of blood count corrected mtCN.

(4) A time-to-event model of incident events (eg mtCN predicting incident events with a cox proportional-hazards model) may be more appropriate than a logistic regression for the mtDNA CN analyses presented in figure 1f and the associated extended data figures given mtDNA CN changes with time and may change in response to particular disease events. Although the corrected mtCN association with MI is in the opposite direction than the uncorrected version, it would appear to be significant even correcting for multiple hypothesis testing ($p=1.4 \times 10^{-5}$). What do the authors make of this? Is the mendelian randomization for MI supportive of this link?

(5) How do the authors interpret the data presented in Fig 3 with respect to the pathogenicity of these variants in MELAS syndrome? Would they suggest that only one of the 10 variants is in fact pathogenic?

(6) In Fig 3, it is also notable that in the one variant that appears to be pathogenic (chrM:3243:A,G), the heteroplasmy is <25% for most samples. This may reflect selection against this variant in all cells or perhaps a cell lineage (as you showed in Walker, 2020 with the depletion of MELAS variants in T-cells). Several of the more common putative MELAS variants (eg chrM14484:T,C) appear to have a bimodal distribution. Is the phenotype different in carriers of these variants who have a low heteroplasmy compared to those with a high heteroplasmy?

(7) Can the authors conduct a phenome-wide association study to identify what other ICD code, biomarker or molecular phenotypes (eg NMR metabolomics) chrM:3243 is associated with and thereby expand the phenotypic spectrum associated with MELAS?

Minor considerations:

(8) An extended data figure (or figure part) illustrating the main questions/different kinds of mitochondrial DNA variation (eg copy number vs heteroplasmy vs replication events) that you are extracting may be helpful to make the manuscript easier to follow for the non-specialist.

(9) The variation that exists by assessment center (Extended data 3J) is notable. Does this persist after correcting for other factors identified (like draw time, month of assessment and so forth)? Does it correlate with socioeconomic factors, differences in ancestry or other demographic differences between assessment center?

(10) Do the authors observe any differences between the two UK biobank WGS sequencing centers with respect to mitochondrial copy number?

(11) The hg38 human reference genome was released ten years ago in 2013. Contemporary

human genetic analyses typically leverage hg38 in the absence of a compelling reason not to. It is unusual to have both hg19 and hg38 coordinates presented in the same manuscript. Here both are included without clear notation for the reader to follow. Several figures include both for example Figure 4g and 4i the y-axis refers to hg38 and the x axis refers to hg19.

(12) Fig 3: why include any variant with fewer than 10 carriers on the figure if associated phenotype data is not presented? It may be more space efficient to include only rows with phenotype data presented

(13) On page 51, line 20: there appears to be an erroneous inclusion of the phrase "Click or tap here to enter text" which is likely a typo.

(14) Page 12, line 43-44 "It is notable that prior studies have suggested that length variation at" [...], would consider referencing the prior study here.

(15) Data availability: individual level data can be shared within the All of Us workbench with any registered user. I would suggest depositing the individual mtCN and mt heteroplasmy call data within an All of Us Researcher Workbench workspace that can be viewed by any registered All of Us researcher as has been done recently for Master, Nature Medicine 2022; which was published in the Researcher Workbench Workspace Library (<https://workbench.researchallofus.org/library>).

-Alexander Bick

Author Rebuttals to Initial Comments:

Referee expertise:

Referee #1: mitochondrial genomics

Referee #2: human genetics and genomics

Referee #3: human genetics and genomics

Referees' comments:

Referee #1 (Remarks to the Author):

In the manuscript by Gupta, Neale, Mootha, and colleagues, the authors investigate covariation between germline nuclear DNA genotypes and mtDNA copy number/genotypic variation. The manuscript is extensive and covers significant territory, producing new results that challenge pre-existing literature on the determinants of mtDNA copy number variation in whole blood, patterns of inheritance in mtDNA polymorphisms, and nuclear determinants of mtDNA heteroplasmy. It is well-written, and at moments reads as the definitive manuscript on the topic. The manuscript reads longer than typical Nature papers, and I hope it stays that way.

We thank the Referee for the positive feedback.

Below, I elaborate on some concerns and questions related to their findings that I hope the authors will find constructive.

1. I noted that the approach taken by MtSwirl of building self-reference sequences addresses potential issues in the mapping of mtDNA-derived reads from ancestries with high dissimilarity to the reference genome. I have two related questions: (a) do the changes in mtCN shown in EDF1b/c have any affect on the findings in the GWAS? Would the significant loci in Figure 1 have been identified without this approach? (b) Are some (probably low-heteroplasmy if answering in the affirmative) variants discovered by MtSwirl which would have been missed using a more conventional approach? I want to emphasize that it's OK to answer in the negative to both. But, if in the affirmative, it would be interesting to describe these at least briefly in the text, as they may have broader implications for understanding mtDNA genotypic variation in underrepresented groups.

These are good questions. Due to significant computational costs, we only benchmarked mtSwirl and the “vanilla” method head-to-head using a diverse subset of the gnomAD cohort, based on which we believe the answer to both questions is no:

- (a) UKB predominantly consists of “European” haplogroups (**Extended data figure 2a**), with only 2466 individuals (<2%) with the “African” haplogroups L0, L1, and L2 (which show the largest copy number shifts in **Extended data figure 1c**). Thus, while a more diverse cohort for mtCN analysis may reveal greater benefits to using mtSwirl, we believe that in UKB the genetic architecture observed using mtSwirl will be highly similar to that expected using the “vanilla” method. We further tested our mtCN GWAS with top-level mtDNA haplogroup as a covariate, which would account for ancestry- and haplogroup-specific mean shifts in mtCN even using the “vanilla” pipeline, and found that inclusion of this covariate did not appreciably change the observed genetic architecture.
- (b) Based on our benchmarking in gnomAD, we find many low heteroplasmy variants that are detected using the “vanilla” method but *not* using mtSwirl (**Extended data figure 1f**). We believe that these calls were likely artifactual due to read misalignment, possibly from NUMTs, which we resolve using mtSwirl. In this paper, we conservatively restricted to heteroplasmy $\geq 5\%$ to avoid residual contamination from polymorphic NUMTs (Laricchia et al. 2022 Genome Res). We find that above 5%, variant calls are extremely similar between mtSwirl and the “vanilla” method (**Extended data figure 1f-1g**).

2. What are the units in Fig 1C? Are these the number of mtDNA copies per haploid nuclear genome, or the absolute number of mtDNA copies? In either case, the number is slightly lower than I would have expected, although the single cell copy number data I'm familiar with is in solid tissues which of course are expected to have higher copy numbers (e.g. PMC6836731)

The units of figures 1b and 1c are mtCN per diploid nuclear genome, based on the following formula: $mtCN = 2 * \text{mean or median mtDNA coverage} / \text{mean nucDNA coverage}$

For clarity, we have added the following text to the manuscript to be explicit about the units:

“Following adjustment for all identified covariates (**Methods, Supplementary note 2, 3**), we found that covariate-adjusted mtCN (which we term mtCN_{adj}) was unimodal in UKB across 178,134 subjects with an average of 61.66 copies per diploid nuclear genome (**Extended data figure 3d**).”

We have also updated the **Figures 1, 5** and **Extended data figure 3, 4** legends to clarify these units.

Regarding the mtCN detected in UKB, we agree with the reviewer that the observed median of ~60 mtDNA molecules / diploid nuclear genome number (i.e., ~60 mtDNA molecules/cell) is on the lower end of the expected spectrum of mtCN in blood. However, prior work has observed vastly different mtCN estimates across different datasets (Laricchia et al. 2022 Genome Res):

In AllofUs we observe a bimodal distribution of mtCN (**Extended data figure 3b**), with one AoU mode resembling that of UKB and the other mode centered much higher, around ~200 copies per diploid nuclear genome. At present we do not know the source of these differences, but we speculate it is due to differences in the DNA extraction method (e.g., UKB may have used a single method uniformly, whereas AofU samples may have been derived from a mix of protocols or may have involved DNA extraction using both whole blood and buffy coat fractions). Indeed, DNA extraction methods have been shown to explicitly influence the observed mtCN (Longchamps et al. 2020):

Thus, we believe that the low mtCN in UKB is likely a function of the technical procedure used to process blood samples, and we believe that differences in mtCN across individuals remain meaningful within UKB due to the uniformity of processing.

3. I enjoyed the analysis in Figure 2, especially as it blended a detailed mechanistic understanding of mtDNA replication with its implications on sequencing and sequencing depth. I was, however, confused about the inset in Panel B and the analysis that was done. As I understood it, the residuals from the regression immediately preceding “relatedness analysis in UKB” were used in the GWAS presented in Figure 2. This seems a bit strange, as the size of the residuals appears to increase as a function of coverage (and therefore, unless I’m wrong, absolute mtDNA copy number). In other words, mtDNA copy number and the residual are positively correlated. If this is the case, then the GWAS being performed is somehow (I’m imagining in a complicated way) confounded by the absolute mtDNA copy number of each sample. There are approaches to ameliorate this, which I think basically amount to standardizing the residuals around the baseline mtDNA copy number. I may be wrong on this, and if so I apologize, but if not, then it would be interesting to see what the results look like when correcting for this issue, and if the hits are preserved.

The reviewer properly understands the trait used in the GWAS – the inset panels were confusing. The previous insets intended to show the regression from which the residual was obtained, rather than the residuals themselves that were used for the GWAS. We have updated this figure now to explicitly show both the regression from which the residuals were obtained, and the residuals used for the GWAS, and have tweaked the text to reflect this.

The residuals were computed by using the discordance in DNA coverage between the DNA primer region and either the RNA primer region (**Figure 2b**) or the 7S DNA region (**Figure 2d**). As the scatterplots in **Figure 2b** and **Figure 2d** indicate, there is a strong linear relationship between coverages at the DNA primer, RNA primer, and 7S DNA regions – this is governed by mtDNA copy number. The discordance metric is obtained by taking the residuals from the regression of DNA primer coverage onto RNA primer coverage (**Figure 2b**, purple arrow) or DNA primer coverage onto 7S DNA coverage (**Figure 2d**, blue arrow). Since aggregate mtDNA copy number should not, in theory, differentially impact coverage in one region of mtDNA versus another, by analyzing the discordance between two mtDNA regions we are inherently correcting for mtCN. This residual, our measure of coverage discordance, is now indicated as a density beneath each scatter plot – this is the quantity used for GWAS. We hope that this new visualization, shown below, is clearer:

To be rigorous, we repeated our coverage discordance GWAS, now controlling for mtCN, and observed no notable differences in genetic architecture, confirming our approach:

4. Have the authors considered making an MGME Thr265Ile construct and investigating if such a mutation disrupts DNA binding and processing of the intermediates in the CR? The structural analysis is interesting but speculative.

We have toned down the speculation in the Main text:

“Missense variants in *POLG*, *MGME1*, and *MCAT* all show PIP > 0.1 after fine-mapping, and the highest PIP variant at the *MGME1* locus causes p.Thr265Ile, which is within the *MGME1* exonuclease domain (Figure 2d).”

And have moved the speculation into the Discussion:

“We speculate that the putatively causal variant in *MGME1*, p.Thr265Ile, may directly impact DNA binding by disrupting a hydrogen bond within a helix-forming part of the DNA binding pocket of the *MGME1* exonuclease domain (Figure 2d).”

Testing the impact of the variant is certainly interesting, but outside the scope of the current work. In the Discussion we now state:

“Future studies are required to evaluate the impact of the identified candidate causal variants on mtDNA replication and maintenance.”

5. The analysis presented in Figure 3 is, like the preceding analyses, very interesting. There is a confusing aspect to the presentation which I understand (the authors are plotting confidence intervals, but only coloring instances where the result survives FDR correction). This leads to a slightly puzzling visualization, where some significant hits are not colored. One way to address this is simply to comment on whether statistical power limits your ability to confidently detect these associations. Some certainly look like they would make the cut with just a few more samples.

We have now updated the main text to include a comment on limited power at the end of this section:

“Due to their low frequency of detection in the UKB sample, we do not have the statistical power to exclude the presence of more subtle intermediate phenotypes among the other tested variants.”

6. The observation in Figure 4c is interesting, and although I really shouldn't be asking, I'd like to know how this stands with respect to other observations in the field. Has this been described before? In either case, I was hoping the authors could comment on the biological factors which might be driving the accumulation of SNPs but not indels with age. I'm not sure that the data in-hand is suitable for this sort of question, and I understand if their response is limited to informed speculation.

(Please note that we have now revised Figure 4 such that the former Figure 4c is now Figure 4b.)

Certainly, age-associated accrual of heteroplasmic variation in the mtDNA has been discussed in the past. In the Discussion, we now cite a classic reference from Attardi's group (Michikawa et al. 1999 Science) showing an age-dependent increase in heteroplasmic mtDNA SNVs in humans:

“A striking finding from our work is that nearly everyone harbors heteroplasmic mtDNA variants obeying two key principles: (i) heteroplasmic SNVs are typically somatic and accrue with age sharply after age 70, while (ii) heteroplasmic indels are found in >60% of individuals, do not accrue with age, and are usually inherited as mixtures within the same maternal lineage. The accrual of point mutations with age has been reported (Michikawa et al., 1999), however to our knowledge the stability of detected indels with age has not been previously appreciated.”

However, to the best of our knowledge, we believe that we are the first to show that this behavior is related to the type of variation observed in the human population, with SNV heteroplasmies tending to be somatic and showing age accumulation and indel heteroplasmies tending to be much more common and tending to be quantitatively inherited with no age accumulation.

In theory, the observed accumulation of heteroplasmic SNVs with age could be driven by one or more of multiple factors:

1. The observed variants may reflect somatic replication errors that accumulate with age. We observe that heteroplasmic SNVs tend not to be correlated between mother and offspring, suggesting they were not transmitted in the germline. Prior work by Kennedy et al. 2013 (PLOS Genet) showed a very similar mutational spectrum for heteroplasmic SNVs to the one we observe (**Extended data figure 8f**), with the most frequent mutations being transitions (C->T and T->C variants). The authors of this study argue that these SNVs are a signature of polymerase gamma misincorporation or deamination reactions. Of note, the detected mutational spectra argue against oxidative damage as the 8-oxo-dG alteration commonly caused by oxidative damage tends to produce C->A transversions, which we (and Kennedy et al. 2013) rarely observe.

Once these variants are introduced somatically (e.g., via replication errors), several mechanisms may allow for heteroplasmy fractions to rise above our 5% detection threshold in blood. For instance, this could occur via simple drift as mutations accrue over time. It is also possible that certain somatic mtDNA SNVs could exhibit a “selective advantage” at the level of mtDNA, mitochondria, or cells, pushing heteroplasmy fractions above 5% over time. Positive selection for certain somatic SNVs has been previously proposed (Li et al. 2015 PNAS).

2. The age-associated rise in detected variants could also be driven by pre-existing low levels of heteroplasmy followed by clonal hematopoiesis, which has been shown to follow a very similar age-related curve (Jaiswal et al. 2014 NEJM):

We believe that the stability of indel heteroplasmy burden with age is a novel observation. Specifically, we find that most detected indels (heteroplasmy > 5%) tend to be quantitatively inherited (**Figure 4d**) and tend to be located in the non-coding region (NCR) of the mtDNA (**Figure 4a**). This NCR predominance may reflect negative selection against indel heteroplasmies in the coding regions which would likely produce highly deleterious frameshift mutations. Negative selection may thus oppose age-related somatic indel variation accrual at most mtDNA positions, resulting in

the detectable heteroplasmic indel burden being dominated by indel length polymorphisms that are not under high negative selection (i.e., in the non-coding region, where we find a high burden of inherited indels which have existed in the maternal lineage).

7. It is relatively well-appreciated in the literature that sequencing of homopolymers can be subject to artifacts and sequencing errors. It is somewhat obscured in Figure 4d, but my question relates to whether the heteroplasmies of indels specifically at long homopolymers such as 567, 16183 are preserved across siblings (it's already demonstrated for 302 in 5b). I suspect they will be and, if so, then this is a valuable piece of corroborating evidence that the heteroplasmy quantification is not subject to artifacts associated with sequencing homopolymers. If they are not associated across siblings, are there any replicate samples of the same individual that can be investigated to demonstrate reproducibility of homopolymer heteroplasmies? Related to this question, I wonder if the compositional form of the data (i.e. that an increase in 302A>ACC will necessarily decrease 302A or 302A>C) artificially influences any results. I suspect not, but perhaps the authors could comment on this.

We agree that several sites of interest (302, 567, 16183) are adjacent to homopolymeric stretches and thus require special care in interpretation. We have newly produced **Extended data figure 9** which shows the transmission patterns of all variants included in UKB GWAS (**Extended data figure 9a**), all variants at position 567 used for GWAS (**Extended data figure 9b**), all variants at position 955 used for GWAS (**Extended data figure 9c**), and all variants at positions 16179-16183 used for GWAS (**Extended data figure 9d**). In all cases, we observe the expected pattern of maternal-offspring and sibling-sibling heteroplasmy sharing as seen for chrM:302 (**Figure 5b**) and for all heteroplasmic variants in UKB (**Figure 4c**). We include only siblings at position 955 as this was a rare heteroplasmic variant and we observed it too infrequently to discern a meaningful pattern among UKB parent-offspring pairs. We have reproduced this new figure below:

Extended data figure 9. Transmission patterns of mtDNA heteroplasmic variants used for nuclear genetic analysis.

a. Heteroplasmy correlations for 39 common heteroplasmies (see **Extended data figure 7i**). Inset text corresponds to the number of familial pairs included in the analysis. **b.** Heteroplasmy correlations for all tested variants at position 567. **c.** Heteroplasmy correlations for all tested variants at position 955. **d.** Heteroplasmy correlations for all tested variants at positions 16179-16183. For panels **a**, **b**, and **d**, individual plots correspond to mother-offspring (left), father-offspring (middle), and sibling-sibling pairs (right). For panel **c**, the single plot corresponds to sibling-sibling pairs. For all panels, corresponding legend is on the right.

In addition to the reassuring transmission pattern, there are several other lines of evidence indicating that these heteroplasmic variants are not pure artifact:

1. Not only does our GWAS yield statistically significant loci after multiple comparisons correction, but our identified nuclear loci are highly biologically plausible and implicate mitochondrial proteins involved in mtDNA replication and maintenance, and several have missense variation nominated as causal via fine-mapping (**Supplemental figure S3**; e.g., POLG2, DGUOK) and/or show colocalization with eQTLs (e.g., PNP).
2. The identified nuclear genetic architecture is shared across multiple mtDNA sites (e.g., 567, 955, 16179, 16182), with associations near PNP, DGUOK, and POLG2 arising for heteroplasmies across multiple different mtDNA regions.
3. Others in the past have also assayed length heteroplasmies at position 302, adjacent to the longest poly-C stretch in the mtDNA, using alternative methods and in alternative tissues. For instance, Lee et al. 2005 Ann. Hum Gen show quantification of length polymorphisms in the poly-C tract at position 302, including heteroplasmy of A>ACC and A>AC, using PCR amplification and size-based separation followed by analysis by electropherogram. Shin et al. 2004 performed clonal expansion of single CD34 cells followed by amplification and Sanger sequencing, identifying similar insertions at position 302.

We would not expect any of these observations if our mtDNA heteroplasmy calls originated primarily from amplification or sequencing artifact.

We have additionally completed a new analysis of whole genome sequences from 1000 Genomes, which uses EBV-transformed cell lines rather than whole blood and contains 602 trios. We find a strikingly similar heteroplasmy spectrum at the chrM:302 position as seen in UKB, and importantly also observe quantitative maternal transmission of heteroplasmy levels of the sites we focus on (302, 567, 955, 16179-16183). This implies that the observed heteroplasmy can be relatively stable through the process of EBV transformation and cell culture propagation in both maternal and offspring samples. We have upgraded our Figure 4 with a new panel **Figure 4d** which includes heteroplasmy transmission results for all detected variants from the 1000G cell lines. We have also added the following text to the results:

“We additionally analyzed WGS from 602 trios from 1000 Genomes (1000G), finding a similar pattern (**Figure 4d**). Unlike UKB blood samples, 1000G samples underwent EBV transformation to create cell lines prior to WGS (1000 Genomes Project Consortium et al., 2015; Byrska-Bishop et al., 2022), implying that the maintenance of these heteroplasmic indels is robust and can be quantitatively maintained through both maternal transmission and cell culture, albeit with some additional variance (**Figure 4d**).”

To confirm that this heteroplasmy transmission was also observed for the 39 variants tested in our GWAS (i.e., indel length variants near homopolymeric stretches), we performed an additional transmission analysis restricting to these 39 variants, observing very similar results with a high degree of maternal transmission. We have added this analysis, as well as the chrM:302 composition spectrum in the 1000G cell lines (which is strikingly similar to that observed in UKB, **Figure 5c**), to **Supplementary note 6**. The associated new supplementary figure is below:

Supplemental figure S5. Spectrum of transmission and chrM:302 length heteroplasmy variation across 1000G cell lines. **a.** Transmission of the 39 tested common heteroplasmy in 1000G between mother-offspring (left) and father-offspring (right) pairs across 602 families. **b.** Composition of indel variants at position chrM:302 across the 1000G cohort.

Taken together, we believe that our results provide confidence in our analytical approach and in our variant calls with respect to potential sequencing and PCR artifacts. We have added these points to **Supplementary note 6**, titled “assessed indel variants are unlikely to be artifacts”.

The reviewer’s point on the compositional form of the heteroplasmy data is a very important one. This “sum-to-1” property is a unique challenge for assessing sites with multiple co-existing alleles on the mitochondrial DNA. It is possible that this phenomenon may induce correlations between heteroplasmy levels for different alleles at a single site (e.g., 302 A>ACC and 302 A>AC, as the reviewer noted). However, this should not induce correlations across multiple mtDNA variant locations along the mtDNA (e.g., position 302, 567, 955, 16182) and would not explain the shared genetic architecture we observe. In this study, we take care to avoid conclusions drawn from correlations between heteroplasmy levels within the same site for this reason, and instead focus on correlations between each heteroplasmy and the nuclear genetic landscape or other phenotypes (e.g., mtCN, disease traits).

8. I’m intrigued but also puzzled about the model the authors propose for the regulation of 302 heteroplasmy levels in Figure 5. If one of the 302 alleles (let’s say A>AC) promotes additional DNA replication relative to the reference allele, should that not drive that mtDNA haplotype to fixation (and fast?, perhaps even over the lifetime of a single cell?). What prevents fixation of an mtDNA allele which has a replicative advantage?

At present we do not know why the observed heteroplasmy levels are so stable and why they don’t become fixed, and thus we can only speculate about the possibilities:

- The observed effect sizes (e.g., **Figure 4h, 4j, 5h**) are importantly aggregating effects over the course of the lifespan of the individuals included in the study. The per-replication effect sizes are likely a small fraction of these effect sizes.
- The effects we observe appear to only occur (1) if the mtDNA heteroplasmy is inherited and (2) if the individual carries a particular nucDNA genotype. Thus, it is possible that across several generations, fixation may not occur as the same maternal mtDNA lineage may interact with differing nuclear genetic backgrounds as a function of autosomal transmission from both maternal and paternal lineages.
- There may be balancing selection at play across generations for indel variants at these sites, potentially to attempt to maintain a reservoir of mtDNA variation in the population. We speculate on this in the final paragraph of our Discussion in a new sentence:

“In the current paper, we have shown that quantitative mtDNA traits in the population can be under both *cis*-acting control (via mtDNA variation) and *trans*-acting control (via nucDNA variation), and it is possible that these effects balance each other to maintain heteroplasmy across generations.”

9. As I said in my introduction, I enjoyed how much ground the manuscript covered. There were moments, though, where I felt the narrative flow or organization suddenly shift in an unexpected way. The best example of this was when the “Spectrum of mtDNA sequence variation across 253,583 individuals” section started. I don’t have a specific suggestion, but it feels as if the key results of this section could be introduced earlier or even first without significantly disrupting the flow of the manuscript.

In the revised Introduction of the manuscript, we now outline the major sections of the paper, stating that we begin by quantifying mtDNA copy number at biobank scale, and that we then move to calling mtDNA sequence variation. We have also now added transition sentences to several results sections (including the section referenced here) to improve exposition and flow.

10. Perhaps I missed it: are detailed counts of snps/indels at each position from UKB/AllofUs available as a data supplement or in a repository? This would obviously be a significant resource for the community.

We have produced a supplementary table with an allele frequency estimate for homoplasmic and heteroplasmic variants passing QC across both biobanks – see **supplementary tables 5 and 6**. Please note that AoU requires censoring of any counts corresponding to a group with 20 or fewer individuals. We will additionally make our AllofUs individual level calls available to those with CDR access in a new workspace at the time of publication, and are working with the UKB to release our corrected and raw mtCN measures as well as variant calls via the UKB data repository on the same timeline.

Referee #2 (Remarks to the Author):

In this study, Gupta and colleagues took advantage of existing large whole-genome DNA sequencing datasets (UKBB and AoU) to characterize variation between humans in terms of mitochondrial DNA (mtDNA) copy number and mtDNA sequence (heteroplasmy). Mitochondria plays essential roles in physiology, and it is therefore not surprising that defects in mitochondrial functions can result in human diseases (both rare and common). Here, the authors processed whole-genome sequence data to test previous reports that have connected mitochondria with human diseases, and to identify nuclear DNA sequence variants that associate (via GWAS) with mtDNA copy number, mtDNA coverage and heteroplasmy. The novelty of the study resides in the size of the datasets, the breadth of the results, as well as the technical and analytical robustness of the analyses. Findings in this study will have many (immediate?) implications for cell biologists working on mitochondria, and also potentially investigators working on human diseases in which mitochondria are involved. The manuscript is clear, and the details provided in the Methods section are sufficient.

Thank you for the positive feedback.

1. My only major comment is that I would have liked to see more links between the discoveries made in this paper and human phenotypic variation (beyond the mitochondria phenotypes described here).

Please note that we have indeed examined relationships between several mitochondrial phenotypes and other aspects of human phenotypic variation, including (1) correlations between mtCN_{adj} and 29 common diseases (**Figure 1f, Extended data figures 3k-3l**) as well as bidirectional MR for traits that showed correlation with mtCN_{adj}, (**Extended data figure 6**), (2) correlations between potential intermediate phenotypes and carrier status for any of 10

mtDNA heteroplasmies that are widely considered to be pathogenic for mitochondrial disease (**Figure 3**), and (3) associations between case-only heteroplasmy levels for the 39 common mtDNA heteroplasmies used for genetic analysis and risk for 29 common diseases (**Extended data figure 8**).

We additionally note that certain diseases of interest (e.g., Parkinson's disease) are poorly represented in UKB and thus we are not well-powered for the detection of an association with mtDNA traits in these instances.

I appreciate that this may be the next step, but there is at least one simple analysis that could be done: the authors could take the GWAS variants identified here (one-by-one or in a polygenic score) and perform pheWAS analyses in available biobanks. If it associates with diseases or traits, what types (metabolic, etc.)? And if it does not associate, why?

We have now performed the suggested PheWAS analysis by testing if any lead variant from genetic analyses of mtCN and mtDNA heteroplasmy is significantly associated with any of 2,277 blood biochemistry, ICD10 code, or phecode phenotypes in UKB on which genetic analyses were performed as part of the Pan UKBB initiative. We find very few notable patterns of locus overlap.

For the heteroplasmy traits, only three loci (out of 46) show overlap with any tested PheWAS trait: the *CDA* locus is associated with apolipoprotein A levels; the *PNP* locus is associated with alanine aminotransferase, aspartate aminotransferase, HbA1c, and HDL; and the *VSIG4* locus is associated with albumin and testosterone levels. Zero loci overlap any GWS locus for any tested disease trait in UK Biobank.

For mtCN_{adj}, 21 loci (out of 92) show any overlap with GWS variants for any tested PheWAS trait. Of these, there were only three GWS loci for mtCN which overlap disease traits: the *TERT* locus overlaps several neoplastic phenotypes, the *ABO* locus overlaps several cardiovascular and circulatory phenotypes (e.g., phlebitis and thrombophlebitis, pulmonary heart disease, hyperlipidemia), and the *JMJD1C* locus was identified for phenotypes including cholelithiasis and atrial fibrillation. All of the other 21 mtCN_{adj} loci overlap only the highly polygenic biomarker traits. Alkaline phosphatase is associated with 7 mtCN_{adj} loci, gamma glutamyltransferase is associated with 6, aspartate aminotransferase, sex hormone binding globulin, and HbA1c are associated with 5, alanine aminotransferase, apolipoprotein A, HDL, bilirubin, and protein are associated with 4, and the rest are associated with fewer. Two mtCN_{adj}-associated loci stand out as particularly pleiotropic when it comes to the biomarkers – *ABO* (19 blood biochemistry traits) and *JMJD1C* (18 blood biochemistry traits). We note that extremely few loci near mitochondria-localizing genes, or genes implicated in mtDNA disease, show any GWS signals across any of the tested 2,277 traits in UKB.

The bottom line is that we find many nuclear QTLs (near genes encoding mitochondrial-localized proteins) for mtDNA traits, yet those nuclear QTLs do not appear to be convincingly associated with many disease phenotypes. One potential explanation is that mitochondrial OXPHOS, the major biochemical pathway downstream of mtDNA, is highly robust to modest perturbations (see Vafai and Mootha Nature 2012). In fact, most Mendelian mitochondrial disorders of energy generation show autosomal recessive inheritance, and the few that are dominant are due to dominant negativity, and not due to haploinsufficiency. Classic studies have shown that the function of individual mitochondrial complexes must be inhibited dramatically before there is an observable effect on the "system," e.g., in some tissues, respiration does not detectably decline until complex IV declines by 70% (Rossignol et al. 1999 JBC):

In a previous study (Gupta et al. 2021 eLife) we find that nucDNA-encoded mitochondrial genes (heavily implicated in our current mtCN and heteroplasmy GWAS) are not enriched for age-related disease risk and are highly “haplosufficient” such that heterozygous loss of function variation in these genes tend to not be under appreciable negative selection:

Thus, it is possible that while common genetic variation (as seen in UKB and AofU) within nuclear-encoded mitochondrial genes can quantitatively impact the levels of mtDNA or the fraction of mtDNA heteroplasmy, these quantitative changes in mtDNA phenotypes are not sufficient to meaningfully impact the landscape of disease risk due to the robustness of the organelle to common perturbations.

2. A second related idea, which is more complicated but also more exciting, would be to test if these nuclear DNA sequence variants modify severity (penetrance, expressivity) in rare mitochondrial diseases. Probably outside the scope for this paper, but the idea could be discussed.

This is a great idea, but it would require a larger cohort of individuals with rare mitochondrial disease. We have added the following discussion paragraph:

“Our results have important implications for understanding rare mitochondrial diseases. First, our GWAS nominates new candidate genes for unsolved mitochondrial disease. *PNP*, which has not been previously linked to mtDNA disease, is an excellent example as it participates in purine metabolism in a way analogous to *TYMP* in pyrimidine metabolism, which is linked to mtDNA deletion/depletion syndromes; *PNP* is associated with mtCN_{adj} and the levels of 13 length heteroplasmy variants at three mtDNA sites. Second, we confirm that nearly 1:200 individuals carries a known pathogenic mtDNA variant (Elliott et al., 2008), and now report that such individuals can harbor intermediate phenotypes, e.g., the MELAS A3243G variant is associated with an increased risk for diabetes. Interestingly, the heteroplasmy distribution observed for the MELAS variant appears to be left-shifted, potentially suggesting negative selection as previously observed (Walker et al., 2020). Third, heteroplasmic mtDNA variants tend to be functionally “recessive” in that the number of copies of the wild-type allele is relevant for mtDNA variant pathogenicity. It is tempting to speculate that individuals with a higher mtCN polygenic score (PGS) may be more resilient to pathogenic mtDNA mutations, helping to explain some of the striking phenotypic variability observed between family members that carry the same maternally transmitted pathogenic mtDNA mutations (Lopez Sanchez et al., 2021). Larger, rare disease-focused studies will be required to determine the extent to which the nuclear variants we have identified can modify the penetrance of mtDNA mutations.”

3. Cross-ancestry GWAS results are presented throughout the manuscript. Is it because the authors did not find ancestry-specific genetic associations?

We perform cross-ancestry GWAS throughout the manuscript for two reasons: (1) this produces the maximum power for our association analysis, and (2) fine-mapping of cross ancestry meta-analyses can be improved over single ancestry associations, provided these meta-analyses are performed across uniformly processed genetic data (Kanai

et al. 2022 Cell Gen). To the first point, for instance, in AoU, each of the three ancestry groups have a relatively small sample size (~20k-50k), however the meta-analysis boosts our power for variant detection by combining association results from multiple genetic ancestry groups while minimizing the risk of population stratification as GWAS is still performed within-ancestry.

4. Were gene-based tests to identify rare variants also performed? If not, why? In particular, would rare coding variants help confirm some of the genes implicated (or prioritized) by the GWAS fine-mapping or co-localization results?

In the revised manuscript, we have now included rare-variant association tests using the UKB exomes via SAGIE-GENE+ for all tested traits (mtCN, coverage discrepancy, heteroplasmy) – see **Methods**. All analyses were run for multiple consequence groups (e.g., missense + pLoF, synonymous) with p-values obtained using SKAT, burden, and SKAT-O tests restricting to variants with MAF < 0.01, MAF < 0.001, and MAF < 0.0001. We have included gene-based results for all groups for any genes that achieve trait-wide significance for the Cauchy test (which combines evidence across consequence and MAF groups, see Zhou et al. Nat Gen 2022) as a new **Supplementary table 7**.

We observed interesting rare variant association study (RVAS) associations for mtCN, coverage discrepancy traits, and for chrM:302:A>AC heteroplasmy, with others showing few or no significant associations likely because most heteroplasmies tested were quite rare (many on the order of 2000-5000 samples only). For mtCN and coverage discrepancy traits we observed both convergence and novelty in the rare variant analysis, while the RVAS association landscape for chrM:302:A>AC strongly confirmed an association with *SSBP1*. We have also incorporated RVAS into our gene-assignment approach for GWAS, in which genes attaining RVAS genome-wide significance using Cauchy SKAT-O p-values which also have a nearby genome-wide significant GWAS association are now highlighted. This has the effect of specifically nominating *TOP3A* at the *MIEF2/TOP3A/SHMT1* locus for mtCN_{adj} GWAS. For all other loci with shared GWAS/RVAS associations, our previous fine-mapping-/proximity-based gene-assignment approach identified the same genes found via RVAS Cauchy test.

We have included RVAS “Manhattan” plots for mtCN at the MAF < 0.01 and MAF < 0.0001 cutoffs in **Extended data figure 5** and included QQ plots for chrM:302:A>AC in **Figure 5i**. We have also updated the text to reflect these results. For mtCN:

“We also performed a gene-based rare variant association study (RVAS) for mtCN_{adj} in UKB (**Methods, Supplementary table 7**). In several instances we find convergence with GWAS, including associations with ultra-rare (MAF < 0.0001) missense or loss of function (LoF) variation in *TWINK* and *TFAM* (**Extended data figure 5c**). RVAS provided clarity to other GWAS loci with uncertain gene assignments (e.g., highlighting *TOP3A* in a locus containing several genes, **Figure 1d**) and identified several associations with genes not identified by GWAS. For instance, we found associations with the burden of rare protein-altering variation within genes previously linked to Mendelian mtDNA deletion or depletion disease (*OMA1*, *SAMHD1*), as well as associations with genes unlinked to mitochondria (e.g., *MILR1*) (**Extended data figure 5d**).”

For coverage discrepancy traits:

“RVAS identified additional associations between the levels of missense or LoF variation in novel genes and the 7S DNA and DNA flap coverage discordance, including *OMA1* (**Supplementary table 7**).”

And for the chrM:302:A>AC heteroplasmy:

“Nuclear genetic analyses for chrM:302:A,AC, the most common length heteroplasmy, nominated several genes relevant for mtDNA replication and nucleotide balance (e.g., *SSBP1*, identified by GWAS and corroborated by ultra-rare RVAS, **Figure 5g, 5i**), including several genes not identified in GWAS for other heteroplasmic sites (*CDA*, *MTPAP*, *TFAM*, *TEFM*, *LONP1*, *MCAT*; **Figure 4e, 5g**).”

5. Fig.5e. It looks like the one sample analyze in (e) does not have mitochondria with the reference allele (GmAG7, orange). In c and d, are they individuals with a similar profile (it does not seem, although maybe it is a super rare profile)? In other words, are they individuals in the large WGS datasets with heteroplasmy at chrM:302 that do not have the reference (orange?) allele?

We appreciate this extremely astute observation. This is due to the original method we implemented for quantifying the amount of heteroplasmy attributable to the reference allele, which was computed using the fraction of heteroplasmy not attributed to other QC-pass alleles. However, on further scrutiny it appears that this residual fraction of heteroplasmy for chrM:302 seen in many samples in the previous **Figure 5d, 5e** can be explained by alleles that fail QC (e.g., heteroplasmy < 5%) and were removed from the variant callset. Since QC is performed after heteroplasmy estimation, constructing composition plots on post-QC data erroneously results in any heteroplasmy attributed to “QC-fail” alleles being mis-labeled as reference. We have updated our approach to account for estimated heteroplasmy for QC-fail alleles and place these in the “Other” category. Additionally, in many instances the residual fraction of heteroplasmy attributed to the reference allele was < 5%, which is below our study-wide threshold for confidently calling a variant. We now label any reference heteroplasmy fractions < 5% in the “Other” category as well. While the overall figure looks similar, it now appears that ~40% of samples have undetectable levels of the chrM:302 reference allele. We have updated our figures and **Methods** accordingly. Thank you for picking this up!

6. (Minor) Provide complete legends for the supplementary tables.

We have now updated our supplementary tables with complete legends for the schema of each table.

Signed: Guillaume Lettre

Referee #3 (Remarks to the Author):

In “Nuclear genetic control of mtDNA copy number and heteroplasmy in humans” Gupta, Neale, Mootha and colleagues perform a large scale human genetics analysis to define common genetic variants that alter mtDNA copy number, mtDNA heteroplasmy and their phenotypic consequences. For this analysis they develop a new approach for calling mitochondrial DNA mutations that relies on a per individual consensus sequence rather than a reference genome. They apply this pipeline to whole genome sequencing (WGS) data from ~175,000 individuals from the UK Biobank and ~100,000 individuals from the All of Us cohort. They make several striking observations from this dataset including:

- (1) that much of the prior literature on mtDNA copy number in blood is confounded by a failure to fully account for blood cell counts
- (2) that the sequencing depth of coverage at the 7s DNA/DNA primer/RNA primer region can provide insights into mechanisms underlying mtDNA replication
- (3) that population scale datasets can be used to refine genotype/phenotype associations for mitochondrial disorders like MELAS
- (4) that many MTDNA heteroplasmy have both shared and distinct nuclear genetic determinants

Overall this analysis represents a landmark in the field of mitochondrial genetics both because of the scale of the data and the rigor of the analyses undertaken in particular the author's attention to minimizing confounding in the WGS derived mtDNA phenotypes and the potential artifacts introduced by nucDNA regions of mtDNA origin (NMUTs) in their genetic analyses.

We thank the reviewer for the very positive comments.

The work could be strengthened in several ways:

- (1) The authors derive mtDNA phenotypes from WGS data that is aligned to the hg38 reference genome however they do not take advantage of the dataset for their UK Biobank genetic analyses. Specifically, the genetic association studies in the UK Biobank are based on the hg19 imputed data from UKB-provided imputed v3 variants. As recently described by Halldorsson et al (Nature 2022) An analysis based on genetic variants identified with the whole genome sequencing data would likely identify more associated genetic variants (eg those that are poorly imputed) and also enhance discovery in non-European ancestry individuals where imputation panels are less accurate (particularly in rare variants). This analysis would perhaps identify germline genetic variants that are ancestry specific.

Thank you for this suggestion – we appreciate the limitations of imputation particularly in the context of rare variation in individuals in non-European genetic ancestry groups. We agree that rare variation is likely to be better captured with sequencing-derived data rather than imputation from genotype chip data, and now perform comprehensive rare-variant association testing using hg38-aligned sequencing-derived variant calls (see response to point (2)).

Regarding the benefits in common variant GWAS in non-European populations in UKB, we note that the vast majority of the UKB study population was assigned to the EUR genetic ancestry group (**Extended data figure 1**), with our largest GWAS (mtCN) containing 155998 (>95%) EUR samples and only 3326, 2138, 958, 571, and 381 samples from individuals assigned to CSA, AFR, EAS, MID, and AMR respectively. For most heteroplasmy traits, the numbers are far smaller (**Supplementary table 1**). Thus, due to the properties of the cohort, despite performing careful multi-ancestry analysis and cross-ancestry meta-analysis, the vast majority of our observed genetic signal in UKB is derived from individuals in the EUR group.

In contrast, AoU consists of a much more genetically diverse cohort, with our largest genetic analysis (heteroplasmy of chrM:302:A,AC) having contributions of ~56% from EUR individuals, ~20% each from AFR and AMR individuals, and the rest from smaller genetic ancestry groups. By absolute numbers and proportions, the AoU cohort offers substantially greater power than UKB for individuals assigned non-EUR ancestry groups. Importantly, our analysis of mtDNA heteroplasmy in AoU was performed using variant calls derived from hg38-aligned WGS, avoiding any potential concerns about performance of imputation in non-European ancestry groups. We believe that an evaluation of ancestry-specific effects identified in AoU using WGS-based variant calls can serve as an upper bound for what we may gain from switching to WGS-based calls in UKB:

To assess the discovery of ancestry-specific effect sizes in the diverse AoU cohort, we evaluated p-values from the Cochran's Q test for heterogeneity across ancestry groups (these AoU heterogeneity test statistics will be made available as part of AoU summary statistics on publication). Of the 1521 genome-wide significant associations identified across any tested mtDNA heteroplasmy meta-analysis, the lowest p-value we find is 0.0011, well above the liberal analysis-wide threshold of $0.05/1521 = 0.000033$. Thus, we find no convincing evidence of cross-ancestry heterogeneity in effect size estimates in AoU.

To assess the discovery of associations identified in an ancestry-specific manner due to ancestry-enriched variation in AoU, for each of the 1114 genome-wide significant associations identified in any tested mtDNA heteroplasmy in any of the 5 included individual populations, we compared the MAF of the variant associated with the trait in the specific population to the MAF of the variant in the EUR group. We found only 9 distinct variants across 3 loci with a >10 fold MAF enrichment in the population with a detected association versus EUR in AoU. In 7 of these 9 cases, these variants were identified with $INFO > 0.8$ and passing all QC in UKB, with UKB estimated MAF largely concordant with AoU estimated MAF. The AoU variant chr19:626696:C:CA (not in UKB) is adjacent to chr19:616996:G:C (in UKB), both of which are GWS for chrM:302:A,AC in AFR in AoU. The AoU variant chr17:64570073:C:T (not in UKB) is adjacent to POLG2, a locus identified for several heteroplasmy and copy number traits in UKB (**Figure 4f**). This indicates that no ancestry-specific associations using the WGS-derived variant calls in the more diverse AoU cohort were missed in UKB using imputed variants. We have produced a table showing the 9 ancestry-enriched GWS variants in AoU and their mapping to UKB (var_ukb_b37 refers to the variant found in UKB in b37; all columns with the aou prefix are based on AoU data; all columns with the ukb suffix are based on UKB data):

var_b38	var_ukb_b37	aou_pop_with_association	aou_pop_MAF	aou_EUR_MAF	aou_P	ukb_qc_pass	ukb_info	ukb_pop_MAF	ukb_EUR_MAF
chr1:1422841:G:A	1:1358221	afr	0.011046	3.03E-04	3.74E-08	TRUE	0.89018	0.015026	4.43E-04
chr1:1423292:G:A	1:1358672	afr	0.011046	3.03E-04	3.74E-08	TRUE	0.81241	0.013817	7.49E-04
chr1:1424179:C:G	1:1359559	afr	0.011061	3.03E-04	3.99E-08	TRUE	0.90228	0.013782	4.37E-04
chr1:1424786:G:A	1:1360166	afr	0.011046	3.03E-04	3.74E-08	TRUE	0.83252	0.011817	4.38E-04
chr17:64570073:C:T	Not found	amr	0.037112	0.0022455	3.24E-12	NA	NA	NA	NA
chr19:614222:G:A	19:614222	afr	0.11515	8.79E-04	1.05E-08	TRUE	0.87345	0.12887	2.03E-04
chr19:616973:C:T	19:616973	afr	0.094668	0.00106	1.67E-09	TRUE	0.9029	0.1093	2.23E-04
chr19:616996:G:C	19:616996	afr	0.1031	6.98E-04	1.11E-10	TRUE	0.90212	0.11066	1.46E-04
chr19:626696:C:CA	Not found	afr	0.10779	0.0015343	4.36E-09	NA	NA	NA	NA

Finally, we observe very strong replication of GWAS effect sizes between cross-ancestry meta-analyses performed in UKB versus AoU (**Extended data figure 11c**). Inspection of cross-ancestry meta-analyses from AoU showed a similar association landscape to that seen for cross-ancestry meta-analyses for the same traits in UKB (though UKB had better power in total), again indicating that use of imputed genotypes in UKB likely did not result in a failure to identify genetic associations in the smaller ancestry groups (for which AoU had better power).

Given that we have already analyzed AoU, which has greater power for genetic discovery in non-European genetic ancestry groups than UKB, using hg38-aligned WGS-derived variant calls and (1) do not find evidence of ancestry-enriched nuclear genetic variants associating with mtDNA phenotypes in AoU that were missed by imputed variants in UKB, (2) observe a similar genetic architecture between UKB and AoU, and (3) now perform separate rare-variant association testing using sequencing-derived variant calls in UKB, we believe that the steps of performing new variant QC, reconstructing reference data and pipelines, rerunning common variant GWAS, redoing fine-mapping and more are unlikely to yield additional novel common genetic insights in the EUR-predominant UKB cohort.

To ensure that genome build is not an issue for the use of GWAS summary statistics generated as part of this study, we have performed LiftOver using the recently released bcftools plugin (<https://github.com/freeseek/score>) and have produced an annotation file allowing users to easily use hg19 or hg38 with the UKB association data.

(2) Full use of genetic sequencing data would also enable rare variant association analyses in addition to the common variant association analyses. Such analyses would perhaps identify a convergence of common and rare genetic variants at a subset of loci and the resulting allelic series would further strengthen the conclusion that specific genes are causal.

In the revised manuscript, we now include comprehensive, gene-based rare variant association tests using SAIGE-GENE+ using UKB exome data for all tested traits (mtCN, coverage discrepancy, and heteroplasmy) – see **Methods**. All analyses were run for multiple consequence groups (e.g., missense + pLoF, synonymous) with p-values obtained using SKAT, burden, and SKAT-O tests restricting to variants with $MAF < 0.01$, $MAF < 0.001$, and $MAF < 0.0001$. We have included gene-based results for all groups for any genes that achieve trait-wide significance for the Cauchy test (which combines evidence across consequence and MAF groups, see Zhou et al. Nat Gen 2022) as a new **Supplementary table 7**.

We observed interesting RVAS associations for mtCN, coverage discrepancy traits, and for chrM:302:A>AC heteroplasmy, with others showing few or no significant associations likely because most heteroplasmies tested were quite rare (on the order of 2000-5000 samples only). For mtCN and coverage discrepancy traits we observed both convergence and novelty in the rare variant analysis, while the RVAS association landscape for chrM:302:A>AC strongly confirmed an association with *SSBP1*. We have also incorporated RVAS into our gene-assignment approach for GWAS, in which genes attaining RVAS genome-wide significance using Cauchy SKAT-O p-values which also have a nearby genome-wide significant GWAS association are now highlighted. This has the effect of additionally nominating *TOP3A* at the MIEF2/TOP3A/SHMT1 locus for mtCN_{adj} GWAS. For all other loci with shared GWAS/RVAS associations, our previous fine-mapping-/proximity-based gene-assignment approach identified the same genes found via the RVAS Cauchy test.

We have included RVAS “Manhattan” plots for mtCN at the $MAF < 0.01$ and $MAF < 0.0001$ cutoffs in **Extended data figure 5** and included QQ plots for chrM:302:A>AC in **Figure 5i**. We have also updated the text to reflect these results. For mtCN:

“We also performed a gene-based rare variant association study (RVAS) for mtCN_{adj} in UKB (**Methods, Supplementary table 7**). In several instances we find convergence with GWAS, including associations with ultra-rare ($MAF < 0.0001$) missense or loss of function (LoF) variation in *TWINK* and *TFAM* (**Extended data figure 5c**). RVAS provided clarity to other GWAS loci with uncertain gene assignments (e.g., highlighting *TOP3A* in a locus containing several genes, **Figure 1d**) and identified several associations with genes not identified by GWAS. For instance, we found associations with the burden of rare protein-altering variation within genes previously linked to Mendelian mtDNA deletion or depletion disease (*OMA1*, *SAMHD1*), as well as associations with genes unlinked to mitochondria (e.g., *MILR1*) (**Extended data figure 5d**).”

For coverage discrepancy traits:

“RVAS identified additional associations between the levels of missense or LoF variation in novel genes and the 7S DNA and DNA flap coverage discordance, including *OMA1* (**Supplementary table 7**).”

And for the chrM:302:A>AC heteroplasmy:

“Nuclear genetic analyses for chrM:302:A,AC, the most common length heteroplasmy, nominated several genes relevant for mtDNA replication and nucleotide balance (e.g., *SSBP1*, identified by GWAS and corroborated by ultra-

rare RVAS, **Figure 5g, 5i**), including several genes not identified in GWAS for other heteroplasmic sites (*CDA*, *MTPAP*, *TFAM*, *TEFM*, *LONP1*, *MCAT*; **Figure 4e, 5g**).”

(3) The variation in mtCN by month of assessment and time of day presented in Extended Data 3 is fascinating! Are there any germline genetic loci that associate with the residual between these two factors and the median value? It would be quite novel to identify genetic determinants of circadian rhythm or seasonal disorders identified through analyses of blood count corrected mtCN.

We agree that these associations are quite fascinating, but this analysis is beyond the scope of the current work which is already quite lengthy. Further, we lack repeated measures of mtDNA in UK Biobank, so it is challenging to look at individuals across time.

(4) A time-to-event model of incident events (eg mtCN predicting incident events with a cox proportional-hazards model) may be more appropriate than a logistic regression for the mtDNA CN analyses presented in figure 1f and the associated extended data figures given mtDNA CN changes with time and may change in response to particular disease events. Although the corrected mtCN association with MI is in the opposite direction than the uncorrected version, it would appear to be significant even correcting for multiple hypothesis testing ($p=1.4 \times 10^{-5}$). What do the authors make of this? Is the mendelian randomization for MI supportive of this link?

Thank you for these suggestions – the significant, now **positive**, associations between mtCN_{adj} and cardiovascular phenotypes (MI, Angina, High cholesterol, Ischemic heart disease) as well as the persistent negative correlation with osteoarthritis are interesting. To gain insight into this, as suggested we have now performed bidirectional MR using the same approach taken previously for neutrophil count, performing these analyses for all 5 phenotypes that showed significant correlations with mtCN_{adj}. After corrections for multiple comparisons, we find evidence of a causal relationship for high cholesterol on mtCN_{adj} but not vice versa, with no other causal relationships detected in either direction. The unidirectional nature of the MR association for high cholesterol on mtCN suggests that high cholesterol levels may impact measured blood mtCN_{adj}. This could be occurring via components of cell type composition that we were unable to account for, could be governed by treatment effects (e.g., some effect of cholesterol lowering medication use on mtCN), or could be acting via a direct molecular mechanism in which cholesterol directly influences mtCN – we can only speculate at this time.

Given that elevated cholesterol levels are likely observed in many individuals with the other cardiovascular phenotypes (MI, ischemic heart disease), it is possible that this is contributing to the now-positive association between mtCN_{adj} and the tested cardiovascular traits. That MR is not consistent with a causal role for cardiovascular disease or osteoarthritis in mtCN_{adj}, could be due to low power, or could be consistent with the notion that the observed phenotypic correlation between mtCN_{adj} and MI, angina, ischemic heart disease, and osteoarthritis is governed by yet-unidentified confounding. The high heterogeneity we observe in the relationship between SNP effects suggests against a simple causal relationship linking mtCN_{adj} to these traits and supports a more complex relationship. We have produced **Extended data figure 6** showcasing these results. We also include a zoomed-in version of the plot showing the mtCN versus "high cholesterol" effect sizes for "high cholesterol" genome-wide significant SNPs (**Extended data figure 6k**), which shows visually robust positive correlation that appears to strengthen with blood composition correction. This figure is below:

Extended data figure 6. Bidirectional MR within UKB between mtCN and associated disease traits.

Correlation between effect sizes for lead SNPs detected for raw (left) and adjusted (right) mtCN between the respective mtCN phenotype and **a**. Osteoarthritis, **c**. Angina, **e**. Myocardial infarction, **g**. Ischemic heart disease, **i**. High cholesterol. Correlation between effect sizes for lead SNPs detected for **b**. Osteoarthritis, **d**. Angina, **f**. Myocardial infarction, **h**. Ischemic heart disease, **j**. High cholesterol and raw (left) and adjusted (right) mtCN. A zoomed-in version of **j** is shown in panel **k**. In all panels, error bars represent +/- beta SE, dotted line corresponds to inverse variance weighted least squared regression line; inset corresponds to regression p-value. Regression fits were performed separately for loci genome-wide significant for both mtCN_{raw} and mtCN_{adj} (black) and for loci specific to each (red) for the analysis of mtCN on disease traits.

Finally, we have added these results in the results section:

“We extended these analyses to 24 additional common diseases, finding that in total, 20 showed significantly increased risk with reduced mtCN_{raw}; after correction for blood cell composition, the inverse correlations disappeared

for all traits except for osteoarthritis (**Extended data figure 3k**). Associations with four cardiovascular disease traits even changed direction with $mtCN_{adj}$ now positively correlated with increased risk. In all five cases, Mendelian randomization (MR) did not support a causal role for $mtCN_{raw}$ or $mtCN_{adj}$ after correcting for multiple tests (**Extended data figure 6**).”

and:

“For the few associations that survive blood composition corrections (**Extended data figure 3k**), other mechanisms may be at play. Indeed, MR suggests reverse causation or shows high heterogeneity for other traits, suggesting against simple forward causal relationships in these instances (**Extended data figure 6b**).”

Regarding the time-to-event model, we believe that this interesting idea is beyond the scope of the present work, which is primarily focused on the identification of the nuclear genetic architecture associated with mtDNA phenotypes.

(5) How do the authors interpret the data presented in Fig 3 with respect to the pathogenicity of these variants in MELAS syndrome? Would they suggest that only one of the 10 variants is in fact pathogenic?

We apologize – the previous Figure 3 and its legend were unclear. All 10 of these mtDNA variants are widely accepted as being associated with mtDNA diseases (e.g., are classical / canonical mutations underlying MELAS, MERRF, aminoglycoside-induced ototoxicity, Leber’s hereditary optic neuropathy (LHON)). What is novel here is that individuals that happen to be carriers of the A3243G variant (classically associated with MELAS) appear to be at risk for diabetes. We have now updated the figure and the legend to attempt to clarify this:

Figure 3. Carrier frequencies and intermediate phenotypes for pathogenic mtDNA mutations assessed in UKB.

Table shows carrier frequencies for 10 well-known pathogenic mutations in UKB, with heteroplasmy distributions plotted as jittered points, and annotations corresponding to canonical disease(s) associated with variants. Panels show mean Hemoglobin A1c, triglyceride levels, auditory threshold (via speech recognition threshold test), and visual impairment (via vision test measured as logMAR) among mtDNA pathogenic variant carriers. Only points corresponding to more than 10 measurements are shown. Vertical lines represent per-trait means among individuals with none of the 10 pathogenic mutations detected. Error bars correspond to 1SE. A-IOT=aminoglycoside-induced ototoxicity; LHON=Leber’s hereditary optic neuropathy; MELAS=mitochondrial encephalomyopathy, lactic acidosis, and stroke-like episodes; MERRF=myoclonic epilepsy with ragged red fibers; LS=Leigh syndrome; NARP=neuropathy, ataxia, retinitis pigmentosa.

We cannot conclude that the other tested variants are not pathogenic. For example, the chrM:1555:A,G variant is associated with aminoglycoside induced deafness, and perhaps those in the population carrying this variant simply have not had this antibiotic exposure. These variants are widely accepted as being pathogenic for mitochondrial disease when observed in the appropriate clinical context, are known to show incomplete penetrance, and are quite rare in UKB, thus limiting our power for the detection of more subtle associations with intermediate phenotypes. Several of variants we assess (e.g., LHON variants chrM:14484:T,C and chrM:11778:G,A; aminoglycoside-induced ototoxicity variant chrM:1555:A,G) show highly variable penetrance even within pedigrees, with LHON showing penetrance ranges from 1%-73% across different pedigrees (Lopez Sanchez et al. 2021 AJHG). With this analysis we can simply conclude that there is evidence of intermediate phenotypes among carriers of chrM:3243:A,G. We have added a sentence to this section of the results:

“Due to their low frequency of detection in the UKB sample, we do not have the statistical power to exclude the presence of more subtle intermediate phenotypes among the other tested variants.”

(6) In Fig 3, it is also notable that in the one variant that appears to be pathogenic (chrM:3243:A,G), the heteroplasmy is <25% for most samples. This may reflect selection against this variant in all cells or perhaps a cell lineage (as you showed in Walker, 2020 with the depletion of MELAS variants in T-cells). Several of the more common putative MELAS variants (eg chrM14484:T,C) appear to have a bimodal distribution. Is the phenotype different in carriers of these variants who have a low heteroplasmy compared to those with a high heteroplasmy?

We apologize again for any lack of clarity with our description of the function of the 10 selected known-pathogenic mtDNA variants; as mentioned for (5), chrM:3243:A,G is the canonical pathogenic MELAS variant. We agree that it is interesting that the MELAS variant shows a “left-shifted” heteroplasmy distribution and is indeed consistent with results from Walker et al. 2020. We have now referred to this in our discussion:

“Our results have important implications for understanding rare mitochondrial diseases. First, our GWAS nominates new candidate genes for unsolved mitochondrial disease. *PNP*, which has not been previously linked to mtDNA disease, is an excellent example as it participates in purine metabolism in a way analogous to *TYMP* in pyrimidine metabolism, which is linked to mtDNA deletion/depletion syndromes; *PNP* is associated with mtCN_{adj} and the levels of 13 length heteroplasmy variants at three mtDNA sites. Second, we confirm that nearly 1:200 individuals carries a known pathogenic mtDNA variant (Elliott et al., 2008), and now report that such individuals can harbor intermediate phenotypes, e.g., the MELAS A3243G variant is associated with an increased risk for diabetes. Interestingly, the heteroplasmy distribution observed for the MELAS variant appears to be left-shifted, potentially suggesting negative selection as previously observed (Walker et al., 2020). Third, heteroplasmic mtDNA variants tend to be functionally “recessive” in that the number of copies of the wild-type allele is relevant for mtDNA variant pathogenicity. It is tempting to speculate that individuals with a higher mtCN polygenic score (PGS) may be more resilient to pathogenic mtDNA mutations, helping to explain some of the striking phenotypic variability observed between family members that carry the same maternally transmitted pathogenic mtDNA mutations (Lopez Sanchez et al., 2021). Larger, rare disease-focused studies will be required to determine the extent to which the nuclear variants we have identified can modify the penetrance of mtDNA mutations.”

With respect to the variants showing a bimodal distribution (chrM:1555, chrM:14484, chrM:11778), we have now examined the mean levels of each of the four tested phenotypes in individuals carrying low levels of each heteroplasmy ($0.05 \leq HL < 0.5$) and those carrying high levels ($0.5 \leq HL \leq 1$), finding little evidence of difference (no tests showed $FDR q < 0.1$ within this analysis):

We believe this may be attributable to low penetrance for these diseases (AIOT and LHON), potential environmental modifiers, and relatively limited power to detect small effect sizes. It is also likely that ascertainment bias is at play in the UK Biobank cohort to some degree, with those presenting with severe symptoms of mitochondrial disease potentially less likely to enroll in UKB.

(7) Can the authors conduct a phenome-wide association study to identify what other ICD code, biomarker or molecular phenotypes (eg NMR metabolomics) chrM:3243 is associated with and thereby expand the phenotypic spectrum associated with MELAS?

Thank you for this interesting suggestion. As we only find 53 individuals in our dataset carrying the MELAS variant, our power is limited to identify associations particularly after a phenome-wide scan testing thousands of hypotheses. While the NMR metabolomics data is particularly interesting in this context, there are only 16 individuals who carry the MELAS variant and who also have any NMR metabolomics data available, a number that is likely too small to confidently detect subtle perturbations in the metabolome. To attempt to expand the phenotype space while remaining cognizant of the small number of individuals with the MELAS variant detected, we tested:

1. All 30 biomarker phenotypes for differences between MELAS variant carriers versus individuals without any of the 10 mtDNA known pathogenic variants using a similar approach to that taken for hearing, vision, HbA1c, and triglycerides. We find the following:

These results indicate that the elevation in HbA1c is the most significant association identified among the biomarkers, concordant with our primary analysis. Triglycerides, urea, and cystatin C showed nominally significant elevations and CRP showing a nominally significant decline among MELAS variant carriers which do not survive corrections for testing multiple hypotheses.

2. The same 29 curated case/control phecode/ICD code-based phenotypes corresponding to common diseases which we assessed in our analysis of mtCN_{raw} versus mtCN_{adj}. We used the same approach, restricting to those phenotypes for which there were at least 3 MELAS carriers in the case group.

These results highlight T1D, T2D, and hearing aid use risk as significantly elevated among MELAS variant carriers, strongly concordant with our primary results assessing HbA1c, triglycerides, and hearing test performance. Risk for chronic renal failure appears nominally greater than 0 though does not survive corrections for testing multiple hypotheses; this would be concordant with the associations with diabetes and diabetes-related biomarkers. We note that the number of phenotypes shown above is substantially smaller than the number of phenotypes tested for associations with mtCN because of our criteria requiring at least 3 MELAS variant carriers in the case group.

As mentioned, given the small sample sizes for this analysis, we are limited by low power to observe smaller perturbations across even these selected phenotypes. We believe these analyses will likely be more fruitful with identification of greater numbers of MELAS variant carriers.

Minor considerations:

(8) An extended data figure (or figure part) illustrating the main questions/different kinds of mitochondrial DNA variation (eg copy number vs heteroplasmy vs replication events) that you are extracting may be helpful to make the manuscript easier to follow for the non-specialist.

Thank you for this suggestion. Our manuscript is already quite long, but we have now added text to the introduction which we hope will help readers with an overview of the mtDNA traits presented in the manuscript:

“Here, we characterize the spectrum of mtCN and heteroplasmy across ~300,000 individuals spanning 6 ancestry groups in UK Biobank (UKB) and AllofUs (AoU) and identify their nuclear genetic correlates. After rigorous correction for blood cell composition, we find that mtCN declines with age, is influenced by numerous nuclear genetic loci, and in contrast to prior studies, is not reduced in most common diseases. We observe that mtDNA coverage is heterogeneous in regions corresponding to replication intermediates, and that the degree of this heterogeneity is itself under nuclear genetic control. We then turn to an investigation of mtDNA variants, finding that ~1:192 individuals carries mtDNA variants widely accepted as being pathogenic. We subsequently characterize the spectrum of heteroplasmic mtDNA variants in the population, finding two general patterns: heteroplasmic single nucleotide variants (SNVs) tend to be somatic and accumulate with age; and heteroplasmic indels occur most frequently in the non-coding region, do not vary with age, and are transmitted quantitatively as mixtures in the same maternal lineage. We show that these indel heteroplasmies are present in most individuals and that their levels are controlled by 42 nuclear genetic loci, many with established roles in mtDNA replication and maintenance as well as mitochondrial genes with no prior links to mtDNA biology.”

(9) The variation that exists by assessment center (Extended data 3J) is notable. Does this persist after correcting for other factors identified (like draw time, month of assessment and so forth)? Does it correlate with socioeconomic factors, differences in ancestry or other demographic differences between assessment center?

We have implemented 5 correction models to test for the marginal effects of correcting for a variety of variables on the relationship between mtCN and assessment center:

1. blood covariates (as in Extended data 3j)
2. blood + technical variables (i.e., draw time, month of assessment, etc.)
3. blood + technical + age/sex/age²/interactions
4. blood + technical + age/sex + genetic ancestry variables (PCs 1-20, population assignment)
5. blood + technical + age/sex + genetic ancestry + Townsend Deprivation Index

We find that addition of technical covariates substantially attenuates, but does not fully eliminate, the relationship between blood-corrected mtCN and assessment center. Addition of other terms does not, in most cases, have a substantial effect on the rescaled adjusted mtCN or on its relationship with assessment center.

(10) Do the authors observe any differences between the two UK biobank WGS sequencing centers with respect to mitochondrial copy number?

Thank you for this helpful suggestion. We have now extracted sequencing center information from each of the genomic data files and assessed the relationship of adjusted mtCN with WGS sequencing center and encouragingly see very little difference in the average estimates between the two. We have updated **Extended Data Figure 3d** to now show densities of adjusted mtCN as a function of sequencing center as well as in the aggregate data, and we have reproduced this updated plot below:

To confirm robustness, we also performed the mtCN GWAS including sequencing center as a covariate and find no discernable difference (EUR analysis in UKB shown below):

(11) The hg38 human reference genome was released ten years ago in 2013. Contemporary human genetic analyses typically leverage hg38 in the absence of a compelling reason not to. It is unusual to have both hg19 and hg38 coordinates presented in the same manuscript. Here both are included without clear notation for the reader to follow. Several figures include both for example Figure 4g and 4i the y-axis refers to hg38 and the x axis refers to hg19.

We have now updated the genetic coordinates used in all visualizations to hg38, and included both hg19 and hg38 coordinates in all supplementary tables. We have also now produced a reference file for all UKB variants in hg19 and in hg38 for the community. As mentioned previously, all AoU GWAS were performed using variant calls from hg38-aligned WGS data.

(12) Fig 3: why include any variant with fewer than 10 carriers on the figure if associated phenotype data is not presented? It may be more space efficient to include only rows with phenotype data presented

Thank you for this comment. Carrier frequencies for these 10 variants were assessed in the general population in a classic and important study by Elliott et al. 2008 AJHG; part of our aim for this figure was to provide an updated assessment of these carrier frequencies using UKB. We have retained rows corresponding to less common variants to provide this snapshot of frequencies and heteroplasmy distributions for all tested pathogenic variants, with another important goal being assessment of a set intermediate phenotypes in those variants for which we had sufficient power. We have clarified this dual goal by adjusting the text associated with this figure:

“We next considered mtDNA sequence variation in UKB (**Methods**), with an initial focus on well-established, disease associated mtDNA variants. We began by assessing the carrier rates for ten common pathogenic mtDNA variants associated with maternally inherited diseases, including Leber’s hereditary optic neuropathy, mitochondrial encephalopathy, lactic acidosis, and stroke-like episodes (MELAS), and aminoglycoside-induced ototoxicity (**Figure 3**). We find that ~1:192 individuals in UKB carry at least one of the ten pathogenic mtDNA variants, in agreement with a previous estimate of 1:200 (Elliott et al., 2008).

An open question is whether individuals carrying rare pathogenic mtDNA variants in the population exhibit intermediate disease phenotypes. We can now address this thanks to the rich phenotyping in UKB.”

(13) On page 51, line 20: there appears to be an erroneous inclusion of the phrase “Click or tap here to enter text” which is likely a typo.

Thank you for finding this oversight – we have removed this erroneous text.

(14) Page 12, line 43-44 “It is notable that prior studies have suggested that length variation at” [...], would consider referencing the prior study here.

We have added two citations to this sentence for improved clarity, with further references included in the following two sentences.

(15) Data availability: individual level data can be shared within the All of Us workbench with any registered user. I would suggest depositing the individual mtCN and mt heteroplasmy call data within an All of Us Researcher Workbench workspace that can be viewed by any registered All of Us researcher as has been done recently for Master, Nature Medicine 2022; which was published in the Researcher Workbench Workspace Library (<https://workbench.researchallofus.org/library>).

We are very happy to do this and will make these individual level data available in AoU at the time of publication.

-Alexander Bick

Reviewer Reports on the First Revision:

Referees' comments:

Referee #1 (Remarks to the Author):

The authors have satisfactorily addressed my questions and I have no further concerns. I commend them on a thorough job.

-Ed Reznik

Referee #2 (Remarks to the Author):

The authors have appropriately addressed my comments. I don't have additional concerns.
Guillaume Lettre

Referee #3 (Remarks to the Author):

The authors have robustly addressed all of the points I raised. I appreciate their excellent work and have no additional feedback.